# PROBLEM: SHARDED RESILIENT TRANSACTION PROCESSING WITH MINIMAL COSTS

## Abstract

To enable scalable resilient blockchain systems, several powerful general-purpose approaches toward sharding such systems have been demonstrated. Unfortunately, these approaches all come with substantial costs for ordering and execution of multi-shard transactions.

In this work, we ask whether one can achieve significant cost reductions for processing multi-shard transactions by limiting the type of workloads supported. To initiate the study of this problem, we propose core-CHIMERA (CCHIMERA). CCHIMERA uses *strict UTXO-based environmental requirements* to enable powerful multi-shard transaction processing with an absolute minimum amount of coordination between shards. In the environment we designed CCHIMERA for, CCHIMERA will operate *perfectly* with respect to all transactions proposed and approved by well-behaved clients, but does not provide any other guarantees.

To illustrate that CCHIMERA-like protocols can also be of use in environments with faulty clients, we also demonstrate *two* generalizations of CCHIMERA, *optimistic*-CHIMERA and *resilient*-CHIMERA, that make different tradeoffs in complexity and costs when dealing with faulty behavior and attacks. Finally, we compare these three protocols and show their potential scalability and performance benefits over state-of-the-art general-purpose systems. These results underline the importance of the study of specialized approaches toward sharding in resilient systems.

## 1 Introduction

The advent of blockchain applications and technology has rejuvenated interest of companies, governments, and developers in resilient distributed fully-replicated systems and the distributed ledger technology (DLT) that powers them. Indeed, in the last decade we have seen a surge of interest in reimagining systems and build them using DLT networks. Examples can be found in the financial and banking sector [15, 39, 52], IoT [45], health care [28, 40], supply chain tracking, advertising, and in databases [5,23,30,31,49,50]. This wide interest is easily explained, as blokchains promise to improve resilience against both failures and malicious behavior, while enabling the federated management of data by many participants.

To illustrate this, we look at the financial sector. Current traditional banking infrastructure is often rigid, slow, and creates substantial frictional costs. It is estimated that the yearly cost of transactional friction alone is \$71 billion [8] in the financial sector, creating a strong desire for alternatives. This sector is a perfect match for DLT, as it enables systems that manage digital assets and financial transactions in more flexible, fast, and open federated infrastructures that eliminate the friction caused by individual private databases maintained by banks and financial services providers. Consequently, it is expected that a large part of the financial sector will move towards DLT [18].

At the core of DLT is the *replicated state* maintained by the network in the form of a ledger of transactions. In traditional blockchains, this ledger is fully replicated among all participants using consensus protocols [14,30,37,45,48]. For many practical use-cases, one can choose to use either permissionless consensus solutions that are operated via economic self-incentivization through cryptocurrencies (e.g., Nakamoto consensus [47, 57]), or permissioned consensus solutions that require vetted participation (e.g, PBFT, POE, and HOTSTUFF [16, 32, 59]). Unfortunately, the design of consensus protocols are severely limited in their ability to provide the *high transaction throughput* that is needed to address practical needs, e.g., in the financial sector. Indeed, on the one hand, we see that permissionless solutions can easily scale to thousands of participants, but are severely limited in their transaction processing throughput. E.g., in Ethereum, a popular public permissionless DLT platform, the rapid growth of decentralized finance applications [12] causes its network fees to rise precipitously as participants bid for limited network capacity [7], while Bitcoin can only process a few transactions per second [52]. On the other hand, permissioned solutions can reach much higher throughput. Permissioned blockchains are still fully-replicated resilient systems, however. Hence, the speed by which individual replicas can process transactions provides an upper-bound on the performance of these permissioned blockchains, ruling out scalability. Furthermore, adding replicas will actively decrease performance of a permissioned blockchain, as full replication among more replicas increases the cost of full replication (e.g., via consensus). As such, permissioned blockchains lack the scalability required by many modern data-based applications.

Recently, several general-purpose sharded consensus-based systems have been proposed to combat the limitations of fully-replicated consensus-based systems [1, 3, 4, 17, 34, 53]. In these systems, one partitions the data among several *shards*

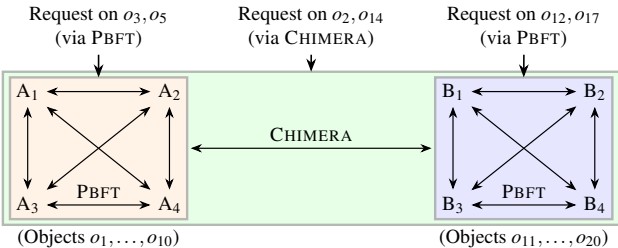

Figure 1: A *sharded* design in which two resilient blockchains each hold only a part of the data. Local decisions within a cluster are made via *traditional* PBFT *consensus*, whereas multi-shard transactions are processed via CHIMERA (proposed in this work).

that each can potentially operate mostly-independent on their data, while only requiring inter-shard coordination to process multi-shard transactions that affect data on several shards (see Figure 1).

The choice of protocol for such multi-shard transaction processing determines greatly the scalability benefits of sharding and the overhead costs incurred by sharding, however [1, 3, 4, 17, 34, 53]. In practice, existing proposals for sharding consensus-based systems have taken a general-purpose approach aiming at serving any workload. Unfortunately, such genericity comes at a cost, and existing proposals either have high coordination costs, incur high latencies, or have severe bottlenecks with multi-shard workloads.

In this work, we ask whether one can improve on the state-of-the-art proposals by *limiting* the type of workloads supported by the systems. In specific, we propose the following problem for further study:

> **Problem.** Can one *reduce* the cost of coordination in the design of sharded consensus-based systems by limiting the types of workloads supported?

In this paper, we give a preliminary *positive* answer for the above problem. In specific, we put forward the CHIMERA family of multi-shard transaction processing protocols that can process UTXO-transactions and uses properties of these transactions to *reduce* coordination to a minimum.

To be able to adapt to the needs of specific use-cases, we propose three variants of CHIMERA:

1. In Section 4, we propose Core-CHIMERA (CCHIMERA), a design specialized for processing *UTXO-like transactions*. CCHIMERA uses strict environmental assumptions on UTXO-transactions to its advantage to yield a *minimalistic* design that only requires a *single local consensus step* in affected shards, an absolute minimum. Furthermore, CCHIMERA requires only a single round of information sharing between shards. This information sharing can be implemented either via an all-to-all communication step (favoring latency over bandwidth usage)

or via an all-to-one-to-all communication step (favoring bandwidth usage over latency).

Even with this minimalistic design, CCHIMERA will operate *perfectly* with respect to all transactions proposed and approved by well-behaved clients (although it may fail to process transactions originating from malicious clients).

To also support more general-purpose environments in which clients are malicious or can legitimately approve conflicting transactions, we propose Optimistic-CHIMERA and Resilient-CHIMERA, *two* generalizations of CCHIMERA that each deal with the strict environmental assumptions of CCHIMERA while preserving the minimalistic design of CCHIMERA:

2. In Section 5, we propose Optimistic-CHIMERA. In the design of Optimistic-CHIMERA (OCHIMERA), we assume that malicious behavior is rare and we optimize the normal-case operations. We do so by keeping the normal-case operations as minimalistic as possible by utilizing a *single* multi-shard consensus step to execute multi-shard transactions in the normal case.

   This multi-shard consensus step combines the local consensus steps of CCHIMERA and the information sharing steps of CCHIMERA into a single step. As with CCHIMERA, this step can either favor latency or bandwidth. When compared to CCHIMERA, the multi-shard consensus step does not require any additional coordination phases in the well-behaved optimistic case, while still being able to lift the environmental assumptions of CCHIMERA and lowering the latency of transaction processing in most cases. In doing so, OCHIMERA does require intricate coordination when recovering from attacks, however.

3. In Section 6, we propose Resilient-CHIMERA. In the design of Resilient-CHIMERA, we assume that malicious behavior is common and we add sufficient coordination to the normal-case operations of CCHIMERA to enable a simpler and localized recovery path, allowing RCHIMERA to operate in a general-purpose fault-tolerant environments without significant costs to recover from attacks.

In Section 7, we show that all three variants of CHIMERA provide strong ordering guarantees based on their usage of UTXO-transactions. Finally, in Section 8, we compare the three CHIMERA protocols and show their potential scalability and performance benefits over state-of-the-art general-purpose systems

## 2 Preliminaries

As permissioned blockchains already have much higher throughputs than permissionless blockchains, we will focus on permissioned blockchains in this paper.

First, we introduce the system model, the sharding model, the data model, the transaction model, and the terminology and notation used throughout this paper.

If $S$ is a set of replicas, then $\mathcal{G}(S)$ denotes the non-faulty *good replicas* in $S$ that always operate as intended, and $\mathcal{F}(S) = S \setminus \mathcal{G}(S)$ denotes the remaining replicas in $S$ that are *faulty* and can act *Byzantine*, deviate from the intended operations, or even operate in coordinated malicious manners. We write $\mathbf{n}_S = |S|$, $\mathbf{g}_S = |\mathcal{G}(S)|$, and $\mathbf{f}_S = |S \setminus \mathcal{G}(S)| = \mathbf{n}_S - \mathbf{g}_S$ to denote the number of replicas in $S$, good replicas in $S$, and faulty replicas in $S$, respectively.

We assume that communication between replicas is *authenticated*: on receipt of a message $m$ from replica $\mathrm{R} \in \mathfrak{R}$, one can determine that $\mathrm{R}$ did sent $m$ if $\mathrm{R} \in \mathcal{G}(\mathfrak{R})$. Hence, faulty replicas are able to impersonate each other, but are not able to impersonate good replicas. To provide authenticated communication under practical assumptions, we can rely on cryptographic primitives such as digital signatures and threshold signatures [41, 54].

Let $\mathfrak{R}$ be a set of replicas. In a *sharded fault-tolerant system* over $\mathfrak{R}$, the replicas are partitioned into sets $\mathrm{shards}(\mathfrak{R}) = \{\mathcal{S}_0, \ldots, \mathcal{S}_{\mathbf{z}}\}$ such that the replicas in $\mathcal{S}_i$, $0 \leq i \leq \mathbf{z}$, operate as an independent Byzantine fault-tolerant system. As each $\mathcal{S}_i$ operates as an independent Byzantine fault-tolerant system, we require $\mathbf{n}_{\mathcal{S}_i} > 3\mathbf{f}_{\mathcal{S}_i}$, a minimal requirement to enable Byzantine fault-tolerance in an asynchronous environment [20, 21]. We assume that every shard $\mathcal{S} \in \mathrm{shards}(\mathfrak{R})$ has a unique identifier $\mathrm{id}(\mathcal{S})$.

**Assumption 2.1.** We assume *coordinating adversaries* that can, at will, choose and control any replica $\mathrm{R} \in \mathcal{S}$ in any shard $\mathcal{S} \in \mathrm{shards}(\mathfrak{R})$ in the sharded fault-tolerant system as long as, for each shard $\mathcal{S}' \in \mathrm{shards}(\mathfrak{R})$, the adversaries only control up to $\mathbf{f}_{\mathcal{S}'}$ replicas in $\mathcal{S}'$.

We use the *object-dataset model* in which data is modeled as a collection of *objects*. Each object $o$ has a unique *identifier* $\mathrm{id}(o)$ and a unique *owner* $\mathrm{owner}(o)$. In the following, we assume that all owners are *clients* of the system that manages these objects. The only operations that one can perform on an object are *construction* and *destruction*. An object cannot be recreated, as the attempted recreation of an object $o$ will result in a new object $o'$ with a distinct identifier ($\mathrm{id}(o) \neq \mathrm{id}(o')$).

Changes to object-dataset data are made via transactions requested by clients. We write $\langle \tau \rangle_c$ to denote a transaction $\tau$ requested by a client $c$. We assume that all transactions are *UTXO-like transactions*: a transaction $\tau$ first produces resources by destructing a set of *input objects* and then consumes these resources in the construction of a set of *output objects*. We do not rely on the exact rules regarding the production and consumption of resources, as they are highly application-specific. Given a transaction $\tau$, we write $\mathrm{Inputs}(\tau)$ and $\mathrm{Outputs}(\tau)$ to denote the input objects and output objects of $\tau$, respectively, and we write $\mathrm{Objects}(\tau) = \mathrm{Inputs}(\tau) \cup \mathrm{Outputs}(\tau)$.

**Assumption 2.2.** Given a transaction $\tau$, we assume that one can determine $\mathrm{Inputs}(\tau)$ and $\mathrm{Outputs}(\tau)$ a-priori. Furthermore, we assume that every transaction has inputs. Hence, $|\mathrm{Inputs}(\tau)| \geq 1$.

Owners of objects $o$ can *express their approval* for transactions $\tau$ that have $o$ as their input. To provide this functionality, we can rely on digital signatures [41].

**Assumption 2.3.** If an owner is well-behaved, then an expression of approval cannot be forged or provided by any other party.[1] Furthermore, a well-behaved owner of $o$ will only express its approval for *a single* transaction $\tau$ with $o \in \mathrm{Inputs}(\tau)$, as only one transaction can consume the object $o$, and the owner will only do so after the construction of $o$.

Let $o$ be an object. We assume that there is a well-defined function $\mathrm{shard}(o)$ that maps object $o$ to the single shard $\mathcal{S} \in \mathrm{shards}(\mathfrak{R})$ that is responsible for maintaining $o$. Given a transaction $\tau$, we write $\mathrm{shards}(\tau) = \{\mathrm{shard}(o) \mid o \in \mathrm{Objects}(\tau)\}$ to denote the shards that are affected by $\tau$. We say that $\tau$ is a *single-shard transaction* if $|\mathrm{shards}(\tau)| = 1$ and is a *multi-shard transaction* otherwise.

## 3 Multi-Shard Transaction Processing

Before we introduce CHIMERA, we put forward the correctness requirements we want to maintain in a multi-shard transaction system in which each shard is itself a set of replicas operated as a Byzantine fault-tolerant system. We say that a shard $\mathcal{S}$ performs an action if every good replica in $\mathcal{G}(\mathcal{S})$ performs that action. Hence, any processing decision or execution step performed by $\mathcal{S}$ requires the usage of a *consensus protocol* [14, 16, 30, 44, 45] that coordinates the operations of individual replicas in the system, e.g., a Byzantine fault-tolerant system driven by PBFT [16], POE [32], or HOTSTUFF [59], or a crash fault-tolerant system driven by PAXOS [44]. As these systems are fully-replicated, each replica holds exactly the same data, which is determined by the *sequence of transactions*—the journal—agreed upon via consensus:

**Definition 3.1.** A *consensus protocol* coordinate decision making among the replicas of a resilient cluster $\mathcal{S}$ by providing a reliable ordered replication of *decisions*. To do so, consensus protocols provide the following guarantees:

1. If good replica $\mathrm{R} \in \mathcal{S}$ makes a $\rho$-th decision, then all good replicas $\mathrm{R}' \in \mathcal{S}$ will make a $\rho$-th decision (whenever communication becomes reliable).

---

[1]Earlier, we assumed a unique owner that can approve transactions and prove object ownership in a unique and non-ambiguous way. This does not preclude shared ownership in which multiple participants own an object, however. In that case, we simply require that such a group of participants can approve transactions via their own agreement process to determine which transactions to support (e.g., via multiple signatures, via threshold signatures, or via other mechanisms).

2. If good replicas $R, Q \in \mathcal{S}$ make $\rho$-th decisions, then they make the same decisions.

3. Whenever a good replica learns that a decision $D$ needs to be made, then it can force consensus on $D$.[2]

Let $\tau$ be a transaction processed by a sharded fault-tolerant system. Processing of $\tau$ does not imply execution: the transaction could be invalid (e.g., the owners of affected objects did not express their approval) or the transaction could have inputs that no longer exists. We say that the system *commits* to $\tau$ if it decides to apply the modifications prescribed by $\tau$, and we say that the system *aborts* $\tau$ if it decides to not do so. Using this terminology, we put forward the following requirements for any sharded fault-tolerant system:

R1 *Validity*. The system must only processes transaction $\tau$ if, for every input object $o \in \mathtt{Inputs}(\tau)$ with a well-behaved owner $\mathtt{owner}(o)$, the owner $\mathtt{owner}(o)$ approves the transaction.[3]

R2 *Shard-involvement*. The shard $\mathcal{S}$ only processes transaction $\tau$ if $\mathcal{S} \in \mathtt{shards}(\tau)$.

R3 *Shard-applicability*. Let $D(\mathcal{S})$ be the dataset maintained by shard $\mathcal{S}$ at time $t$. The shards $\mathtt{shards}(\tau)$ only commit to execution of transaction $\tau$ at $t$ if $\tau$ consumes only existing objects. Hence, $\mathtt{Inputs}(\tau) \subseteq \bigcup \{ D(\mathcal{S}) \mid \mathcal{S} \in \mathtt{shards}(\tau) \}$.

R4 *Cross-shard-consistency*. If shard $\mathcal{S}$ commits (aborts) transaction $\tau$, then all shards $\mathcal{S}' \in \mathtt{shards}(\tau)$ eventually commit (abort) $\tau$.

R5 *Service*. If client $c$ is well-behaved and wants to request a valid transaction $\tau$, then the sharded system will eventually *process* $\langle \tau \rangle_c$. If $\tau$ is shard-applicable, then the sharded system will eventually *execute* $\langle \tau \rangle_c$.

R6 *Confirmation*. If the system processes $\langle \tau \rangle_c$ and $c$ is well-behaved, then $c$ will eventually learn whether $\tau$ is committed or aborted.

The *validity* of transactions is a *local requirement*: whether a transaction $\tau$ is valid can be determined by checking whether all owners of inputs of $\tau$ support that transaction. Typically, ownership is expressed via digital signatures, which can be

---

[2]Many definitions of consensus include a requirement of *non-triviality* instead. Here, we focus on the usage of consensus for operating services that processes requests of clients (e.g., the purpose for which PBFT was designed). In such services, non-triviality is typically provided by assuring that any client can get their requests processed. To do so, clients can send their request $D$ to all good replicas, whom then can collaborate together to force a consensus on $D$. The specific details of such process depend on the details of the consensus protocol.

[3]Determining validity of a transaction can include application-level requirements that should hold in a transaction. If, for example, the objects represent monetary balances, then transactions that produce *more output* than they consume *input* can be considered invalid.

verified deterministically by any replica in any shard independently. Hence, all replicas in all affected shards will make the same conclusion on whether $\tau$ is valid. Likewise, also shard-involvement is a *local requirement*, as individual shards can determine whether they need to process a given transaction. In the same sense, shard-applicability and cross-shard-consistency are *global* requirements, as assuring these requirements requires coordination between the shards affected by a transaction.

In the above and throughout this paper, we will speak of transaction *processing* whenever we look at the steps the system takes after receiving a request (eventually leading to discarding the request when it is invalid, a commit decision, or an abort decision). We will speak of transaction *execution* to refer to transactions that finished processing with either a commit decision or an abort decision.

## 4 Core-CHIMERA: Simple Yet Efficient Transaction Processing

The core idea of CHIMERA is to minimize the coordination necessary for multi-shard ordering and execution of transactions. To do so, CHIMERA combines the semantics of transactions in the object-dataset model with the minimal coordination required to assure shard-applicability and cross-shard consistency. This combination results in the following high-level three-step approach towards processing any transaction $\tau$:

1. *Local inputs*. First, every affected shard $\mathcal{S} \in \mathtt{shards}(\tau)$ locally determines whether it has all inputs from $\mathcal{S}$ that are necessary to process $\tau$.

2. *Cross-shard exchange*. Then, every affected shard $\mathcal{S}$ exchanges these inputs to all other shards in $\mathtt{shards}(\tau)$, thereby pledging to use their local inputs in the execution of $\tau$.

3. *Decide outcome*. Finally, every affected shard $\mathcal{S}$ decides to commit $\tau$ if all affected shards were able to provide all local inputs necessary for execution of $\tau$.

Next, we describe how these three high-level steps are incorporated by CHIMERA into normal consensus steps at each shards. Let shard $\mathcal{S} \in \mathtt{shards}(\mathfrak{R})$ receive client request $\langle \tau \rangle_c$. The good replicas in $\mathcal{S}$ will first determine whether $\tau$ is valid and applicable.

If $\tau$ is not valid or $\mathcal{S} \notin \mathtt{shards}(\tau)$, then the good replicas discard $\tau$. Otherwise, if $\tau$ is valid and $\mathcal{S} \in \mathtt{shards}(\tau)$, then the good replicas utilize *consensus* to force the primary $\mathcal{P}(\mathcal{S})$ to propose in some consensus round $\rho$ the message $m(\mathcal{S}, \tau)_\rho = (\langle \tau \rangle_c, I(\mathcal{S}, \tau), D(\mathcal{S}, \tau))$, in which $I(\mathcal{S}, \tau) = \{ o \in \mathtt{Inputs}(\tau) \mid \mathcal{S} = \mathtt{shard}(o) \}$ is the set of objects maintained by $\mathcal{S}$ that

are input to $\tau$ and $D(\mathcal{S},\tau) \subseteq I(\mathcal{S},\tau)$ is the set of currently-available inputs at $\mathcal{S}$. Only if $I(\mathcal{S},\tau) = D(\mathcal{S},\tau)$ will $\mathcal{S}$ pledge to use the local inputs $I(\mathcal{S},\tau)$ in the execution of $\tau$.

We use *consensus* during the *local inputs* step as it provides an ordered agreement among seqeuences of transactions. This ordered agreement is necessary to acquire a consistent results among all replicas in a shard: all replicas of a shard need to process all transactions they process in the same order, as otherwise they cannot agree on which of the inputs of a transaction $\tau$ are available to $\tau$ in the presence of other transactions with the same inputs.

The acceptance of $m(\mathcal{S},\tau)_\rho$ in round $\rho$ by all good replicas completes the *local inputs* step. Next, during processing of $\tau$, the *cross-shard exchange* and *decide outcome* steps are performed. First, the *cross-shard exchange* step. In this step, $\mathcal{S}$ broadcasts $m(\mathcal{S},\tau)_\rho$ to all other shards in $\mathtt{shards}(\tau)$. To assure that the broadcast arrives, we rely on a reliable primitive for *cross-shard exchange* that guarantees that only approved-upon values can be exchanged. Recently, such primitives have been formalized as cluster-sending [33, 35]:

**Definition 4.1.** Let $\mathcal{S}_1, \mathcal{S}_2$ be two shards. The *cluster-sending problem* is the problem of sending a value $v$ from $\mathcal{S}_1$ to $\mathcal{S}_2$ such that:

1. all good replicas in $\mathcal{S}_2$ *receive* the value $v$;

2. all good replicas in $\mathcal{S}_1$ receive *confirmation* that the value $v$ was received by all good replicas in $\mathcal{S}_2$; and

3. good replicas in $\mathcal{S}_2$ can only receive a value $v$ if all good replicas in $\mathcal{S}_1$ *agreed* upon sending $v$.

After $\mathcal{S}$ broadcasts $m(\mathcal{S},\tau)_\rho$ to all other shards in $\mathtt{shards}(\tau)$, the replicas in $\mathcal{S}$ wait until they receive messages $m(\mathcal{S}',\tau)_{\rho'} = (\langle\tau\rangle_c, I(\mathcal{S}',\tau), D(\mathcal{S}',\tau))$ from all other shards $\mathcal{S}' \in \mathtt{shards}(\tau)$.

After cross-shard exchange comes the final *decide outcome* step. After $\mathcal{S}$ receives $m(\mathcal{S}',\tau)_{\rho'}$ from all shards $\mathcal{S}' \in \mathtt{shards}(\tau)$, it decides to *commit* whenever $I(\mathcal{S}',\tau) = D(\mathcal{S}',\tau)$ for all $\mathcal{S}' \in \mathtt{shards}(\tau)$. Otherwise, it decides *abort*. If $\mathcal{S}$ decides commit, then all good replicas in $\mathcal{S}$ destruct all objects in $D(\mathcal{S},\tau)$ and construct all objects $o \in \mathtt{Outputs}(\tau)$ with $\mathcal{S} = \mathtt{shard}(o)$. Finally, each good replica informs $c$ of the outcome of execution. If $c$ receives, from every shard $\mathcal{S}'' \in \mathtt{shards}(\tau)$, identical outcomes from $\mathbf{g}_{\mathcal{S}''} - \mathbf{f}_{\mathcal{S}''}$ distinct replicas in $\mathcal{S}''$, then it considers $\tau$ to be successfully executed. In Figure 2, we sketched the working of CCHIMERA.

The *cross-shard exchange* step of CCHIMERA at $\mathcal{S}$ involves waiting for other shards $\mathcal{S}'$. Hence, there is the danger of deadlocks if the other shards $\mathcal{S}'$ never perform their cross-shard exchange steps. To assure that such situations do not lead to a deadlock, we employ two techniques.

1. *Internal propagation.* To deal with situations in which some shards $\mathcal{S} \in \mathtt{shards}(\tau)$ did not receive $\langle\tau\rangle_c$ (e.g.,

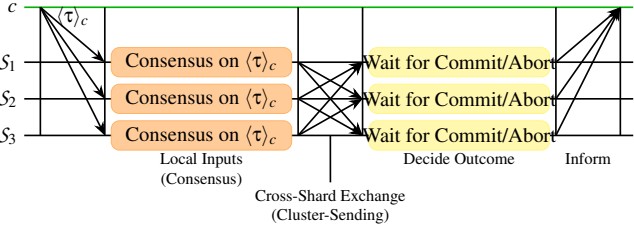

Figure 2: The message flow of CCHIMERA for a 3-shard client request $\langle\tau\rangle_c$ that is committed.

due to network failure or due to a faulty client that fails to send $\langle\tau\rangle_c$ to $\mathcal{S}$), we allow each shard to learn $\tau$ from any other shard. In specific, $\mathcal{S}$ will start consensus on $\langle\tau\rangle_c$ after receiving *cross-shard exchange* related to $\langle\tau\rangle_c$.

2. *Concurrent resolution.* To deal with concurrent transactions that content for the same objects, we allow each shard to accept and process transactions for different rounds concurrently. To assure that concurrent resolution does not lead to inconsistent state updates, each replica implements the following *first-pledge* and *ordered-commit* rules. Let $\tau$ be a transaction with $\mathcal{S} \in \mathtt{shards}(\tau)$ and $\mathtt{R} \in \mathcal{S}$. The *first-pledge* rule states that $\mathcal{S}$ pledges $o$, constructed in round $\rho$, to transaction $\tau$ only if $\tau$ is the first transaction proposed after round $\rho$ with $o \in \mathtt{Inputs}(\tau)$. The *ordered-commit* rule states that $\mathcal{S}$ can abort $\tau$ in any order, but will only commit $\tau$ that is accepted in round $\rho$ after previous rounds finished execution.

The above first-pledge and ordered-commit rules do not need to be enforced or guaranteed, as they specify deterministic behavior for all good replicas. Next, we illustrate the usage of these rules.

*Example* 4.2. Consider two shards $\mathcal{S}_1$ and $\mathcal{S}_2$ affected by transactions $\tau_1$ and $\tau_2$ that each require objects $o_1$ and $o_2$ residing on shards $\mathcal{S}_1$ and $\mathcal{S}_2$, respectively. Now consider the case in which shard $\mathcal{S}_1$ first processes $\tau_1$ and then $\tau_2$, while shard $\mathcal{S}_2$ first processes $\tau_2$ and then $\tau_1$. In this case, shard $\mathcal{S}_1$ will pledge $o_1$ to $\tau_1$ and shard $\mathcal{S}_2$ will pledge $o_2$ to $\tau_2$. Hence, both $\tau_1$ and $\tau_2$ miss inputs and will fail to complete execution. As both transactions will abort, the order in which they abort does not matter.

In this situation, which will only happen if two transactions have the same inputs in violation of Assumption 2.3, will result in an abort for the two transactions $\tau_1$ and $\tau_2$. Transactions that have unique inputs (in line with Assumption 2.3), will always be able to be committed.

Abort decisions at shard $\mathcal{S}$ on a transaction $\tau$ can often be made without waiting for all shards $\mathcal{S}' \in \mathtt{shards}(\tau)$: shard $\mathcal{S}$ can decide abort after it detects $I(\mathcal{S},\tau) \neq D(\mathcal{S},\tau)$ or after it receives the first message $(\langle\tau\rangle_c, I(\mathcal{S}'',\tau), D(\mathcal{S}'',\tau))$ with

$I(\mathcal{S}'',\tau) \neq D(\mathcal{S}'',\tau)$, $\mathcal{S}'' \in \text{shards}(\tau)$. For efficiency, we allow $\mathcal{S}$ to abort in these cases.

**Theorem 4.3.** *If, for all shards $\mathcal{S}^*$, $\mathbf{g}_{\mathcal{S}^*} > 2\mathbf{f}_{\mathcal{S}^*}$, and Assumptions 2.1, 2.2, and 2.3 hold, then Core-*CHIMERA *satisfies Requirements R1–R6 with respect to all transactions that are not requested by malicious clients and do not involve objects with malicious owners.*

*Proof.* Let $\tau$ be a transaction. As good replicas in $\mathcal{S}$ discard $\tau$ if it is invalid or if $\mathcal{S} \notin \text{shards}(\tau)$, CCHIMERA provides *validity* and *shard-involvement*. Next, *shard-applicability* follow directly from the decide outcome step.

If a shard $\mathcal{S}$ commits or aborts transaction $\tau$, then it must have completed the decide outcome and cross-shard exchange steps. Hence, all shards $\mathcal{S}' \in \text{shards}(\tau)$ must have exchanged the necessary information to $\mathcal{S}$. By relying on cluster-sending for cross-shard exchange, $\mathcal{S}'$ requires cooperation of all good replicas in $\mathcal{S}'$ to exchange the necessary information to $\mathcal{S}$. Hence, we have the guarantee that these good replicas will also perform cross-shard exchange to any other shard $\mathcal{S}'' \in \text{shards}(\tau)$. As such, every shard $\mathcal{S}'' \in \text{shards}(\tau)$ will receive the same information as $\mathcal{S}$, complete cross-shard exchange, and make the same decision during the decide outcome step, providing *cross-shard consistency*.

Due to internal propagation and concurrent resolution, every valid transaction $\tau$ will be processed by CCHIMERA as soon as it is send to any shard $\mathcal{S} \in \text{shards}(\tau)$. Hence, every shard in $\text{shards}(\tau)$ will perform the necessary steps to eventually inform the client. As all good replicas $R \in \mathcal{S}$, $\mathcal{S} \in \text{shards}(\tau)$, will inform the client of the outcome for $\tau$, the majority of these inform-messages come from good replicas, enabling the client to reliably derive the true outcome. Hence, CCHIMERA provides *service* and *confirmation*. $\square$

Notice that in the object-dataset model in which we operate, each object can be constructed once and destructed once. Hence, each object $o$ can be part of at-most two committed transactions: the first of which will construct $o$ as an output, and the second of which has $o$ as an input and will consume and destruct $o$. This is independent of any other operations on other objects. As such these two transactions *cannot* happen concurrently. Consequently, we only have concurrent transactions on $o$ if the owner $\text{owner}(o)$ expresses approval for several transactions that have $o$ as an input. By Assumption 2.3, the owner $\text{owner}(o)$ must be malicious in that case. As such, transactions of well-behaved clients and owners will *never abort*.

In the design of CCHIMERA, we take *full* advantage of the above observation: CCHIMERA effectively *eliminates all coordination* when deciding to process a multi-shard transaction due to which all involved shards can process a transaction *independently* with a single consensus step: all communication between shards in CCHIMERA is dedicated to exchange execution state *after* individual shards reach consensus. We can

do so as any *aborts*, which could have been prevented with additional coordination, are always due to malicious behavior by clients and owners of objects. Due to this, CCHIMERA will not undo any pledges of objects to the execution of any transactions. This implies that objects that are involved in malicious transactions can get lost for future usage, while not affecting any transactions of well-behaved clients.

Finally, we remark that CCHIMERA depends on underlying consensus and cluster-sending protocols. The level to which CCHIMERA can deal with asynchronous behavior depends on the particular choices of these protocols.

# 5 Optimistic-CHIMERA: Robust Transaction Processing

In the previous section, we introduced CCHIMERA, a minimalistic multi-shard transaction processing protocol that relies on properties of UTXO-like transactions to maximize performance. Although the design of CCHIMERA is simple yet effective, we see two shortcomings that limits its use. First, CCHIMERA operates under Assumption 2.3, the assumption that any issues arising from concurrent transactions is due to malicious behavior of clients. As such, CCHIMERA chooses to lock out objects affected by such malicious behavior for any future usage. Second, CCHIMERA requires consecutive consensus and cluster-sending steps, which increases its transaction processing latencies. Next, we investigate how to deal with these weaknesses of CCHIMERA *without giving up* on the minimalistic nature of CCHIMERA.

To do so, we propose Optimistic-CHIMERA (OCHIMERA), which is optimized for the *optimistic* case in which we have no concurrent transactions, while providing a recovery path that can recover from concurrent transactions without locking out objects (and without requiring Assumption 2.3). At the core of OCHIMERA is assuring that any issues due to malicious behavior, e.g., concurrent transactions, are *detected* in such a way that individual replicas can recover. At the same time, we want to minimize transaction processing latencies. To bridge between these two objectives, we integrate detection and cross-shard coordination within a single consensus round that runs at each affected shard.

OCHIMERA does not rely on underlying consensus and cluster-sending protocols. For the design of OCHIMERA, we assume *asynchronous communication*: messages can get lost, arrive with arbitrary delays, and in arbitrary order. Consequently, it is impossible to distinguish between, on the one hand, a replica that is malicious and does not send out messages, and, on the other hand, a replica that does send out proposals that get lost in the network. It is well-known that in such an environment, consensus cannot be provided [25, 27]. As such, OCHIMERA is designed to operate in a asynchronous environment in which it will *never cause* data inconsistency and only guarantees *progress* (service and confirmation) even-

tually when communication is reliable for a sufficiently-long period of time. This is the same model of partial asynchronous communication as used by PBFT.

Let $\langle\tau\rangle_c$ be a multi-shard transaction, let $\mathcal{S} \in \text{shards}(\tau)$ be an affected shard with primary $\mathcal{P}(\mathcal{S})$, and let $m(\mathcal{S},\tau)_{v,\rho} = (\langle\tau\rangle_c, I(\mathcal{S},\tau), D(\mathcal{S},\tau))$ be the round-$\rho$ proposal of $\mathcal{P}(\mathcal{S})$ of view $v$ of $\mathcal{S}$. To enable detection of concurrent transactions, OCHIMERA modifies the consensus-steps of the underlying consensus protocol by applying the following high-level idea:

> A replica $R \in \mathcal{S}$, $\mathcal{S} \in \text{shards}(\tau)$, only accepts proposal $m(\mathcal{S},\tau)_{v,\rho}$ for transaction $\tau$ if it gets confirmation that replicas in each other shard $\mathcal{S}' \in \text{shards}(\tau)$ are also accepting proposals for $\tau$. Otherwise, replica $R$ detects failure.

To simplify presentation, we will use a traditional design that uses all-to-all communication between all replicas in all affected shards akin to the design of PBFT [16]. To minimize inter-shard communication (at the cost of latency) one can also utilize threshold signatures to implement all-to-one-to-all communication akin to the design of HOTSTUFF [59] to carry over local prepare and commit certificates between shards via a few constant-sized messages.

Next, we illustrate how to integrate the above idea in the three-phase design of PBFT, thereby turning PBFT into a multi-shard aware consensus protocol:

1. *Global preprepare.* Primary $\mathcal{P}(\mathcal{S})$ must send $m(\mathcal{S},\tau)_{v,\rho}$ to all replicas $R' \in \mathcal{S}'$, $\mathcal{S}' \in \text{shards}(\tau)$. Replica $R \in \mathcal{S}$ only finishes the global preprepare phase after it receives a *global preprepare certificate* consisting of a set $M = \{m(\mathcal{S}'',\tau)_{v'',\rho''} \mid \mathcal{S}'' \in \text{shards}(\tau)\}$ of preprepare messages from all primaries of shards affected by $\tau$.

2. *Global prepare.* After $R \in \mathcal{S}$, $\mathcal{S} \in \text{shards}(\tau)$, finishes the global preprepare phase, it sends prepare messages for $M$ to all other replicas in $R' \in \mathcal{S}'$, $\mathcal{S}' \in \text{shards}(\tau)$. Replica $R \in \mathcal{S}$ only finishes the global prepare phase for $M$ after, for every shard $\mathcal{S}' \in \text{shards}(\tau)$, it receives a *local prepare certificate* consisting of a set $P(\mathcal{S}')$ of prepare messages for $M$ from $\mathbf{g}_{\mathcal{S}'}$ distinct replicas in $\mathcal{S}'$. We call the set $\{P(\mathcal{S}'') \mid \mathcal{S}'' \in \text{shards}(\tau)\}$ a *global prepare certificate.*

3. *Global commit.* After replica $R \in \mathcal{S}$, $\mathcal{S} \in \text{shards}(\tau)$, finishes the global prepare phase, it sends commit messages for $M$ to all other replicas in $R' \in \mathcal{S}'$, $\mathcal{S}' \in \text{shards}(\tau)$. Replica $R \in \mathcal{S}$ only finishes the global commit phase for $M$ after, for every shard $\mathcal{S}' \in \text{shards}(\tau)$, it receives a *local commit certificate* consisting of a set $C(\mathcal{S}')$ of commit messages for $M$ from $\mathbf{g}_{\mathcal{S}'}$ distinct replicas in $\mathcal{S}'$. We call the set $\{P(\mathcal{S}'') \mid \mathcal{S}'' \in \text{shards}(\tau)\}$ a *global commit certificate.*

The above three-phase *global-*PBFT protocol already takes care of the *local input* and *cross-shard exchange* steps. Indeed,

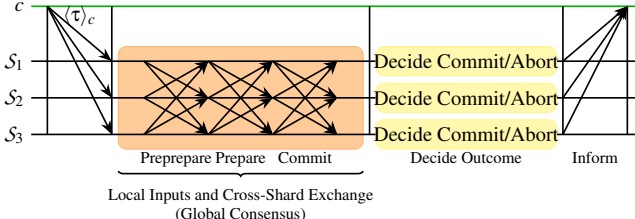

Figure 3: The message flow of OCHIMERA for a 3-shard client request $\langle\tau\rangle_c$ that is committed.

a replica $R \in \mathcal{S}$ that finishes the global commit phase has accepted global preprepare certificate $M$, which contains all information of other shards to proceed with processing. At the same time, $R$ also has confirmation that $M$ is prepared by a majority of all good replicas in each shard $\mathcal{S}' \in \text{shards}(\tau)$ (which will eventually be followed by execution of $\tau$ within $\mathcal{S}'$). With these ingredients in place, only the *decide outcome* step remains.

The decide outcome step at shard $\mathcal{S}$ is entirely determined by the global preprepare certificate $M$. Shard $\mathcal{S}$ decides to *commit* whenever $I(\mathcal{S}',\tau) = D(\mathcal{S}',\tau)$ for all $(\langle\tau\rangle_c, I(\mathcal{S}',\tau), D(\mathcal{S}',\tau)) \in M$. Otherwise, it decides *abort*. If $\mathcal{S}$ decides commit, then all good replicas in $\mathcal{S}$ destruct all objects in $D(\mathcal{S},\tau)$ and construct all objects $o \in \text{Outputs}(\tau)$ with $\mathcal{S} = \text{shard}(o)$. Finally, each good replica informs $c$ of the outcome of execution. If $c$ receives, from every shard $\mathcal{S}' \in \text{shards}(\tau)$, identical outcomes from $\mathbf{g}_{\mathcal{S}'} - \mathbf{f}_{\mathcal{S}'}$ distinct replicas in $\mathcal{S}'$, then it considers $\tau$ to be successfully executed. In Figure 3, we sketched the working of OCHIMERA.

We note that OCHIMERA is not the only multi-shard aware consensus protocol recently proposed (e.g., [3, 4]). What sets OCHIMERA apart is how it guarantees correctness *in all environments*, which is determined by how OCHIMERA deals with *non-optimistic cases* in which failure is detected and recovery is necessary. We will detail recovery next. As a first step, we illustrate the ways in which the normal-case of OCHIMERA can fail (e.g., due to malicious behavior of clients, failing replicas, or unreliable communication).

*Example* 5.1. Consider a transaction $\tau$ proposed by client $c$ and affecting shard $\mathcal{S} \in \text{shards}(\tau)$. First, we consider the case in which $\mathcal{P}(\mathcal{S})$ is malicious and tries to set up a coordinated attack. To have maximum control over the steps of OCHIMERA, the primary sends the message $m(\mathcal{S},\tau)_{v,\rho}$ to only $\mathbf{g}_{\mathcal{S}''} - \mathbf{f}_{\mathcal{S}''}$ good replicas in each shard $\mathcal{S}'' \in \text{shards}(\tau)$. By doing so, $\mathcal{P}(\mathcal{S})$ can coordinate the faulty replicas in each shard to assure failure of any phase at any replica $R' \in \mathcal{S}'$, $\mathcal{S}' \in \tau$:

1. To prevent $R'$ from finishing the global preprepare phase (and start the global prepare phase) for an $M$ with $m(\mathcal{S}',\tau)_{v',\rho'} \in M$, $\mathcal{P}(\mathcal{S})$ simply does not send $m(\mathcal{S},\tau)_{v,\rho}$ to $R'$.

2. To prevent $R'$ from finishing the global prepare phase

(and start the global commit phase) for $M$, $\mathcal{P}(\mathcal{S})$ instructs the faulty replicas in $\mathcal{F}(\mathcal{S})$ to not send prepare messages for $M$ to $\text{R}'$. Hence, $\text{R}'$ will receive at-most $\mathbf{g}_{\mathcal{S}} - \mathbf{f}_{\mathcal{S}}$ prepare messages for $M$ from replicas in $\mathcal{S}$, assuring that it will not receive a local prepare certificate $P(\mathcal{S})$ and will not finish the global prepare phase for $M$.

3. Likewise, to prevent $\text{R}'$ from finishing the global commit phase (and start execution) for $M$, $\mathcal{P}(\mathcal{S})$ instructs the faulty replicas in $\mathcal{F}(\mathcal{S})$ to not send commit messages to $\text{R}'$. Hence, $\text{R}'$ will receive at-most $\mathbf{g}_{\mathcal{S}} - \mathbf{f}_{\mathcal{S}}$ commit messages for $M$ from replicas in $\mathcal{S}$, assuring that it will not receive a local commit certificate $C(\mathcal{S})$ and will not finish the global commit phase for $M$.

None of the above attacks can be attributed to faulty behavior of $\mathcal{P}(\mathcal{S})$ as unreliable communication can result in the same outcomes for $\text{R}'$. Furthermore, even if communication is reliable and $\mathcal{P}(\mathcal{S})$ is good, replica $\text{R}'$ can see the same outcomes due to malicious behavior of the client or of primaries of other shards in $\text{shards}(\tau)$:

1. The client $c$ can be malicious and not send $\tau$ to $\mathcal{S}$. At the same time, all other primaries $\mathcal{P}(\mathcal{S}'')$ of shards $\mathcal{S}'' \in \text{shards}(\tau)$ can be malicious and not send anything to $\mathcal{S}$ either. In this case, $\mathcal{P}(\mathcal{S})$ will never be able to send any message $m(\mathcal{S}, \tau)_{v,\rho}$ to $\text{R}'$, as no replica in $\mathcal{S}$ is aware of $\tau$.

2. If any primary $\mathcal{P}(\mathcal{S}'')$ of $\mathcal{S}'' \in \text{shards}(\tau)$ is malicious, then it can prevent some replicas in $\mathcal{S}$ from starting the global prepare phase, thereby preventing these replicas to send prepare messages to $\text{R}'$. If $\mathcal{P}(\mathcal{S}'')$ prevents sufficient replicas in $\mathcal{S}$ from starting the global prepare phase, $\text{R}'$ will be unable to finish the global prepare phase.

3. Likewise, any malicious primary $\mathcal{P}(\mathcal{S}'')$ of $\mathcal{S}'' \in \text{shards}(\tau)$ can prevent replicas in $\mathcal{S}$ from starting the global commit phase, thereby assuring that $\text{R}'$ will be unable to finish the global commit phase.

To deal with malicious behavior, OCHIMERA needs a robust recovery mechanism. Indeed, the main difference of the multi-shard consensus of OCHIMERA and the single-shard consensus PBFT is that OCHIMERA will use a single primary *per* shard whereas PBFT only has a single primary. This difference affects the capability of individual replicas to detect the root cause of disruptions of the normal-case operations (as several primaries could be the root cause of such disruptions). As such, we cannot simply build the robust recovery mechanism on top of traditional view-change approaches: these traditional view-change approaches require that one can identify a single source of failure (when communication is reliable), namely the current primary. To remedy this, the recovery mechanisms of OCHIMERA has components that perform *local view-change* and that perform *global state recovery*.

```
1:  event R ∈ S is unable to finish round ρ of view v do
2:      if R finished in round ρ the global prepare phase for M,
             but is unable to finish the global commit phase then
3:          Let P be the global prepare certificate of R for M.
4:          if R has a local commit certificate C(S″) for M then
5:              for S′ ∈ shards(τ) do
6:                  if R did not yet receive a local commit certificate C(S′) then
7:                      Broadcast ⟨VCGlobalSCR : M, P, C(S″)⟩ to all replicas in S′.
8:          else Detect the need for local state recovery of round ρ of view v (Figure 5).
9:      else Detect the need for local state recovery of round ρ of view v (Figure 5).
10:     (Eventually repeat this event if R remains unable to finish round ρ.)

11: event R′ ∈ S′ receives message ⟨VCGlobalSCR : M, P, C(S″)⟩ from R ∈ S do
12:     if R′ did not reach the global commit phase for M then
13:         Use M, P, and C(S″) to reach the global commit phase for M.
14:     else Send a commit message for M to R.
```

Figure 4: The view-change *global short-cut recovery path* that determines whether $\text{R}$ already has the assurance that the current transaction will be committed. If this is the case, then $\text{R}$ requests only the missing information to proceed with execution. Otherwise, $\text{R}$ requires at-least local recovery (Figure 5).

Next, we will detail the working of the recovery mechanisms of OCHIMERA. To simplify presentation, we will focus on the recovery of a single transaction. The techniques presented are straightforward to generalize to any history of zero-or-more transactions. The pseudo-code for the recovery protocol can be found in Figure 4. Next, we describe the working of this recovery protocol in detail.

Let $\text{R} \in \mathcal{S}$ be a replica that determines that it cannot finish a round $\rho$ of view $v$. First, $\text{R}$ determines whether it already has a *guarantee* on which transaction it has to process in round $\rho$. This is the case when the following conditions are met: $\text{R}$ finished the global prepare phase for $M$ with $m(\mathcal{S}, \tau)_{v,\rho} \in M$ and has received a local commit certificate $C(\mathcal{S}'')$ for $M$ from some shard $\mathcal{S}'' \in \text{shards}(\tau)$. In this case, $\text{R}$ can simply request all missing local commit certificates directly, as $C(\mathcal{S}'')$ can be used to prove to any involved replica $\text{R}' \in \mathcal{S}'$, $\mathcal{S}' \in \text{shards}(\tau)$, that $\text{R}'$ also needs to commit to $M$. To request such missing commit certificates of $\mathcal{S}'$, replica $\text{R}$ sends out VCGlobalSCR messages to all replicas in $\mathcal{S}'$ (Line 7 of Figure 4). Any replica $\text{R}'$ that receives such a VCGlobalSCR message can use the information in that message to reach the global commit phase for $M$ and, hence, provide $\text{R}$ with the requested commit messages (Line 11 of Figure 4).

If $\text{R}$ does not have a *guarantee* itself on which transaction it has to process in round $\rho$, then it needs to determine whether any other replica (either in its own shard or in any other shard) has already received and acted upon such a guarantee. To initiate such local and global state recovery, $\text{R}$ simply detects the current view as faulty. To do so, $\text{R}$ broadcasts a VCRecoveryRQ message to all other replicas in $\mathcal{S}$ that contains all information $\text{R}$ collected on round $\rho$ in view $v$ (Line 4 of Figure 5). Other replicas $\text{Q} \in \mathcal{S}$ that already have *guarantees* for round $\rho$ can help $\text{R}$ by providing all missing information (Line 6 of Figure 5). On receipt of this informa-

tion, R can proceed with the round (Line 7 of Figure 5). If no replicas can provide the missing information, then eventually all good replicas will detect the need for local recovery, this either by themselves (Line 1 of Figure 5) or after receiving VCRecoveryRQ messages of at-least $\mathbf{f}_S + 1$ distinct replicas in $S$, of which at-least a single replica must be good (Line 10 of Figure 5).

Finally, if a replica R receives $\mathbf{g}_S$ VCRecoveryRQ messages, then it has the guarantee that at least $\mathbf{g}_S - \mathbf{f}_S \geq \mathbf{f}_S + 1$ of these messages come from good replicas in $S$. Hence, due to Line 10 of Figure 5, all $\mathbf{g}_S$ good replicas in $S$ will send VCRecoveryRQ, and, when communication is reliable, also receive these messages. Consequently, at this point, R can start the new view by electing a new primary and awaiting the NewView proposal of this new primary (Line 12 of Figure 5). If R is the new primary, then it starts the new view by proposing a NewView. As other shards *could* have already made final decisions depending on local prepare or commit certificates of $S$ for round $\rho$, we need to assure that such certificates are not invalidated. To figure out whether such final decisions have been made, the new primary will query other shards $S'$ for their state whenever the NewView message contains global preprepare certificates for transactions $\tau$, $S' \in \text{shards}(\tau)$, but not a local commit certificate to *guarantee* execution of $\tau$ (Line 17 of Figure 5).

The new-view process has three stages. First, the new primary P proposes the new-view via a NewView message (Line 12 of Figure 5). If necessary, the new primary P also requests the relevant global state from any relevant shard (Line 1 of Figure 6). The replicas in other shards will respond to this request with their local state (Line 9 of Figure 6). The new primary collects these responses and sends them to all replicas in $S$ via a NewViewGlobal message. Then, after P sends the NewView message to R ∈ $S$, R determines whether the NewView message contains sufficient information to recover round $\rho$ (Line 15 of Figure 6), contains sufficient information to wait for any relevant global state (Line 17 of Figure 6), or to determine that the new primary must propose for round $\rho$ (Line 19 of Figure 6). If R determines it needs to wait for any relevant global state, then R will wait for this state to arrive via a NewViewGlobal message. Based on the received global state, R determines to recover round $\rho$ (Line 21 of Figure 6), or determines that the new primary must propose for round $\rho$ (Line 24 of Figure 6).

Next, we will prove the correctness of the view-change of OCHIMERA. First, using a standard quorum argument, we prove that in a single round of a single view of $S$, only a single global preprepare message affecting $S$ can get committed by any other affected shards:

**Lemma 5.1.** *Let $\tau_1$ and $\tau_2$ be transactions with $S \in (\text{shards}(\tau_1) \cap \text{shards}(\tau_2))$. If $\mathbf{g}_S > 2\mathbf{f}_S$ and there exists shards $S_i \in \text{shards}(\tau_i)$, $i \in \{1,2\}$, such that good replicas $R_i \in \mathcal{G}(S_i)$ reached the global commit phase for global preprepare certificate $M_i$ with $m(S,\tau_i)_{v,\rho} \in M_i$, then $\tau_1 = \tau_2$.*

```
1:  event R ∈ S detects the need for local state recovery of round ρ of view v do
2:      Let M be any latest global preprepare certificate accepted for round ρ by R.
3:      Let S be M and any prepare and commit certificates for M collected by R.
4:      Broadcast ⟨VCRecoveryRQ : v,ρ,S⟩.

5:  event Q ∈ S receives messages ⟨VCRecoveryRQ : v,ρ,S⟩ of R ∈ S and Q has

        1. started the global prepare phase for M with m(S,τ)_{w,ρ} ∈ M;
        2. a global prepare certificate for M;
        3. a local commit certificate C(S″) for M
    do
6:      Send ⟨VCLocalSCR : M,P,C(S″)⟩ to R ∈ S.

7:  event R ∈ S receives message ⟨VCLocalSCR : M,P,C(S″)⟩ from Q ∈ S do
8:      if R did not reach the global commit phase for M then
9:          Use M, P, and C to reach the global commit phase for M.

10: event R ∈ S receives messages ⟨VCRecoveryRQ : v_i,ρ,S_i⟩, 1 ≤ i ≤ f_S + 1,
        from f_S + 1 distinct replicas in S do
11:     R detects the need for local state recovery of round ρ of view min{v_i | 1 ≤ i ≤
        f_S + 1}.

12: event R ∈ S receives messages ⟨VCRecoveryRQ : v,ρ,S_i⟩, 1 ≤ i ≤ g_S,
        from distinct replicas in S do
13:     if id(R) ≠ (v+1) mod n_S then
14:         (R awaits the NewView message of the new primary, Line 14 of Figure 6.)
15:     else
16:         Broadcast ⟨NewView : ⟨VCRecoveryRQ : v,ρ,S_i⟩ | 1 ≤ i ≤ g_S⟩ to all replicas
            in S.
17:         if there exists a S_i that contains global preprepare certificate M,
                but no S_j contains a local commit certificate for M then
18:             R initiates global state recovery of round ρ (Line 1 of Figure 6).
```

Figure 5: The view-change *local short-cut recovery path* that determines whether some Q can provide R with the assurance that the current transaction will be committed. If this is the case, then R only needs this assurance, otherwise $S$ requires a new view (Figure 6).

*Proof.* We prove this property using contradiction. We assume $\tau_1 \neq \tau_2$. Let $P_i(S)$ be the local prepare certificate provided by $S$ for $M_i$ and used by $R_i$ to reach the global commit phase, let $S_i \subseteq S$ be the $\mathbf{g}_S$ replicas in $S$ that provided the prepare messages in $P_i(S)$, and let $T_i = S_i \setminus \mathcal{F}(S)$ be the good replicas in $S_i$. By construction, we have $|T_i| \geq \mathbf{g}_S - \mathbf{f}_S$. As all replicas in $T_1 \cup T_2$ are good, they will only send out a single prepare message per round $\rho$ of view $v$. Hence, if $\tau_1 \neq \tau_2$, then $T_1 \cap T_2 = \emptyset$, and we must have $2(\mathbf{g}_S - \mathbf{f}_S) \leq |T_1 \cup T_2|$. As all replicas in $T_1 \cup T_2$ are good, we also have $|T_1 \cup T_2| \leq \mathbf{g}_S$. Hence, $2(\mathbf{g}_S - \mathbf{f}_S) \leq \mathbf{g}_S$, which simplifies to $\mathbf{g}_S \leq 2\mathbf{f}_S$, a contradiction. Hence, we conclude $\tau_1 = \tau_2$. □

Next, we use Lemma 5.1 to prove that any global prepare certificate that *could* have been accepted by any good affected replica is preserved by OCHIMERA:

**Proposition 5.1.** *Let $\tau$ be a transaction and $m(S,\tau)_{v,\rho}$ be a preprepare message. If, for all shards $S^*$, $\mathbf{g}_{S^*} > 2\mathbf{f}_{S^*}$, and there exists a shard $S' \in \text{shards}(\tau)$ such that $\mathbf{g}_{S'} - \mathbf{f}_{S'}$ good replicas in $S'$ reached the global commit phase for M with $m(S,\tau)_{v,\rho} \in M$, then every successful future view of $S$ will recover M and assure that the good replicas in $S$ reach the commit phase for M.*

*Proof.* Let $v^* \leq v$ be the first view in which a global prepare

```
 1: event P ∈ 𝒮 initiates global state recovery of round ρ using ⟨NewView : V⟩ do
 2:     Let T be the transactions with global preprepare certificates for round ρ of 𝒮 in
        view V.
 3:     Let S be the shards affected by transactions in T.
 4:     Broadcast ⟨VCGlobalStateRQ : v,ρ,V⟩ to all replicas in 𝒮' ∈ S.
 5:     for 𝒮' ∈ S do
 6:         Wait for VCGlobalStateRQ messages for V from g_{𝒮'} distinct replicas in 𝒮'.
 7:         Let W(𝒮') be the set of received VCGLOBALSTATERQ messages.
 8:     Broadcast ⟨NewViewGlobal : V,{W(𝒮') | 𝒮' ∈ S}⟩ to all replicas in 𝒮.

 9: event R' ∈ 𝒮' receives message ⟨VCGlobalStateRQ : v,ρ,V⟩ from P ∈ 𝒮 do
10:     if R' has a global preprepare certificate M with m(𝒮,τ)_{w,ρ} ∈ M
            and reached the global commit phase for M then
11:         Let P be the global prepare certificate for M.
12:         Send ⟨VCGlobalStateR : v,ρ,V,M,P⟩ to P.
13:     else Send ⟨VCGlobalStateR : v,ρ,V⟩ to P.

14: event R ∈ 𝒮 receives valid ⟨NewView : V⟩ message from replica P do
15:     if there exists a ⟨VCRecoveryRQ : v_i,ρ,S_i⟩ ∈ V that contains
            a global preprepare certificate M with m(𝒮,τ)_{w,ρ} ∈ M,
            a global prepare certificate P for M, and a local commit certificate C(𝒮'')
        for M then
16:         Use M, P, and C to reach the global commit phase for M.
17:     else if there exists a ⟨VCRecoveryRQ : v_i,ρ,S_i⟩ ∈ V that contains
            a global preprepare certificate M,
            but no ⟨VCRecoveryRQ : v_j,ρ,S_j⟩ ∈ V contains a local commit certificate
        for M then
18:         R detects the need for global state recovery of round ρ (Line 20 of Figure 6).
19:     else (P must propose for round ρ.)

20: event R ∈ 𝒮 receives valid ⟨NewViewGlobal : V,W⟩ from P ∈ 𝒮 do
21:     if any message in W is of the form ⟨VCGlobalStateR : v,ρ,V,M,P⟩ then
22:         Select ⟨VCGlobalStateR : v,ρ,V,M,P⟩ ∈ W with latest view w,
            m(𝒮,τ)_{w,ρ} ∈ M.
23:         Use M and P to reach the global commit phase for M.
24:     else (P must propose for round ρ.)
```

Figure 6: The view-change *new-view recovery path* that recovers the state of the previous view based on a NewView proposal of the new primary. As part of the new-view recovery path, the new primary can construct a global new-view that contains the necessary information from other shards to reconstruct the local state.

certificate $M^*$ with $m(\mathcal{S},\tau^*)_{v^*,\rho} \in M^*$ satisfied the premise of this proposition. Using induction on the number of views after the first view $v^*$, we will prove the following two properties on $M^*$:

1. every good replica that participates in view $w$, $v^* < w$, will recover $M^*$ upon entering view $w$ and reach the commit phase for $M^*$; and

2. no replica will be able to construct a local prepare certificate of $\mathcal{S}$ for any global preprepare certificate $M^\dagger \neq M^*$ with $m(\mathcal{S},\tau^\dagger)_{w,\rho} \in M^\dagger$, $v^* < w$.

The base case is view $v^*+1$. Let $S' \subseteq \mathcal{G}(\mathcal{S}')$ be the set of $\mathbf{g}_{\mathcal{S}'} - \mathbf{f}_{\mathcal{S}'}$ good replicas in $\mathcal{S}'$ that reached the global commit phase for $M^*$. Each replica $\mathrm{R}' \in S'$ has a local prepare certificate $P(\mathcal{S})$ consisting of $\mathbf{g}_{\mathcal{S}}$ prepare messages for $M^*$ provided by replicas in $\mathcal{S}$. We write $S(\mathrm{R}') \subseteq \mathcal{G}(\mathcal{S})$ to denote the at-least $\mathbf{g}_{\mathcal{S}} - \mathbf{f}_{\mathcal{S}}$ good replicas in $\mathcal{S}$ that provided such a prepare message to $\mathrm{R}'$.

Consider any valid new-view proposal $\langle \text{NewView} : V \rangle$ for view $v^*+1$. If the conditions of Line 15 of Figure 6 hold for global preprepare certificate $M^\dagger$ with $m(\mathcal{S},\tau^{\ddagger})_{w,\rho} \in M^{\ddagger}$, then

we recover $M^{\ddagger}$. As there is a local commit certificate for $M^{\ddagger}$ in this case, the premise of this proposition holds on $M^{\ddagger}$. As $v^*$ is the first view in which the premise of this proposition hold, we can use Lemma 5.1 to conclude that $w = v^*$, $M^{\ddagger} = M^*$, and, hence, that the base case holds if the conditions of Line 15 of Figure 6 hold. Next, we assume that the conditions of Line 15 of Figure 6 do not hold, in which case $M^*$ can only be recovered via global state recovery. As the first step in global state recovery is proving that the condition of Line 17 of Figure 6 holds. Let $T \subseteq \mathcal{G}(\mathcal{S})$ be the set of at-least $\mathbf{g}_{\mathcal{S}} - \mathbf{f}_{\mathcal{S}}$ good replicas in $\mathcal{S}$ whose VCRecoveryRQ message is in $V$ and let $\mathrm{R}' \in S'$. We have $|S(\mathrm{R}')| \geq \mathbf{g}_{\mathcal{S}} - \mathbf{f}_{\mathcal{S}}$ and $|T| \geq \mathbf{g}_{\mathcal{S}} - \mathbf{f}_{\mathcal{S}}$. Hence, by a standard quorum argument, we conclude $S(\mathrm{R}') \cap T \neq \emptyset$. Let $\mathrm{Q} \in (S(\mathrm{R}') \cap T)$. As $\mathrm{Q}$ is good and send prepare messages for $M^*$, it must have reached the global prepare phase for $M^*$. Consequently, the condition of Line 17 of Figure 6 holds and to complete the proof, we only need to prove that any well-formed NewViewGlobal message will recover $M^*$.

Let $\langle \text{NewViewGlobal} : V,W \rangle$ be any valid global new-view proposal for view $v^*+1$. As $\mathrm{Q}$ reached the global prepare phase for $M^*$, any valid global new-view proposal must include messages from $\mathcal{S}' \in \text{shards}(\tau)$. Let $U' \subseteq \mathcal{S}'$ be the replicas in $\mathcal{S}'$ of whom messages VCGlobalStateR are included in $W$. Let $V' = U' \setminus \mathcal{F}(\mathcal{S}')$. We have $|\mathcal{S}'| \geq \mathbf{g}_{\mathcal{S}'} - \mathbf{f}_{\mathcal{S}'}$ and $|V'| \geq \mathbf{g}_{\mathcal{S}'} - \mathbf{f}_{\mathcal{S}'}$. Hence, by a standard quorum argument, we conclude $\mathcal{S}' \cap V' \neq \emptyset'$. Let $\mathrm{Q}' \in (\mathcal{S}' \cap V')$. As $\mathrm{Q}'$ reached the global commit phase for $M^*$, it will meet the conditions of Line 23 of Figure 6 and provide both $M^*$ and a global prepare certificate for $M^*$. Let $M^{\ddagger}$ be any other global preprepare certificate in $W$ accompanied by a global prepare certificate. Due to Line 22 of Figure 6, the global preprepare certificate for the newest view of $\mathcal{S}$ will be recovered. As $v^*$ is the newest view of $\mathcal{S}$, $M^{\ddagger}$ will only prevent recovery of $M^*$ if it is also a global preprepare certificate for view $v^*$ of $\mathcal{S}$. In this case, Lemma 5.1 guarantees that $M^{\ddagger} = M^*$. Hence, any replica $\mathrm{R}$ will recover $M^*$ upon receiving $\langle \text{NewViewGlobal} : V,W \rangle$.

Now assume that the induction hypothesis holds for all views $j$, $v^* < j \leq i$. We will prove that the induction hypothesis holds for view $i+1$. Consider any valid new-view proposal $\langle \text{NewView} : V \rangle$ for view $i+1$ and let $M^{\ddagger}$ with $m(\mathcal{S},\tau^{\ddagger})_{w,\rho} \in M^{\ddagger}$ be any global preprepare certificate that is recovered due to the new-view proposal $\langle \text{NewView} : V \rangle$. Hence, $M^{\ddagger}$ is recovered via either Line 16 of Figure 6 or Line 23 of Figure 6. In both cases, there must exist a global prepare certificate $P$ for $M^{\ddagger}$. As $\langle \text{NewView} : V \rangle$ is valid, we must have $w \leq i$. Hence, we can apply the second property of the induction hypothesis to conclude that $w \leq v^*$. If $w = v^*$, then we can use Lemma 5.1 to conclude that $M^{\ddagger} = M^*$. Hence, to complete the proof, we must show that $w = v^*$. First, the case in which $M^{\ddagger}$ is recovered via Line 16 of Figure 6. Due to the existence of a global commit certificate $C$ for $M^{\ddagger}$, $M^{\ddagger}$ satisfies the premise of this proposition. By assumption, $v^*$ is the first view for which the premise of this proposition holds. Hence, $w \geq v^*$,

in which case we conclude $M^{\ddagger} = M^*$. Last, the case in which $M^{\ddagger}$ is recovered via Line 23 of Figure 6. In this case, $M^{\ddagger}$ is recovered via some message $\langle \text{NewViewGlobal} : V, W \rangle$. Analogous to the proof for the base case, $V$ will contain a message VCRecoveryRQ from some replica $Q \in S(R')$. Due to Line 2 of Figure 5, $Q$ will provide information on $M^*$. Consequently, a prepare certificate for $M^*$ will be obtained via global state recovery, and we also conclude $M^{\ddagger} = M^*$. □

Lemma 5.1 and Proposition 5.1 assure that no transaction that could-be-committed by any replica will ever get lost by the system. Next, we bootstrap these technical properties to prove that all good replicas can always recover such could-be-committed transactions.

**Proposition 5.2.** *Let $\tau$ be a transaction and $m(S, \tau)_{v,\rho}$ be a preprepare message. If, for all shards $S^*$, $\mathbf{g}_{S^*} > 2\mathbf{f}_{S^*}$, and there exists a shard $S' \in \text{shards}(\tau)$ such that $\mathbf{g}_{S'} - \mathbf{f}_{S'}$ good replicas in $S'$ reached the global commit phase for $M$ with $m(S, \tau)_{v,\rho} \in M$, then every good replica in $S$ will accept $M$ whenever communication becomes reliable.*

*Proof.* Let $R \in S$ be a good replica that is unable to accept $M$. At some point, communication becomes reliable, after which $R$ will eventually trigger Line 1 of Figure 4. We have the following cases:

1. If $R$ meets the conditions of Line 4 of Figure 4, then $R$ has a local commit certificate $C(S'')$, $S'' \in \text{shards}(\tau)$. This local commit certificate certifies that at least $\mathbf{g}_{S''} - \mathbf{f}_{S''}$ good replicas in $S''$ finished the global prepare phase for $M$. Hence, the conditions for Proposition 5.1 are met for $M$ and, hence, any shard in $\text{shards}(\tau)$ will maintain or recover $M$. Replica $R$ can use $C(S'')$ to prove this situation to other replicas, forcing them to commit to $M$, and provide any commit messages $R$ is missing (Line 11 of Figure 4).

2. If $R$ does not meet the conditions of Line 4 of Figure 4, but some other good replica $Q \in S$ does, then $Q$ can provide all missing information to $R$ (Line 6 of Figure 5). Next, $R$ uses this information (Line 7 of Figure 5), after which it meets the conditions of Line 4 of Figure 4.

3. Otherwise, if the above two cases do not hold, then all $\mathbf{g}_S$ good replicas in $S$ are unable to finish the commit phase. Hence, they perform a view-change. Due to Proposition 5.1, this view-change will succeed and put every replica in $S$ into the commit phase for $M$. As all good replicas in $S$ are in the commit phase, each good replica in $S$ will be able to make a local commit certificate $C(S)$ for $M$, after which they meet the conditions of Line 4 of Figure 4. □

Finally, we use Proposition 5.2 to prove *cross-shard-consistency*.

**Theorem 5.2.** *Optimistic-*CHIMERA *maintains cross-shard consistency.*

*Proof.* Assume a single good replica $R \in S$ executes a transaction $\tau$ (by committing or aborting). Hence, it accepted some global preprepare certificate $M$ with $m(S, \tau)_{v,\rho} \in M$. Consequently, $R$ has local commit certificates $C(S')$ for $M$ of every $S' \in \text{shards}(\tau)$. Hence, at least $\mathbf{g}_{S'} - \mathbf{f}_{S'}$ good replicas in $S'$ reached the global commit phase for $M$, and we can apply Proposition 5.2 to conclude that any good replica $R'' \in S''$, $S'' \in \text{shards}(\tau)$ will accept $M$. As $R''$ bases its execution decision for $\tau$ on the same global prepare certificate $M$ as $R$, they will both make the same decision, completing the proof. □

Due to the similarity between OCHIMERA and CCHIMERA, one can use the details of Theorem 4.3 to prove that OCHIMERA provides *validity*, *shard-involvement*, and *shard-applicability*. Via Theorem 5.2, we proved *cross-shard-consistency*. We cannot prove *service* and *confirmation*, however. The reason for this is simple: even though OCHIMERA can detect and recover from accidental faulty behavior and accidental concurrent transactions, OCHIMERA is not designed to gracefully handle targeted attacks: OCHIMERA is optimistic in the sense that it is optimized for the situation in which faulty behavior (including concurrent transactions that content for the same objects) is rare. Still, in all cases, OCHIMERA maintains cross-shard consistency, however. Moreover, in the optimistic case in which shards have good primaries and no concurrent transactions exist, progress is guaranteed whenever communication is reliable:

**Proposition 5.3.** *If, for all shards $S^*$, $\mathbf{g}_{S^*} > 2\mathbf{f}_{S^*}$, and Assumptions 2.1, 2.2, and 2.3 hold, then Optimistic-*CHIMERA *satisfies Requirements R1–R6 in the optimistic case.*

OCHIMERA cannot defend against denial-of-service attacks targeted at blocking individual replicas and shards from participating. Unfortunately, no existing consensus protocol is able to deal with such attacks. Furthermore, as is the case for other multi-shard consensus protocols, coordinated attempts can prevent OCHIMERA from making progress in periods when the optimistic assumption does not hold. At the core of such attacks is the ability for malicious clients and malicious primaries to corrupt the operations of shards coordinated by good primaries, as already shown in Example 5.1. Due to Theorem 5.2, such attacks will *never* affect consistency in OCHIMERA, however.

To further reduce the impact of targeted attacks, one can make primary election non-deterministic, e.g., by using shard-specific distributed coins to elect new primaries in individual shards [11, 13]. Finally, we remark that we have presented OCHIMERA with a per-round checkpoint and recovery method. In this simplified design, the recovery path only has to recover at-most a single round. Our approach can easily be generalized to a more typical multi-round checkpoint and

recovery method, however. Furthermore, we believe that the way in which OCHIMERA extends PBFT can easily be generalized to other consensus protocols, e.g., POE [32] and HOTSTUFF [59].

# 6 Resilient-CHIMERA: Transaction Processing Under Attack

In the previous section, we introduced OCHIMERA, a general-purpose minimalistic and efficient multi-shard transaction processing protocol. OCHIMERA is designed with the assumption that malicious behavior is rare, due to which it can minimize coordination in the normal-case while requiring intricate coordination when recovering from attacks. As an alternative to the optimistic approach of OCHIMERA, we can apply a *pessimistic* approach to CCHIMERA to gracefully recover from concurrent transactions that is geared towards minimizing the influence of malicious behavior altogether (without requiring Assumption 2.3). Next, we explore such a pessimistic design via *resilient*-CHIMERA (RCHIMERA).

The design of RCHIMERA builds upon the design of CCHIMERA by adding additional coordination to the cross-shard exchange and decide outcome steps. As in CCHIMERA, the acceptance of $m(\mathcal{S}, \tau)_\rho$ in round $\rho$ by all good replicas completes the *local inputs* step. Before cross-shard exchange, the replicas in $\mathcal{S}$ destruct the objects in $D(\mathcal{S}, \tau)$, thereby fully pledging these objects to $\tau$ until the commit or abort decision. Then, $\mathcal{S}$ performs cross-shard exchange by broadcasting $m(\mathcal{S}, \tau)_\rho$ to all other shards in $\mathrm{shards}(\tau)$, while the replicas in $\mathcal{S}$ wait until they receive messages $m(\mathcal{S}', \tau)_{\rho'}$ from all other shards $\mathcal{S}' \in \mathrm{shards}(\tau)$.

After cross-shard exchange comes the final *decide outcome* step. After $\mathcal{S}$ receives $m(\mathcal{S}', \tau)_{\rho'}$ from all shards $\mathcal{S}' \in \mathrm{shards}(\tau)$, the replicas force a *second consensus step* that determines the round $\rho^*$ at which $\mathcal{S}$ decides *commit* (whenever $I(\mathcal{S}', \tau) = D(\mathcal{S}', \tau)$ for all $\mathcal{S}' \in \mathrm{shards}(\tau)$) or *abort*. If $\mathcal{S}$ decides commit, then, in round $\rho^*$, all good replicas in $\mathcal{S}$ construct all objects $o \in \mathrm{Outputs}(\tau)$ with $\mathcal{S} = \mathrm{shard}(o)$. If $\mathcal{S}$ decides abort, then, in round $\rho^*$, all good replicas in $\mathcal{S}$ reconstruct all objects in $D(\mathcal{S}, \tau)$ (rollback). Finally, each good replica informs $c$ of the outcome of execution. If $c$ receives, from every shard $\mathcal{S}' \in \mathrm{shards}(\tau)$, identical outcomes from $\mathbf{g}_{\mathcal{S}'} - \mathbf{f}_{\mathcal{S}'}$ distinct replicas in $\mathcal{S}'$, then it considers $\tau$ to be successfully executed. In Figure 7, we sketched the working of RCHIMERA.

We notice that processing a multi-shard transaction via RCHIMERA requires *two* consensus steps per shard. In some cases, we can eliminate the second step, however. First, if $\tau$ is a multi-shard transaction with $\mathcal{S} \in \mathrm{shards}(\tau)$ and the replicas in $\mathcal{S}$ accept $(\langle\tau\rangle_c, I(\mathcal{S}, \tau), D(\mathcal{S}, \tau))$ with $I(\mathcal{S}, \tau) \neq D(\mathcal{S}, \tau)$, then the replicas can immediately abort whenever they accept $(\langle\tau\rangle_c, I(\mathcal{S}, \tau), D(\mathcal{S}, \tau))$. Second, if $\tau$ is a single-shard transaction with $\mathrm{shards}(\tau) = \{\mathcal{S}\}$, then the replicas in $\mathcal{S}$ can

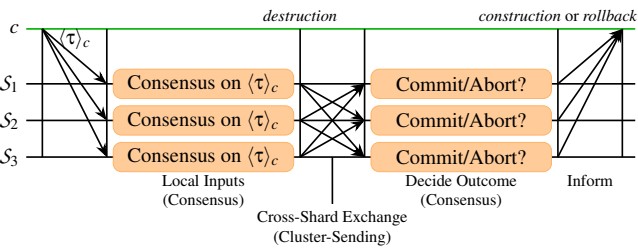

Figure 7: The message flow of RCHIMERA for a 3-shard client request $\langle\tau\rangle_c$ that is committed.

immediately decide commit or abort whenever they accept $(\langle\tau\rangle_c, I(\mathcal{S}, \tau), D(\mathcal{S}, \tau))$. Hence, in both cases, processing of $\tau$ at $\mathcal{S}$ only requires a single consensus step at $\mathcal{S}$. Next, we prove the correctness of RCHIMERA:

**Theorem 6.1.** *If, for all shards $\mathcal{S}^*$, $\mathbf{g}_{\mathcal{S}^*} > 2\mathbf{f}_{\mathcal{S}^*}$, and Assumptions 2.1, 2.2, and 2.3 hold, then Resilient-CHIMERA satisfies Requirements R1–R6.*

*Proof.* Let $\tau$ be a transaction. As good replicas in $\mathcal{S}$ discard $\tau$ if it is invalid or if $\mathcal{S} \notin \mathrm{shards}(\tau)$, RCHIMERA provides *validity* and *shard-involvement*. Next, *shard-applicability* follow directly from the decide outcome step.

If a shard $\mathcal{S}$ commits or aborts transaction $\tau$, then it must have completed the decide outcome and cross-shard exchange steps. Hence, all shards $\mathcal{S}' \in \mathrm{shards}(\tau)$ must have exchanged the necessary information to $\mathcal{S}$. By relying on cluster-sending for cross-shard exchange, $\mathcal{S}'$ requires cooperation of all good replicas in $\mathcal{S}'$ to exchange the necessary information to $\mathcal{S}$. Hence, we have the guarantee that these good replicas will also perform cross-shard exchange to any other shard $\mathcal{S}'' \in \mathrm{shards}(\tau)$. Consequently, every shard $\mathcal{S}'' \in \mathrm{shards}(\tau)$ will receive the same information as $\mathcal{S}$, complete cross-shard exchange, and make the same decision during the decide outcome step, providing *cross-shard consistency*.

A client can force service on a transaction $\tau$ by choosing a shard $\mathcal{S} \in \mathrm{shards}(\tau)$ and sending $\tau$ to all good replicas in $\mathcal{G}(\mathcal{S})$. By doing so, the normal mechanisms of consensus can be used by the good replicas in $\mathcal{G}(\mathcal{S})$ to force acceptance on $\tau$ in $\mathcal{S}$ and, hence, bootstrapping acceptance on $\tau$ in all shards $\mathcal{S}' \in \mathrm{shards}(\tau)$. Due to cross-shard consistency, every shard in $\mathrm{shards}(\tau)$ will perform the necessary steps to eventually inform the client. As all good replicas $R \in \mathcal{S}$, $\mathcal{S} \in \mathrm{shards}(\tau)$, will inform the client of the outcome for $\tau$, the majority of these inform-messages come from good replicas, enabling the client to reliably derive the true outcome. Hence, RCHIMERA provides *service* and *confirmation*. $\square$

As with CCHIMERA, RCHIMERA depends on underlying consensus and cluster-sending protocols and the level to which RCHIMERA can deal with asynchronous behavior depends on the particular choices of these protocols.

## 7 The Ordering of Transactions in CHIMERA

Having introduced the three variants of CHIMERA in Sections 4, 5, and 6, we will now analyze the ordering guarantees provided by CHIMERA. We further refer to Section 8 for a detailed comparison of the three variants of CHIMERA. Here, we will show that CHIMERA provides serializable execution [6, 9].

The data model utilized by CCHIMERA, OCHIMERA, and RCHIMERA guarantees that any object $o$ can only be involved in at-most *two* committed transactions: one that *constructs* $o$ and another one that *destructs* $o$. Assume the existence of such transactions $\tau_1$ and $\tau_2$ with $o \in \mathtt{Outputs}(\tau_1)$ and $o \in \mathtt{Inputs}(\tau_2)$. Due to *cross-shard-consistency* (Requirement R4), the shard $\mathtt{shard}(o)$ will have to execute both $\tau_1$ and $\tau_2$. From these observations, we can derive a serializable order on all committed transactions:

**Theorem 7.1.** *A sharded fault-tolerant system that uses the object-dataset data model, processes UTXO-like transactions, and satisfies Requirements R1-R5 commits transactions in a serializable order.*

*Proof.* Assume the existence of transactions $\tau_1$ and $\tau_2$ with $o \in \mathtt{Outputs}(\tau_1)$ and $o \in \mathtt{Inputs}(\tau_2)$. Due to *shard-applicability* (Requirement R3), shard $\mathtt{shard}(o)$ will execute $\tau_1$ strictly before $\tau_2$. Now consider the relation

$$\prec := \{(\tau, \tau') \mid (\text{the system committed to } \tau \text{ and } \tau') \wedge \\ (\mathtt{Outputs}(\tau) \cap \mathtt{Inputs}(\tau') \neq \emptyset)\}.$$

Obviously, we have $\prec(\tau_1, \tau_2)$. To prove that all committed transactions are executed in a *serializable* ordering, we first prove the following:

> If we interpret transactions as nodes and $\prec$ as an edge relation, then the resulting graph is *acyclic*.

The proof is by contradiction. Let $G$ be the graph-interpretation of $\prec$. We assume that graph $G$ is cyclic. Hence, there exists transactions $\tau_0, \dots, \tau_{m-1}$ such that $\prec(\tau_i, \tau_{i+1})$, $0 \leq i < m-1$, and $\prec(\tau_{m-1}, \tau_0)$. By the definition of $\prec$, we can choose objects $o_i$, $0 \leq i < m$, with $o_i \in (\mathtt{Outputs}(\tau_i) \cap \mathtt{Inputs}(\tau_{(i+1) \bmod m}))$. Due to *cross-shard-consistency* (Requirement R4), the shard $\mathtt{shard}(o_i)$, $0 \leq i < m$, executed transactions $\tau_i$ and $\tau_{(i+1) \bmod m}$. Consider $o_i$, $0 \leq i < m$, and let $t_i$ be the time at which shard $\mathtt{shard}(o_i)$ executed $\tau_i$ and constructed $o_i$. Due to *shard-applicability* (Requirement R3), we know that shard $\mathtt{shard}(o_i)$ executed $\tau_{(i+1) \bmod m}$ strictly after $t_i$. Moreover, also shard $\mathtt{shard}(o_{(i+1) \bmod m})$ must have executed $\tau_{(i+1) \bmod m}$ strictly after $t_i$ and we derive $t_i < t_{(i+1) \bmod m}$. Hence, we must have $t_0 < t_1 < \cdots < t_{m-1} < t_0$, a contradiction. Consequently, $G$ must be acyclic.

To derive a serializable execution order for all committed transactions, we simply construct a directed acyclic graph in which transactions are nodes and $\prec$ is the edge relation. Next, we *topologically sort* the graph to derive the searched-for ordering. □

We notice that CHIMERA only provides serializability for *committed* transactions: concurrent transactions that content for the same objects will always be aborted and, hence, will not be executed and will not affect the serializable order of execution of transactions. It is this flexibility in dealing with aborted transactions that allows all variants of CHIMERA to operate with minimal and fully-decentralized coordination between shards; while still providing strong isolation for all committed transactions.

## 8 Analysis of the Three CHIMERA Variants

In the previous sections, we proposed three variants of CHIMERA and showed their correctness. Next, we analyze the benefits and costs of the three CHIMERA multi-shard transaction processing protocols, compare them with state-of-the-art multi-shard transaction processing protocols, and evaluate the impact of malicious behavior on CHIMERA. A summary of this analysis can be found in Figure 8.

We did not detail the exact message complexity of the three CHIMERA protocols. For CCHIMERA and RCHIMERA we measure the complexity in the number of consensus steps and cluster-sending steps they require. Implementation-wise, one can choose to either implement these steps with all-to-all communication (as in PBFT) or with all-to-one-to-all communication (as in HOTSTUFF) to optimize for either low latency or low bandwidth usage. Similarly, we can implement also OCHIMERA with all-to-one-to-all communication instead of all-to-all communication.

*Remark* 8.1. A common technique to improve the transaction throughput of consensus-based systems is by processing a batch holding many transactions per consensus decision. To simplify presentation, we have chosen to present the three CHIMERA protocols without such batching. Both CCHIMERA and RCHIMERA can easily be generalized to process transactions in batches: at a per-shard level, they use standard consensus protocols that operate independently of other shards. Hence, instead of one transaction per consensus decision, both can include a batch of transactions in their consensus decisions (after which they perform the steps related to the transactions in that batch in order).

For OCHIMERA, blocks of transactions are more challenging as OCHIMERA uses a single multi-shard consensus step that includes all replicas of all shards affected by a transaction. Still, OCHIMERA can be generalized to process batches of transactions that affect the same set of shards. Such a generalization requires additional machinery, however: multi-shard batches can lead to several shards proposing distinct batches that include the same transaction, however. It is possible to

deal with such issues with existing techniques (e.g., by assigning each transaction to a single batch-proposal shard based on the digest of that transaction) [30].

## 8.1 A Comparison of CHIMERA Variants

First, Figure 8 provides a high-level comparison of the costs of each of the three CHIMERA protocols to process a single transaction $\tau$ that affects $s = |\text{shards}(\tau)|$ distinct shards. For the normal-case behavior, we compare the complexity in the number of *sequential communication phases* (which, in the idle case, are the main determinant for client latencies), the number of *consensus steps* per shard and *cross-shard exchange* steps between shards (which together determine the bandwidth costs and put an upper bound on throughput). As one can see, all three protocols have a low number of *phases*, due to which all three can provide low latencies toward clients. In environments in which cross-shard communication has low latency, OCHIMERA will be able to provide lower latencies than both CCHIMERA and RCHIMERA, as its optimistic design eliminates one phase of communication (at the cost of requiring cross-shard communication in every phase).

Next, we compare how the three protocols deal with malicious behavior by clients and by replicas. If no clients behave malicious, then all transactions will *commit*. In all three protocols, malicious behavior by clients can lead to the existence of concurrent transactions that affect the same object. Upon detection of such concurrent transactions, all three protocols will *abort*. The consequences of such an abort are different in the three protocols.

In CCHIMERA, objects affected by aborted transactions remain pledged and cannot be reused. In practice, this loss of objects can provide an incentive for clients to not behave malicious, but does limit the usability of CCHIMERA in non-incentivized environments. Both OCHIMERA and RCHIMERA deal with concurrent transactions by aborting them via the normal-case of the protocol. The three CHIMERA protocols are resilient against malicious replicas: only malicious primaries can affect the normal-case operations of these protocols. If the behavior of a primary is disrupting the normal-case operations, then in CCHIMERA and RCHIMERA such behavior is dealt with by the recovery mechanisms of the underlying consensus protocol (e.g., in PBFT such disruptions will eventually lead to a view-change whenever communication is reliable), whereas OCHIMERA will utilize the view-change recovery mechanisms outlined in Section 5. In both CCHIMERA and RCHIMERA, dealing with a malicious primary in a shard can be done completely in isolation of all other shards. In OCHIMERA, which is optimized with the assumption that failures are rare, the failure of a primary while processing a transaction $\tau$ can lead to view-changes in all shards affected by $\tau$.

In conclusion, we see that the three CHIMERA variants each make their own tradeoff between *normal-case costs* and

ability to deal with faulty behavior (by both clients and other replicas), with RCHIMERA being robust against any attack at the cost of 2 consensus decisions per transaction per involved shard.

## 8.2 Comparison With the State-of-the-Art

Several recent papers have proposed specialized systems that combine sharding with consensus-based resilient systems. Examples include systems such as AHL [17], BYSHARD [34], CAPER [3], CHAINSPACE [1], RINGBFT [53], and SHARPER [4], which all use sharding for data management and transaction processing. Next, we compare the design of CHIMERA in detail with AHL [17], CHAINSPACE [1], RINGBFT [53], and SHARPER [4], and briefly look at BYSHARD [34] and CAPER [3].

**AHL [17].** AHL uses a *centralized* commit protocol to order all multi-shard transactions. In specific, AHL [17] uses a reference committee that leads a *centralized two-phase commit protocol* (Centralized 2PC) [29, 51] that is implemented via consensus steps and cluster-sending. Furthermore, AHL uses non-blocking locks to provide transaction isolation due to which valid transactions can be aborted, whereas in CHIMERA only faulty transactions (e.g., by malicious clients) are aborted. By using Centralized 2PC, AHL eliminates any all-to-all communication between shards affected by a transaction in favor of one-to-all communication between the reference committee and the affected shards. Due to this, AHL takes five consecutive consensus rounds, more than twice the number of rounds required by the costliest CHIMERA variants. As reported in the original evaluation of AHL [17, Section 7.3], the reference committee will become a bottleneck for performance when processing workloads heavy in multi-shard transactions (even if none of these transactions are concurrent), while AHL shows excellent performance when processing single-shard transactions [34].

**CHAINSPACE [1].** CHAINSPACE uses a *distributed two-phase commit protocol* (Distributed 2PC) [29, 51], that is implemented via consensus steps and cluster-sending, to order all multi-shard transactions. Furthermore, similar to AHL, CHAINSPACE uses non-blocking locks to provide transaction isolation due to which valid transactions can be aborted. The operations of this commit protocol are similar to the design of RCHIMERA, except that CHAINSPACE does not take advantage of any specific properties of the data model (e.g., to provide isolation). A further minor difference between CHAINSPACE and RCHIMERA is that CHAINSPACE distinguishes between shards that are used as inputs and shards that are used as outputs and only informs output shards after the input shards finish processing a transaction, due to which transaction processing in CHAINSPACE takes one more round as in RCHIMERA.

| Protocol | Principle Technique | Phases[a] (Cross-Shard) | Consensus Steps | | Cross-Shard Communication[b] | Transaction Abort Causes | Transaction Concurrency and Ordering | Failure Recovery (method and when) |
|---|---|---|---|---|---|---|---|---|
| | | | Total | Sequential | | | | |
| CCHIMERA | Independent Consensus UTXO Data Model | 4 (1) | $s$ | 1 | 1 (CS, A2A) | Faulty Only | Data Model Pledges (Incentive) | Local Recovery Local Primary Failure |
| OCHIMERA | Multi-Shard Consensus UTXO Data Model | 3 (3) | $s$ | 1 | 3 (GC, A2A) | Faulty Only | Data Model Aborts | Local and Global Recovery Any Primary Failure |
| RCHIMERA | Distributed Commit UTXO Data Model | 7 (1) | $2s$ | 2 | 1 (CS, A2A) | Faulty Only | Data Model Aborts | Local Recovery Local Primary Failure |
| AHL [17] | Reference Committee Non-Blocking Locks | 19 (4) | $2s+2$ | 5 | 4 (CS, O2A) | Failed Locks | Reference Committee Locks & Aborts | Local Recovery Local Primary Failure |
| CHAINSPACE [1] | Distributed Commit locking Locks | 11 (2) | $2s$ | 3 | 2 (CS, A2A) | Failed Locks | Distributed Commit Locks & Aborts | Local Recovery Local Primary Failure |
| RINGBFT [53] | Linear Commit Blocking Locks | $8s - 5$ ($2s - 2$) | $2s - 1$ | $2s - 1$ | $2s - 2$ (CS, O2O) | Invalid Only | Linear Commit Blocking Locks | Local Recovery Local Primary Failure |
| SHARPER [4] | Multi-Shard Consensus Shard-Wide Blocking Locks | 3 (3) | $s$ | 1 | 3 (GC, A2A) | Failed Locks (Shard-Wide) | Multi-Shard Consensus Shard-Wide Locks & Aborts | Global Recovery Any Primary Failure, Concurrency |

[a]Total number of consecutive communication phases. For protocols that use a local consensus protocol, we count three consecutive phases per consensus step (e.g., using PBFT), and we count a single phase per cluster-sending step.

[b]We write *CS* to indicate *cluster-sending* and *MS* to indicate *multi-shard consensus*; and we write *A2A* to denote all-to-all communication, O2A to denote one-to-all or all-to-one communication, and O2O to denote one-to-one communication between involved shards.

Figure 8: Comparison of the three CHIMERA protocols for processing a transaction that affects $s$ shards. We compare the normal-case complexity. the mechanism used to deal with concurrent transactions (due to malicious clients), and the mechanisms used to provide failure recovery.

**RINGBFT [53].** RINGBFT uses a *linear two-phase commit protocol* (Linear 2PC) [29, 51], that is implemented via consensus steps and cluster-sending, to order all multi-shard transactions. Due to the usage of Linear 2PC, RINGBFT is able to utilize blocking locks in a deadlock-free manner to provide transaction isolation. Due to this usage of locks, RINGBFT is the only protocol besides CHIMERA that is able to always process valid transactions without spurious aborts. Furthermore, the usage of Linear 2PC minimizes cross-shard communication costs, as all communication is between pairs-of-affected-shards (no all-to-all, one-to-all, or all-to-one communication). The benefits of RINGBFT come at a cost, however, as the linear design imposes a *linear* amount of consecutive consensus and cross-shard communication steps in terms of the shards affected by the transaction, whereas all other proposals require a constant number of consecutive steps.

**SHARPER [4].** SHARPER uses a *multi-shard consensus protocol* to order all multi-shard transactions. The operations of this multi-shard consensus protocol are conceptually similar to the design of OCHIMERA, except that SHARPER does not take advantage of any specific properties of the data model (e.g., to provide isolation or to simplify recovery). Furthermore, SHARPER requires that affected shards process their multi-shard transactions in a common processing order, due to which SHARPER can only processing a single multi-shard transaction at a time. In effect, this imposes a per-shard lock on multi-shard transaction processing, limiting concurrent execution even in the absence of transactions that content for the same data objects. Finally, the philosophy of SHARPER is to serve as a single unified protocol that can support both PAXOS-style crash fault-tolerance and malicious behavior, and it remains an important research question as to whether SHARPER can be extended to the general-purpose unreliable communication and attack models supported by OCHIMERA. In specific, we believe OCHIMERA improves on the resilience of SHARPER by providing a *robust* local and global view-change mechanism that can deal with per-shard replica failures, per-shard primary failures, and coordinated attacks by replicas and clients to disrupt global consensus steps.

**BYSHARD [34] and CAPER [3].** BYSHARD [34] proposes a framework in which one can evaluate many distinct protocols based on the application of two-phase commit and two-phase locking in a consensus-based environment. Specific instances of BYSHARD correspond with the approaches taken by CHAINSPACE and RINGBFT, while AHL can be seen as a restricted case of the BYSHARD protocols that utilize distributed orchestration. The differences between, on the one hand, CHIMERA and, on the other hand, AHL, CHAINSPACE, and RINGBFT, extend to the BYSHARD framework. The design of CAPER [3] shares similarities with the design of SHARPER.

## 8.3 The Performance Potential of CHIMERA

Next, we modelled the performance benefits of CHIMERA. To do so, we have modeled the maximum throughput of each of these protocols in an environment where each shard has seven replicas (of which two can be faulty) and each replica has a bandwidth of 1 Gbit/s. We have chosen to optimize CCHIMERA, OCHIMERA, and RCHIMERA to minimize *pro-*

Figure 9: Throughput of the three CHIMERA protocols as a function of the number of shards when processing a workload in which 50% of the transactions are multi-shard transactions.

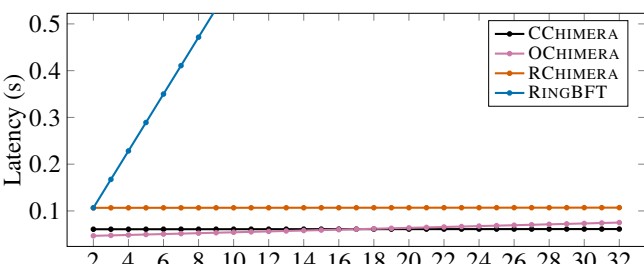

Figure 10: Latency of processing a transaction during multi-shard transaction processing as a function of the number of affected shards, assuming that the transaction affects 64 objects, the network has a bandwidth of 1 Gbit/s, the message delay is 15 ms, and each shard has 7 replicas.

*cessing latencies* over minimizing bandwidth usage, as reducing processing latencies is the goal of the design of CHIMERA. In specific, we do *not* use request batching, we use a one-phase broadcast-based cross-shard exchange steps, and we do not use *threshold signatures*. In cases when one does not want to optimize for processing latencies and individual replicas have spare computational power, then one can utilize threshold signatures to further boost throughput by a constant factor (at the cost of the per-transaction processing latency).

In Figure 9, we have visualized the maximum attainable throughput (the number of transactions processed per second while processing a workload of 32M transactions) for each of the protocols as a function of the number of shards and as a function of the number of objects affected by each transaction when processing a workload with 50% multi-shard

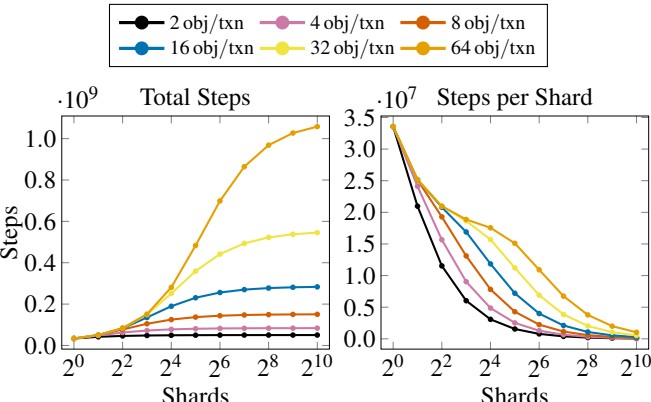

Figure 11: Cumulative number of shards affected by a workload of 32M transactions as a function of the size of transactions and the number of shards. On the *left*, the *total* number of steps. On the *right*, the *average* number of steps per shard.

transactions. As a baseline for comparison, we have also included AHL [17]. For AHL, we used an additional shard as a reference committee (hence, if we use $n$ shards in the experiment, then AHL can use $n + 1$). We have chosen to *only* include AHL in our comparison as it has a similar design goal as the three CHIMERA protocols, while having a substantial different design. We excluded CHAINSPACE, as it has a design very similar to RCHIMERA (but slightly more costly due to some overheads RCHIMERA can avoid). For RINGBFT, we notice that it is optimized for a completely different design goal: RINGBFT is optimized to minimize communication costs and maximize throughput (by inducing very high latencies for multi-shard transactions), whereas the three CHIMERA protocols aim to maximize throughput while also minimizing latency. We refer to Figure 10 for a comparison of the multi-shard transaction processing latencies for the three CHIMERA protocols and for RINGBFT.

In our workloads, the *ratio of multi-shard transactions* is high: we want to study how the multi-shard transaction processing protocols we compare differ in their operations and we are especially interested in the performance of the system when dealing with *multi-shard transactions* that require substantial coordination to deal with contention. Indeed, in workloads that mainly consist of *single-shard transactions*, each of the multi-shard transaction protocols we look at will fall back to the same underlying single-shard consensus protocol to efficiently process such single-shard transactions.

In Figure 11, we have visualized the number of per-shard steps performed by the system (for CCHIMERA and OCHIMERA, this is equivalent to the number of per-shard consensus steps, for RCHIMERA this is half the number of per-shard consensus steps). In general, we see that an increase in shards has two effects:

1. Simple single-shard transactions can be dispersed over

more shards. Hence, increasing the number of shards will reduce the average number of shard steps each shard has to process with respect to *single-shard transactions*. Furthermore, significantly increasing the number of shards will distribute the *multi-shard transactions* over these shards, reducing the cost of these transactions per shard. Both effects will result in drastically improved performance when scaling to many shards.

2. Large transactions can become more complex when increasing the number of shards, as more shards can hold objects relevant of large transactions. For example, a transaction that affects 16 objects in an environment with four shards can affect at-most four shards, while in an environment with 16 shards it can affect at-most 16 shards). Hence, for large transactions, we only see reductions in the per-shard cost to process these transactions when scaling beyond the number of shards large transactions can affect.

In a general-purpose sharded system without any specific bottlenecks, the above will result in great scalability as soon as the number of shards far outgrows the size of transactions. This behavior is clearly observable for all three CHIMERA protocols. Indeed, all three CHIMERA protocols have excellent scalability: increasing the number of shards will increase the overall throughput of the system. Sharding does come with clear overheads, however, increasing the number of shards also increases the number of shards affected by each transaction, thereby increasing the overall number of consensus steps. This is especially true for very large transactions that affect many objects (that can affect many distinct shards). Hence, as one can see from the results, the benefits of sharding are the strongest when processing mainly single-shard transactions or when scaling beyond the size of individual transactions.

In the comparison between OCHIMERA and RCHIMERA, we see that OCHIMERA (implemented with all-to-all communication) will outperform RCHIMERA whenever transactions involve few shards (due to them involving few objects). In this case, the communication cost of the three cross-shard communication steps that are part of the multi-shard consensus of OCHIMERA is lower than the cost of the second local consensus round in RCHIMERA. When transactions affect many shards, RCHIMERA outperforms OCHIMERA, as RCHIMERA only has a single cross-shard communication step per transaction (and all other communication is local within a shard). In all cases, CCHIMERA will outperform the other protocols with respect to transaction throughput. Furthermore, by design CCHIMERA will always have lower latencies than RCHIMERA (due to the fewer consensus and cluster-sending steps CCHIMERA performs). In environments in which inter-shard and intra-shard communication have similar (high) message delays, OCHIMERA will typically have lower latencies than CCHIMERA due to the large impact message delays have on the latency of consensus and cluster-sending

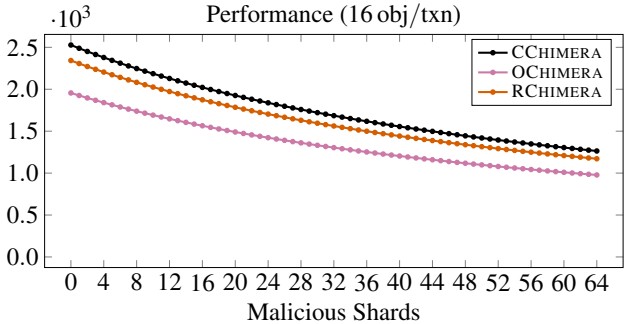

Figure 12: Throughput of the three CHIMERA protocols as a function of the number of malicious shards, as measured by the transaction cost at each shard.

steps, this even when CCHIMERA has higher throughput than OCHIMERA.

In comparison with AHL, we see a large improvement in performance. Due to the high ratio of multi-shard transactions, the performance of AHL for processing multi-shard transactions is bottlenecked by the throughput of the reference committee used by AHL. These findings are in line with the original evaluation of AHL [17, Section 7.3]. A closer look at the data does reveal *excellent* scalability of AHL with regards to single-shard transactions: although the reference committee has a full load while processing all multi-shard transactions, all shards *except* the reference committee show a very low load (that can be used to process many more single-shard transactions during the experiment).

### 8.4 CHIMERA and Malicious Behavior

Finally, we have modeled the maximum throughput of each of the three CHIMERA protocols in an environment in which some shards are impacted by malicious replicas. Unless stated otherwise, we use the same setting as in Section 8.3. We have used 64 shards and four objects per shard and we measure the performance of the three CHIMERA protocols as a function of the number of shards that are affected by malicious behavior.

Within a shard, only malicious primaries have a strong impact on the performance of that shard. Furthermore, malicious primaries that completely disrupt normal-case operations will be replaced. Hence, to maximize the malicious impact, we have chosen for malicious primaries that try to *throttle* the performance of the system by slowing down their own operations. In the experiment, we have chosen that these primaries do so by halving the speed by which their shards operate.

In Figure 12, we have visualized the average attainable throughput for shards processing workloads in which 50% of the transactions are multi-shard as a function of the number of shards that are affected by malicious behavior (and can affect the speed by which some of the multi-shard transactions processed are processed) for each of the CHIMERA protocols. We see that each malicious shard will slow down each CHIMERA

protocol with respect to those transactions that are affected by these shards. At the same time, transactions not handled by malicious shards are unaffected and will be processed at normal speed.

## 9 Related Work

Distributed systems are typically employed to either increase reliability (e.g., via consensus-based fault-tolerance) or to increase performance (e.g., via sharding). Consequently, there is abundant literature on such distributed systems, distributed databases, and sharding (e.g., [51, 55, 56]) and on consensus-based fault-tolerant systems (e.g., [10, 14, 19, 30, 55]). Furthermore, in Section 8.2, we reviewed related work on multi-shard permissioned consensus-based systems. Next, we focus on other works that deal with sharding in fault-tolerant systems.

In Section 8.2, we have only compared with other sharded resilient systems with similar environmental assumptions. Besides these sharded systems, several other resilient systems such as OMNILEDGER [42] and RAPIDCHAIN [61] have been proposed. These systems make very different environmental assumptions (e.g., different threat and communication models) due to which these systems are incomparable to the CHIMERA protocols and the systems considered in Section 8.2.

A few fully-replicated consensus-based systems utilize sharding at the level of consensus decision making, this to improve consensus throughput *without* adopting a multi-shard design [2, 22, 26, 31]. In these systems, only a small subset of all replicas, those in a single shard, participate in the consensus on any given transaction, thereby reducing the costs to replicate this transaction without improving storage and processing scalability.

Recently, there has also been promising work on sharding and techniques supporting sharding for permissionless blockchains. Examples include techniques to enable sidechains, blockchain relays, and atomic swaps [23, 24, 36, 38, 43, 58, 60], which each enable various forms of cooperation between blockchains (including simple cross-chain communication and cross-chain transaction coordination). Unfortunately, these permissionless techniques are several orders of magnitudes slower than comparable techniques for traditional fault-tolerant systems, making them incomparable with the design of CHIMERA discussed in this work.

## 10 Conclusion

In this paper, we took a new look at the problem of multi-shard transaction processing in consensus-based systems. In specific, we proposed the study of *sharded consensus-based systems* that use restrictions on the workloads supported to improve performance over general-purpose methods.

To initiate this study, we introduced Core-CHIMERA, Optimistic-CHIMERA, and Resilient-CHIMERA, three fully distributed approaches towards multi-shard fault-tolerant transaction processing. The design of these approaches is geared towards processing UTXO-like transactions in sharded distributed ledger networks. Due to the usage of UTXO-like transactions, the three CHIMERA variants can minimize cost to an absolute minimum, while maximizing performance, thereby showing the potential of *restricting the types of supported workloads*. This potential is further underlined by our comparison with the state-of-the-art protocols, in which we see that the three CHIMERA variants both have lower costs and complexity.

Although the workloads supported by CHIMERA are minimalistic, we believe that our results can be generalized to more-general settings. In specific, we believe that the combination of sharding and *Conflict-free Replicated Data Types* (CRDTs) [46] has great potential to provide high performance in a consensus-based environment.

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

# Supplemental Materials

See attached the rebuttal comments based on the previous reviews. We have applied the changes proposed in these rebuttal comments in the revised paper.

# Response to the Reviewers' comments
## *"Problem: Sharded Resilient Transaction Processing with Minimal Costs"*

Anonymous authors

October 1, 2022

We like to thank each of the three reviewers for their careful review and their detailed feedback of our previous manuscript. To address the reviewers comments, we have detailed below our rebuttal to the reviewers comments and the corresponding revisions (in red).

**Review #1 (reviewer 3UEy)**

> **Review:** The most major element that should be improved is the accuracy and subjectiveness. The paper mentions "absolute minimum amount of coordination" repeatedly but does not offer any method or model to quantify whether this is indeed the case.
>
> - Even in the CChimera protocol (the simplest), the step #2 cross-shard exchange entails an all-to-all communication across all shards in a given transaction.
> - It is not clear if this is truly the "absolute" minimum, as I can imagine other approaches with various tradeoffs that would still work in the given model (ex: the client handling the cross-shard coordination step).
> - The paper should either (a) rephrase their contribution to make it more accurate and less subjective, or (b) discuss why their proposed approach offers the minimum amount of coordination.

We agree that our original description can be made more precise as to what being *minimized* mean. Furthermore, in all three protocols we present various implementations are possible to trade off between bandwidth and latency (e.g., all-to-all communication versus all-to-one-to-all communication using threshold signatures).

Both CCHIMERA and OCHIMERA *minimize* the number of agreement (consensus) steps each involved shard performs, as they both only perform one such step. We believe that *one* agreement step is the minimum in any sharded environment if a form of serializable transaction execution is to be provided.

> **Revise:** We will include the above clarification in the revision.

> **Review:** I enjoyed seeing how the paper builds on the insight of "limiting the types of workload supported". Generally, this kind of leveraging of application-specific semantics is indeed very powerful. Since this paper applies itself specifically to the UTXO model, and praises "minimum" coordination, it would be helpful to explain why is consensus even necessary within a cluster, since that part was not clear. In the local inputs step, in particular, it seems like a Byzantine reliable broadcast primitive might have sufficed. I would be curious to see an explanation on why is consensus necessary.
>
> . . .
>
> The solution in §4 seems to rely on a key property that the "set of currently-available inputs at S" (a shard) is known, which is a set in constant change, but it was not clear to me how this set was determined. This is likely related to the UTXO model from what I can tell.

As the transactions are UTXO-like, one can determine from the structure of a transaction what its input and output objects are. Furthermore, each object is maintained by exactly a single shard. Hence, while processing a transaction, one only has to check whether the affected input objects are available at those shards that should maintain them. To assure that this availability check is done consistently among all replicas in an affected shard, all replicas in that shard need to process their shard-specific availability checks for distinct transactions in the same order, as otherwise they will not agree on which objects are available for which transactions. As such a strict ordering of these shard-specific steps is required, which is naturally provided by consensus.

> **Review:** §4: The cluster-sending protocol is important, and should be at least described, not delegated to a reference [33].

> **Revise:** We will include a full definition and provide references to effective cluster-sending protocols in the revision.

> **Review:** It is not clear how can a consensus protocol (such as the ones referenced in this paper) can be used so that "whenever a good replica learns that a decision D needs to be made, then it can force consensus on D"; this assumption should be detailed more, with the explanation in section 4 (local inputs) being insufficient.

The way in which replicas can force consensus on $D$ will depend on the internals of the consensus protocol. Next, we shall sketch an approach that works with PBFT (and most other primary-backup protocols that follow in the footsteps of PBFT).

Say, for example, that a good replica $R$ of cluster $C$ either (1) receives a client request that has not yet been processed; or (2) receives details on a client request already processed by another cluster (e.g., due to cross-shard exchange). Let $\tau$ be this request. In both cases, $R$ knows that the primary $P$ of cluster $C$ needs to initiate processing of $\tau$. To force processing of $\tau$, $R$ can forward $\tau$ to all replicas in $C$. Next, all good replicas are aware that $\tau$ needs to be processed and, hence, all good replicas can forward $\tau$ to primary $P$ with the request to initiate a round of consensus. If primary $P$ fails to do so on time, all good replicas in $C$ will consider $P$ to be faulty and initiate one-or-more view-changes until a new primary is found that is able and willing to initiate consensus on $\tau$.

> **Revise:** In the revision, we will expand on the definitions of consensus and, as mentioned before, cluster-sending to detail what building blocks we need and how they can be provided by typical consensus and cluster-sending protocols.

> **Review:** Given that the paper assumes the UTXO model, it was not clear how is integrity guaranteed. By integrity I mean that the values of the objects being destructed should be equal to the values of objects produced (if Alice has 10 tokens, she cannot send 10 to Bob and 2 to Carol).

We abstracted out and any integrity checking routines, including balance-integrity constraints. If simple UTXO transactions are used that destroy and recreate objects (with some balance attached to them) then one can enforce that all transactions maintain balance-integrity before any processing steps (and discard transactions that do not maintain balance-integrity). For more complex transactions (e.g., non-UTXO transactions), a proper integrity-constraint mechanism can be coupled with the decision to commit or abort transactions (as at that point all information regarding the transaction is available to all involved shards, due to which all replicas can make a deterministic decision as to whether balance-integrity is maintained).

We note that both OCHIMERA and RCHIMERA can be adapted to provide balance-integrity. CCHIMERA cannot, however, as balances associated with transactions in CCHIMERA that get aborted are never returned.

> **Revise:** As integrity constraints are required in many applications, we will include a sketch of how to incorporate integrity constraints in CHIMERA.

**Review:** The authors should avoid vague statements such as "If malicious primaries behave sufficiently malicious to affect the normal-case operations, their behavior is detected, and the primary is replaced". Both malicious-fault detection and consensus reconfiguration are very difficult and subtle problems (impossible under various conditions), but this paper deals with both via "view-change", which skips a lot of relevant algorithmic parts.

We agree that the wording can be improved and the reviewer is correct that fault detection is generally impossible, while PBFT-style view-changes are only possible when communication is reliable.

What we mean by our wording can be made more formal. In specific, in CCHIMERA and RCHIMERA, we rely on the Byzantine fault-tolerant primitives used to deal with any malicious behavior. In OCHIMERA, which adjusts a PBFT-style consensus to a multi-shard setting, malicious primaries can either:

- Be *non-disruptive* of the normal-case: sufficient replicas (either good or faulty) participate in the preprepare, prepare, and commit steps to assure that at-most $\mathbf{f}_S$ good replicas per shard $S$ cannot commit.

  In this case, the normal case of OCHIMERA can simply proceed to the next transaction to process. A view-change will only occur if (1) at-least a single good replica $R$ could not commit; and (2) the faulty replicas choose to support $R$ in an attempt to view-change. If such a view-change happens, then the view-change protocol guarantees that any transaction that was committed by any good replica will be preserved (Proposition 5.1).

- Be *disruptive* of the normal-case: if at-least $\mathbf{f}_S + 1$ food replicas in a shard $S$ fail to commit, then the normal-case of OCHIMERA will get disrupted and a view-change will be initiated.

Of course, besides the above cases, progress of the normal-case can also be halted due to unreliable communication. In that case, no progress will be made (no new transactions will be processed) until communication becomes reliable, after which the normal-case can recover via the view-change protocol that guarantees that any transaction that was committed by any good replica will be preserved even during intervals of unreliable communication (Proposition 5.1).

**Revise:** We will include the above clarification in Section 5.

**Review:** I'm confused by the fact that the workload includes 50% multi-shard transactions, yet the baseline (AHL) seems to degrade or at best retain the same performance. This is not what I would expect, since 50% of transaction are single-shard, if I understood correctly, and then at least these 50% should turn up partitioned across different shards, manifesting into increased throughput, or I'm misunderstanding something here.

In general, an increase in shards has two effects: (1) simple single-shard transactions can be dispersed over more shards (which would result in drastically improved performance); and (2) multi-shard transactions can become more complex to process (e.g., a transaction that affects 16 objects in an environment with 4 shards can affect at-most 4 shards, while in an environment with 16 shards it can affect at-most 16 shards).

Now consider a general sharded system without any specific bottlenecks. For multi-shard transactions that affect $n$ objects, you would expect that any number of up-to-$n$ shards will induce little or even negative scalability benefits (due to the second effect), while moving beyond $n$ shards will improve scalability. This behavior is clearly observable for the CHIMERA protocols.

As remarked by the reviewer, AHL behaves rather different. Due to the design of AHL, processing of *all* multi-shard transactions passes through a reference committee (e.g., implemented by a separate shard). Hence, this reference committee itself becomes the bottleneck as soon as the reference committee cannot process multi-shard transactions as fast as the other shards process their share of the single-shard transactions. Indeed, in AHL deployments with a high amount of multi-shard transactions (which is the case in our experiments), the reference committee will see a 100% load, while all other shards are underutilized (and could easily process many more single-shard transactions at the same time).

**Revise:** Further explain the performance that we see in the experiments.

**Review:** The evaluation in both Fig. 9 and Fig. 10 deals with common-case, i.e., clients and nodes are all "good". In order to assess scalability and resiliency differences across the 3 Chimera versions, it would be important to model & evaluate what happens under non-graceful conditions. I'm suggesting this because the paper centers on resiliency, so it would make sense to address that property.

Currently, we indeed only focus on performance differences in the normal case, and these performance differences follow directly from the differences in communication patterns of the three protocols. In primary-backup protocols, the performance impact of faulty replicas that are *not* the primary are typically minor (assuming the faulty replicas do not flood the network). The impact of faulty primaries is significant, however, as primaries can throttle performance without raising suspicion.

**Revise:** We will include a throttle-based experiment in which some of the shards have malicious primaries that throttle their performance.

**Review:** The paragraph "The choice of protocol.." (§1) would benefit a lot if it cited certain concrete examples (of prior work) that it is making reference to.

**Revise:** We will include references to recent general-purpose protocols (those mentioned in Section 8.2).

**Review:** You mention a couple of times that clients "approve" transaction and not sure if I understand what that means.

We used different terminology in the introduction (approve) and in the remainder of the paper (owner support). Thanks for pointing our attention to this inconsistency.

**Revise:** Make the usage of approve and support of a transaction by a client consistent.

**Review:** In the Abstract an Intro the paper makes it sound like it applies beyond UTXO ("..can be of use in non-UTXO environments") but this was never shown. That statement is confusing and should be clarified.

The way to do so depends on the type of data model supported and whether a strict serializable ordering of the transactions is required. For example, to support a general-purpose data model with serializable ordering in RChimera, one can replace object destruction by a (non-blocking) *lock* (in which case failure to obtain a lock would match failure to acquire input objects).

**Revise:** We will add details on how to generalize OChimera and RChimera to general-purpose workloads.

**Review #2 (reviewer Sr97)** We thank the reviewer for the detailed and long list of minor comments and suggestions to improve the writing of the paper. Next, we will only focus on the other comments.

**Review:** The differences between the variants are not very well explained. The comparison with related work and the simulations could be improved. Some parts of the paper are hard to parse.

**Review:** Abstract, "absolute minimum": This is a bit vague and strong, consider making a more concrete claim or removing this phrase.

We agree that our original description can be made more precise as to what being *minimized* mean. Both CChimera and OChimera *minimize* the number of agreement (consensus) steps each involved shard performs, as they both only perform one such step. We believe that *one* agreement step is the minimum in any sharded environment if a form of serializable transaction execution is to be provided.

**Revise:** We will include the above clarification in the revision.

**Review:** Sec. 1, par. 3, "permissioned solutions [...] lack scalability as their performance is bound by the speed of individual participants.": Unclear, please clarify and ideally give a reference

Permissioned blockchains are fully-replicated resilient systems. Hence, an upper-bound on the performance of such systems is always provided by the speed by which individual replicas can process transactions. Furthermore, adding replicas will actively decrease the upper-bound on performance, as full replication among more replicas in a resilient system increases the cost of full replication (e.g., via consensus).

**Revise:** We will add the above clarification in the revision.

**Review:** Sec. 1, last par.: According to the Abstract, O/RChimera relax the UTXO assumption. If that is correct, in the non-UTXO setting, what guarantees do these two variations provide? If that is incorrect, please rephrase the Abstract.

CChimera both requires the UTXO-like transactions and *only* guarantees successful processing of transactions that are produced in a well-behaved manner (conforming Assumption 2.3). Transactions that do not conform to Assumption 2.3 will get stuck in the system (see Theorem 4.1 and the paragraph thereafter).

Both OChimera and RChimera are able to deal with transactions that are not produced in a well-behaved manner. In specific, they both can guarantee transaction execution even if concurrent transactions require the same inputs (in that case, one or more such concurrent transactions will get aborted, after which one of the transactions can be reprocessed successfully).

Furthermore, both OChimera and RChimera can be extended to non-UTXO data models by replacing object destruction by a (non-blocking) locks (in which case failure to obtain a lock would match failure to acquire input objects).

**Revise:** We will add details on how to generalize OChimera and RChimera to general-purpose workloads.

**Review:** Sec. 2, p. 3, par. 2: The network assumption is confusing. Since you need the reliable bounded-delay communication assumption (ak.a. Delta-synchronous network assumption) for crucial parts of the protocol, the asynchronous communication assumption does not apply. E.g. you need synchrony for the correctness of the argument of the proof of Theorem 4.1 towards cross-shard consistency.

We agree that the network assumptions of Section 2 can be further clarified.

For OChimera, the network assumption apply directly, as OChimera is designed to operate in an asynchronous environment with intervals of reliable bounded-delay communication. OChimera will always guarantee consistency, but can only provide service (execute transactions) during these reliable intervals. As such, OChimera uses the network assumptions of Section 2, which are the same as most Pbft-style consensus protocols.

For CChimera and RChimera, the network assumptions are determined by the choice of the underlying consensus and cluster-sending protocols. One can provide similar guarantees as OChimera by making appropriate choices for these underlying protocols.

**Revise:** Clarify the network assumptions and how they directly influence OChimera and indirectly influence CChimera and RChimera.

**Review:** After Assumption 2.1: It should be mentioned that having only 1 owner per object makes your application-level expressivity weaker than Bitcoin, which supports multisigs, time-locks, hashlocks etc. Can shared ownership be emulated via MuSig-like (`https://popeller.io/schnorr-musig2`) key aggregation?

We assumed a unique owner (right after Assumption 2.1), as we require that owners can prove ownership and express their support for transactions in a unique and non-ambiguous way. This assumption is used in the normal-case commit path of CCHIMERA. This does not preclude shared ownership in which multiple participants own an object, however. In that case, we simply require that such a group of participants can prove ownership and express their support for transactions in a unique and non-ambiguous way. In that case the group of participants need their own agreement process to determine which transactions to support, while that support can be expressed either via multiple signatures (one per participant in the group), via threshold signatures, or via another mechanism.

**Review:** Assumption 2.2: The first part of the assumption is unclear. What does a-priori mean in this context?

Given a transaction, one can determine (1) which objects are inputs and outputs of that transaction; and (2) at which shards each of these objects should reside. One way to implement this assumption is by assigning objects to shards by their unique identifier (e.g., in a round-robin manner) or by assigning objects to shards by their digest value.

**Revise:** Clarify the intend of Assumption 2.2 and provide a sketch of how the various assumptions we rely on can be provided in practice.

**Review:** Assumption 2.3: There are Bitcoin protocols (e.g. Lightning Network) in which an honest owner has to sign multiple conflicting txs. Possibly this is a non-issue, since such protocols typically also require shared ownership, which is precluded in your model.

CCHIMERA is not designed to be able to process multiple transactions that use the same inputs (see Theorem 4.1). Hence, CCHIMERA only guarantees successful execution of transactions that satisfy Assumption 2.3. In the case multiple transactions have the same inputs, CCHIMERA will be able to process *one* of these transactions *if* all shards end up processing the same transaction first. In that case, CCHIMERA does not guarantee successfull execution, however. Both OCHIMERA and RCHIMERA do not require Assumption 2.3, as they both can abort and rerun transactions when there is a conflict.

**Revise:** Expand on the need for Assumption 2.3 in CCHIMERA.

**Review:** Sec. 4: Parties process only one transaction per round instead of aggregating them in atomic blocks. This design choice should be justified.

Both CCHIMERA and RCHIMERA can easily process transactions in blocks: at a per-shard level, consensus decisions are made independently of other shards. Hence, instead of one transaction per consensus decision, both can include a batch of transactions in their consensus decisions (after which they perform the steps related to the transactions in that batch in order).

For OCHIMERA, blocks of transactions are more challenging as OCHIMERA uses a single multi-shard consensus step that includes all replicas of all shards affected by a transaction. Still, OCHIMERA can be generalized to process blocks of transactions that affect the same set of shards. Without further precautions, such a setup can lead to several shards proposing distinct blocks that include the same transaction, however. It is possible to deal with such issues with further techniques (e.g., by either assuring that only a single shard can construct blocks that contain a given transaction based on the digest of that transaction). To keep presentation simple, we have chosen to present only the simple case in which each round processes a single transaction.

**Revise:** Include a remark that sketches how the three protocols can be generalized to atomic blocks of transactions.

**Review:** Sec. 4, p. 4, col. 2, par. 3: What happens if a transaction looks valid to shard $S$ but invalid to shard $S'$? It seems that S will wait forever for a $m(S', \tau)$ message that will never come, as $S'$ sends nothing in that case. There should be some form of timeout, after which a "processed"(#1) transaction is dropped.

Assuming that a transaction $\tau$ is syntactically correct (which can be determined by any replica independently), the *validity* of $\tau$ is determined by checking whether all owners of inputs of $\tau$ support that transaction. Typically, ownership is expressed via digital signatures, which can be verified deterministically by any replica independently. Hence, all replicas in all affected shards will make the same conclusion on whether $\tau$ is valid.

**Revise:** At the end of Section 3, we have provided details on local requirements and global requirements. We will extend this part by including details on how validity is a local requirement (that can be checked deterministically by replicas independently of other replicas).

**Review:** Sec. 4, p. 5, bullet 2: The first-pledge rule does not specify how to resolve disagreements between parties on which tx is the first.

Sec. 4, p. 5, bullet 2: The two rules are not discussed in the rest of the paper. How are they guaranteed to hold?

Consider two shards $S_1$ and $S_2$ affected by transactions $\tau_1$ and $\tau_2$ that each require objects $O_1$ and $O_2$ residing on shards $S_1$ and $S_2$, respectively. Now consider the case in which shard $S_1$ first processes $\tau_1$ and then $\tau_2$, while shard $S_2$ first processes $\tau_2$ and then $\tau_1$. In this case, shard $S_1$ will pledge $O_1$ to $\tau_1$ and shard $S_2$ will pledge $O_2$ to $\tau_2$. Hence, both $\tau_1$ and $\tau_2$ miss inputs and will fail to complete execution. As both transactions will abort, the order in which they abort does not matter.

Note that the above situation, in which we always have an abort, will only happen if two transactions have the same inputs. If a transaction has unique inputs, it will always be able to commit (without interference of other replicas).

The above rules do not need to be guaranteed, as they specify deterministic behavior for all good replicas.

**Review:** (#1) Secs. 3 & 4: The terminology "process" vs "execute" is very unclear. Normally these two terms are interchangeable. I think that one can see a tx but decide not to process it, or process it but then not execute it, or execute it but then not commit it. In detail, my understanding on the lifecycle of a transaction is: a party learns about it, validates it and ensures it belongs to its shard. It then processes it, i.e. it tries to achieve consensus with its other shard members on the message $m(S, \tau)$. Once this is achieved, the party waits for the cross-shard exchange to complete and then it executes $\tau$, i.e. it updates its local state by destructing Inputs($\tau$) and constructing Outputs($\tau$) of its shard. Lastly it commits to these changes by achieving consensus on the execution of tau with its other shard members. Only the last step, commitment, respects round ordering. Whether I'm right or not, this workflow should be clearly described with all relevant terms introduced and explicitly defined (not necessarily in delineated definitions, but clearly enough). Figure 2 should also be expanded to show all these phases with the respective names. Lastly, if my understanding of the flow is correct, I propose renaming "processing" to "intra-shard processing" and "executing" to "inter-shard processing" or equivalent.

The workflow outlined by the reviewer is correct and we agree that the paper can benefit from clearly-defined terms for processing and execution of transactions.

**Revise:** Include formal definitions of transaction *execution* (all involved replicas reaching the commit or abort state of that transaction) and *processing* (validation and execution), and related terms. Clarify which steps of the protocols correspond to the above terms (the workflow).

**Review:** (#2) Sec. 4, "cross-shard exchange": since it doesn't completely trust other shards (it verifieas that $I(S', \tau) = D(S', \tau)$ after all) and in order to ensure that no inputs are unmatched, shard $S$ should also verify that each input of $\tau$ appears in exactly one $m(S', \tau)$ message.

Proof of Theorem 4.1, par. 1: If I understand correctly, without the verification of (#2), shard-applicability does not hold.

The check $I(S', \tau) = D(S', \tau)$ verifies that all necessary inputs requested by the transaction are available (necessary, as an object can be used only once). As each object $O$ belong to a single shard $S$, only the message of that shard $S$ can contain $O$ and all other messages would not be well-formed.

Assuming that the cross-shard exchange step is performed with a cluster-sending step that *only* allows well-formed messages, which would be the only messages agreed-upon by all good replicas in a shard, to be exchanged by that shard, we do not further need to consider whether messages are well-formed after a successful exchange.

We note that this reliance on the correctness of the cross-shard exchange step is necessary: otherwise, if faulty replicas would be able to successfully exchange messages to other shards without agreement of good replicas, they would even be able to exchange well-formed messages that *lie* about which objects are available and which are not.

**Revise:** In response to reviewer #1, we will already include a full definition of cluster-sending. Furthermore, we will clarify that we require the guarantees of cluster-sending for assuring that well-formed (and truthful) messages are exchanged during the cross-shard exchange step.

**Review:** Sec. 4, last par.: This optimisation severely limits the usability of the blockchain. As mentioned earlier, it precludes protocols in which honest parties sign various alternative transactions that spend the same output. It additionally precludes techniques like Bitcoin's Replace-by-fee. Also, since agreement with other shards is anyway needed to securely construct the output objects, the bottleneck of communication still exists. Therefore the benefit of the optimization is unclear. Given the drawbacks, it must be argued that this approach is supported by the strong experimental efficiency improvements.

The reviewer is correct that CCHIMERA is limited to a subset of all possible use-cases, as it puts restrictions on how owners can behave. The resultant benefit is that transaction processing can be handled by only a single round of per-shard consensus. Due to this, CCHIMERA has a lower cost (and higher performance) than other protocols. Due to the usage of logarithmic scales, the performance differences between the CHIMERA protocols are not that clear, however.

**Revise:** Rework the presentation of the experimental results to further underline the performance differences between the CHIMERA protocols.

**Review:** Sec. 4: Communication complexity is entirely missing. Is it quadradic in the number of parties of the biggest shard? Do other factors (number of shards, total number of parties, number of inputs/outputs, size of blockchain) influence it? This possibly needs its own subsection.

Sec. 8: There is extensive discussion on the number of phases and steps of each protocol, but the efficiency of each such step is not extensively discussed. If each Chimera step is very expensive, the total latency could be worse than other protocols with more steps.

The exact complexity will depend on the choice of protocols that provide the consensus and cluster-sending primitives. We have already provided a breakdown of the complexity in terms of these primitives (Figure 8). CCHIMERA indeed performs the cross-shard exchange step as an all-to-all communication step, as this is the fastest way to perform this step (in terms of latency).

At the cost of latency, one can change this step into an all-to-one-to-all communication step. Such a change would only be of real benefit if typical transactions each affect many shards. Hence, we have not included details on such a protocol.

**Revise:** We expand on the analysis of Figure 8 by including the full message complexity for a reasonable choice of consensus and cluster-sending primitives. For PBFT (consensus), the cost of the consensus step (per shard) will be quadratic in the number of replicas in that shard. The size of transactions will play a role in both the consensus step and the cross-shard exchange step. In typical cluster-sending protocols, the complexity of an exchange between shards depends on the number of replicas in each shard.

**Review:** Sec. 5, p. 6, Modified PBFT: How does this protocol differ from PBFT between all parties of shards(tau)?

The main difference is that PBFT has a single primary, whereas OCHIMERA has a single primary *per* shard. This difference affects the capability of individual replicas to detect the root cause of failures of the normal-case operations (as several primaries could be the root cause).

**Review:** Sec. 5, p. 7, second-last par. & Fig. 5, line 10: What if an honest party receives $f_S + 1$ or more VCRecoveryRQ messages but it can respond with a valid VCLocalSCR message? It seems that this case would be problematic, as it would trigger an unneeded local state recovery. Is there something preventing this?

This would trigger a local state recovery, as in normal situations this case only happens if a primary is malicious (for which replacement is always a good thing, even if we can fix up the issues with a `VSLocalSCR` message.

**Review:** Sec. 5. Fig. 4, line 1: How much time does $R$ wait before concluding this event happened?

Sec. 5. Fig. 4 & 5: This code seems to assume that the preprepare phase has completed successfully, because in the case of need for local recovery in Fig. 4, Fig. 5 assumes that there is some M for round $\rho$. Discuss what happens when no such $M$ exists.

Sec. 5, Prop. 5.1: "every successful future view of $S$ will recover $M$" undefined

Sec. 5, Prop. 5.1: $M$ refers only to a single tx (c.f. def. in "Global preprepare"). How can a replica "recover" $M$ in every future view, even in views that refer to future transactions? I think you need to specify that you refer to future views of the same round.

For brevity, we focused only on the case of a single transaction (to be recovered) in the case of a multi-shard consensus, as the recovery of multi-shard transactions is the part where OCHIMERA significantly differs from PBFT. Based on the detailed feedback of this reviewer, we believe that our description became too brief.

**Revise:** We will revise the presentation of the view-change algorithms to include a full treatment of all necessary details and on how to adjust the mechanisms described to deal with the recovery of any number of transactions.

**Review:** Sec. 5, Thm. 5.2 proof: The proof is persuasive in the accept & commit case, but the argument is not clear in the abort case.

An unfortunate choice of terminology does not help interpret Theorem 5.2, as the term *commit* has two distinct and unrelated roles: namely the commit phase of the consensus step, and the commit decision on transaction execution. Any transaction (whether its execution results in a commit or an abort) is processed by consensus step in the same way, the difference only being on how replicas execute the transaction after consensus is reached (on when to execute and on what the inputs are).

**Revise:** We will disambiguate the terminology used in Theorem 5.2

**Review:** Subsec. 8.1: It would be better to add clearly distinct use-cases for each of the three protocol variations. Especially the difference of Core and Resilient Chimera elude me.

Due to other comments by the reviewer, we will already further clarify the limitations of CCHIMERA (it provides no guarantees when dealing with conflicting transactions). Due to the limitations of CCHIMERA, it can outperform the other protocols, which we shall further underline with the extended experiments.

**Review:** Subsec. 8.3, par. 2: Why did you choose only AHL for comparison? Justify.

We originally opted not to include CHAINSPACE, as it will roughly perform the same as RCHIMERA (just with slightly higher latencies). Due to the linear design of RINGBFT, which is completely optimized for reducing bandwidth usage by eliminating any all-to-all cross-shard exchange steps between shards, it has the potential to outperform the CHIMERA protocols in terms of transaction throughput. This comes at the cost of exceptionally high latencies, however.

The design of SHARPER excludes running the massively multi-shard workloads we envision, as it places shard-wide locks on shards while processing any multi-shard transactions. Hence, data on SHARPER with the workloads we use will not yield useful results. Finally, BYSHARD is a framework in which many protocols can be modeled and specific variants of BYSHARD match the behavior of AHL, CHAINSPACE, and RINGBFT.

The breakdown in Figure 10 provides the number of shards affected by each of the entire workloads (given the object distribution used in the experiments). Hence, this number is independent of the protocol used, and will impact the cost of all CHIMERA protocols in the same way.

**Revise:** Further clarify what is exactly shown in Figure 10, how it relates to the workloads of Figure 9, how this figure correlates to the CHIMERA protocols, and provide a table with workload statistics for each experimental case (number of replicas/number of objects per transaction).

**Review:** Subsec. 8.3: The simulations assume a very large number of very small shards. Under such circumstances, the per-shard honest supermajority assumption is unreasonable. Please add simulations with a much larger number of parties per shard – a smaller total number of shards is OK.

This is a great suggestion and will further underline the differences between the CHIMERA variants, as larger clusters sizes will increase the impact of consensus (as part of the overall cost of transaction processing).

**Revise:** Rework the experiments to include larger clusters.

**Review:** Subsec. 8.3, "A closer look at the data does reveal excellent scalability of AHL": Are you referring to the experiments of [17]?

Both [17] and [34] and our own results show that AHL has excellent scalability in some environments: namely when workloads consist mainly of single-shard transactions. Due to the design of AHL, processing of *all* multi-shard transactions passes through a reference committee (e.g., implemented by a separate shard), however. Hence, this reference committee itself becomes the bottleneck as soon as the reference committee cannot process multi-shard transactions as fast as the other shards process their share of the single-shard transactions. Indeed, in AHL deployments with a high amount of multi-shard transactions (which is the case in our experiments), the reference committee will see a 100% load, while all other shards are underutilized (and could easily process many more single-shard transactions at the same time).

**Revise:** Further clarify the behavior of AHL in the experiments.

## Review #3 (reviewer TgEp)

**Review:** 1-In the paper, it says that, "Due to this, CCHIMERA will not undo any pledges of objects to the execution of any transactions." So let's say I submit a transaction with inputs I1, I2 where I1 resides in S1, and where I2 is an invalid input, doesn't exist in any shard.=S1 will locally run consensus, and broadcast $m(I1, S1)$. However, it is not going to receive $m(I2, S2)$, and will abort. What does this abort do exactly? If I later submit tx <I1, O1> to send only I1, am I not able to do so? If so, what exactly is the benefit of this? Or am I understanding this incorrectly?

In the situation sketched by the reviewer, the original transaction $\tau$ that uses both $I_1$ and $I_2$ must be a well-formed transaction and valid (the owners of $I_1$ and $I_2$ expressed their support for the transaction), as otherwise the transaction got discarded without further processing.

Hence, to arrive at the situation sketched, the object $I_2$ must already been used by some transaction and, hence, no longer exist. In this case, shard $S_1$ will pledge $I_1$ to $\tau$ and never release this pledge. Hence, the owner of $I_1$ (who signed off on $\tau$) will loose access to $I_1$ and no future transactions can process $I_1$ (Theorem 4.1).

The main benefit of this drastic consequence of trying to use objects in several transactions is protocol simplicity: as well-behaved owners are required to only try to perform a single transaction with a particular object as an input, a shard $S$ that is affected by transaction $\tau$ can permanently pledge any input objects at $S$ required by $\tau$ to $\tau$ *before* shard $S$ can even verify whether the other shards participating in $\tau$ can make all required pledges for $\tau$. Indeed, in CCHIMERA shards decide to commit $\tau$ before *any* coordination between shards happens, and only use coordination to figure out the end result of that commit decision (if they wrongfully decided to commit, execution of $\tau$ gets stuck and objects used by $\tau$ are lost).

Both OCHIMERA and RCHIMERA allow for a more general-purpose environment in which the system can *abort* transactions if it turns out that not all their inputs are available. This comes at the cost of more coordination between shards (and, hence, less throughput), however.

**Revise:** Further clarify the strengths and limitations of CCHIMERA at the end of Section 4.

**Review:** 2- There are some cross-shard transactions that are not considered in paper, but introduced in sharded blockchain papers such as: https://eprint.iacr.org/2017/406.pdf, https://eprint.iacr.org/2018/460.pdf, https://eprint.iacr.org/2022/413.pdf. A brief compare/contrast with these methods would be helpful. Especially the third papers cross-shard method also seem to be particularly suited to UTXO model.

We are familiar with OMNILEDGER and RAPIDCHAIN and both use rather different approaches to sharding than the CHIMERA protocols and the other protocols we compare with. We can outline the difference between our approach and theirs in the related work section, however. As the third paper is rather recent, we are not yet familiar with the details of that paper, and we have to study it in a bit more depth before we can conclusively outline their approach.

**Revise:** Review these three approaches and include relevant details in the related work.

**Review:** 3- Is there a particular reason to only include AHL in the experimental analysis as the baseline? Could it be expanded to some other cross-shard methods mentioned in the paper?

As stated in response to the previous reviewer, we originally opted not to include Chainspace, as it will roughly perform the same as RChimera (just with slightly higher latencies). Due to the linear design of RingBFT, which is completely optimized for reducing bandwidth usage by eliminating any all-to-all cross-shard exchange steps between shards, it has the potential to outperform the Chimera protocols in terms of transaction throughput. This comes at the cost of exceptionally high latencies, however.

The design of SharPer excludes running the massively multi-shard workloads we envision, as it places shard-wide locks on shards while processing any multi-shard transactions. Hence, data on SharPer with the workloads we use will not yield useful results. Finally, ByShard is a framework in which many protocols can be modeled and specific variants of ByShard match the behavior of AHL, Chainspace, and RingBFT.

**Revise:** We extend the experiments to include Chainspace and RingBFT. To make such experiments insightful, we will also include simulations to determine minimum processing latencies; this to show the differences in design approach between, on the one hand RingBFT and, on the other hand, the Chimera protocols.

# PROBLEM: SHARDED RESILIENT TRANSACTION PROCESSING WITH MINIMAL COSTS

**Anonymous authors**

## Abstract

To enable scalable resilient blockchain systems, several powerful general-purpose approaches toward sharding such systems have been demonstrated. Unfortunately, these approaches all come with substantial costs for ordering and execution of multi-shard transactions.

In this work, we ask whether one can achieve significant cost reductions for processing multi-shard transactions by limiting the type of workloads supported. To initiate the study of this problem, we propose core-CHIMERA (CCHIMERA). CCHIMERA uses *strict UTXO-based environmental requirements* to enable powerful multi-shard transaction processing with an absolute minimum amount of coordination between shards. In the environment we designed CCHIMERA for, CCHIMERA will operate *perfectly* with respect to all transactions proposed and approved by well-behaved clients, but does not provide any other guarantees.

To illustrate that CCHIMERA-like protocols can be of use in non-UTXO environments, we also demonstrate *two* generalizations of CCHIMERA, *optimistic*-CHIMERA and *resilient*-CHIMERA, that make different tradeoffs in complexity and costs when dealing with faulty behavior and attacks. Finally, we compare these three protocols and show their potential scalability and performance benefits over state-of-the-art general-purpose systems. These results underline the importance of the study of specialized approaches toward sharding in resilient systems.

## 1 Introduction

The advent of blockchain applications and technology has rejuvenated interest of companies, governments, and developers in resilient distributed fully-replicated systems and the distributed ledger technology (DLT) that powers them. Indeed, in the last decade we have seen a surge of interest in reimagining systems and build them using DLT networks. Examples can be found in the financial and banking sector [15, 38, 50], IoT [43], health care [28, 39], supply chain tracking, advertising, and in databases [5,23,30,31,47,48]. This wide interest is easily explained, as blokchains promise to improve resilience against both failures and malicious behavior, while enabling the federated management of data by many participants.

To illustrate this, we look at the financial sector. Current traditional banking infrastructure is often rigid, slow, and creates substantial frictional costs. It is estimated that the yearly cost of transactional friction alone is $71 billion [8] in the financial sector, creating a strong desire for alternatives. This sector is a perfect match for DLT, as it enables systems that manage digital assets and financial transactions in more flexible, fast, and open federated infrastructures that eliminate the friction caused by individual private databases maintained by banks and financial services providers. Consequently, it is expected that a large part of the financial sector will move towards DLT [18].

At the core of DLT is the *replicated state* maintained by the network in the form of a ledger of transactions. In traditional blockchains, this ledger is fully replicated among all participants using consensus protocols [14, 30, 36, 43, 46]. For many practical use-cases, one can choose to use either permissionless consensus solutions that are operated via economic self-incentivization through cryptocurrencies (e.g., Nakamoto consensus [45, 55]), or permissioned consensus solutions that require vetted participation (e.g, PBFT, POE, and HOT-STUFF [16, 32, 57]). Unfortunately, the design of consensus protocols are severely limited in their ability to provide the *high transaction throughput* that is needed to address practical needs, e.g., in the financial sector. Indeed, on the one hand, we see that permissionless solutions can easily scale to thousands of participants, but are severely limited in their transaction processing throughput. E.g., in Ethereum, a popular public permissionless DLT platform, the rapid growth of decentralized finance applications [12] causes its network fees to rise precipitously as participants bid for limited network capacity [7], while Bitcoin can only process a few transactions per second [50]. On the other hand, permissioned solutions can reach much higher throughputs, but still lack scalability as their performance is bound by the speed of individual participants.

Recently, several general-purpose sharded consensus-based systems have been proposed to combat the limitations of fully-replicated consensus-based systems [1, 3, 4, 17, 34, 51]. In these systems, one partitions the data among several *shards* that each can potentially operate mostly-independent on their data, while only requiring inter-shard coordination to process multi-shard transactions that affect data on several shards (see Figure 1).

The choice of protocol for such multi-shard transaction processing determines greatly the scalability benefits of sharding and the overhead costs incurred by sharding, however. In prac-

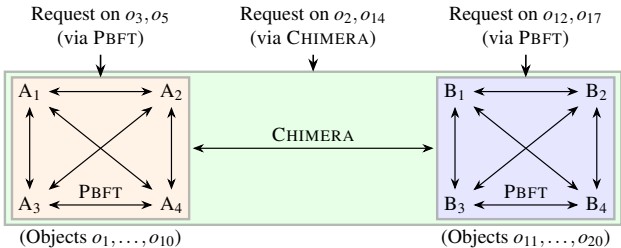

Figure 1: A *sharded* design in which two resilient blockchains each hold only a part of the data. Local decisions within a cluster are made via *traditional* PBFT *consensus*, whereas multi-shard transactions are processed via CHIMERA (proposed in this work).

tice, existing proposals for sharding consensus-based systems have taken a general-purpose approach aiming at serving any workload. Unfortunately, such genericity comes at a cost, and existing proposals either have high coordination costs, have high latencies, or have severe bottlenecks with multi-shard workloads.

In this work, we ask whether one can improve on the state-of-the-art proposals by *limiting* the type of workloads supported by the systems. In specific, we propose the following problem for further study:

> **Problem.** Can one *reduce* the cost of coordination in the design of sharded consensus-based systems by limiting the types of workloads supported?

In this paper, we give a preliminary *positive* answer for the above problem. In specific, we put forward the CHIMERA family of multi-shard transaction processing protocols that can process UTXO-transactions and uses properties of these transactions to minimize coordination. To be able to adapt to the needs of specific use-cases, we propose three variants of CHIMERA:

1. In Section 4, we propose Core-CHIMERA (CCHIMERA), a design specialized for processing *UTXO-like transactions*. CCHIMERA uses strict environmental assumptions on UTXO-transactions to its advantage to yield a *minimalistic* design that only requires *local consensus* in affected shards and does not require any coordination between shards beyond a single round of information sharing, a minimal amount. Even with this minimalistic design, CCHIMERA will operate *perfectly* with respect to all transactions proposed and approved by well-behaved clients (although it may fail to process transactions originating from malicious clients).

To also support more general-purpose environments, we propose Optimistic-CHIMERA and Resilient-CHIMERA, *two* generalizations of CCHIMERA that each deal with the strict envi-

ronmental assumptions of CCHIMERA while preserving the minimalistic design of CCHIMERA:

2. In Section 5, we propose Optimistic-CHIMERA. In the design of Optimistic-CHIMERA (OCHIMERA), we assume that malicious behavior is rare and we optimize the normal-case operations. We do so by keeping the normal-case operations as minimalistic as possible by utilizing a *single* multi-shard consensus step to execute multi-shard transactions in the normal case. When compared to CCHIMERA, this step does not require any additional coordination phases in the well-behaved optimistic case, while still being able to lift the environmental assumptions of CCHIMERA. In doing so, OCHIMERA does require intricate coordination when recovering from attacks, however.

3. In Section 6, we propose Resilient-CHIMERA. In the design of Resilient-CHIMERA, we assume that malicious behavior is common and we add sufficient coordination to the normal-case operations of CCHIMERA to enable a simpler and localized recovery path, allowing RCHIMERA to operate in a general-purpose fault-tolerant environments without significant costs to recover from attacks.

In Section 7, we show that all three variants of CHIMERA provide strong ordering guarantees based on their usage of UTXO-transactions. Finally, in Section 8, we compare the three CHIMERA protocols and show their potential scalability and performance benefits over state-of-the-art general-purpose systems

## 2 Preliminaries

As permissioned blockchains already have much higher throughputs than permissionless blockchains, we will focus on permissioned blockchains in this paper.

First, we introduce the system model, the sharding model, the data model, the transaction model, and the terminology and notation used throughout this paper.

If $S$ is a set of replicas, then $\mathcal{G}(S)$ denotes the non-faulty *good replicas* in $S$ that always operate as intended, and $\mathcal{F}(S) = S \setminus \mathcal{G}(S)$ denotes the remaining replicas in $S$ that are *faulty* and can act *Byzantine*, deviate from the intended operations, or even operate in coordinated malicious manners. We write $\mathbf{n}_S = |S|$, $\mathbf{g}_S = |\mathcal{G}(S)|$, and $\mathbf{f}_S = |S \setminus \mathcal{G}(S)| = \mathbf{n}_S - \mathbf{g}_S$ to denote the number of replicas in $S$, good replicas in $S$, and faulty replicas in $S$, respectively.

Let $\mathfrak{R}$ be a set of replicas. In a *sharded fault-tolerant system* over $\mathfrak{R}$, the replicas are partitioned into sets $\text{shards}(\mathfrak{R}) = \{\mathcal{S}_0, \ldots, \mathcal{S}_\mathbf{z}\}$ such that the replicas in $\mathcal{S}_i$, $0 \leq i \leq \mathbf{z}$, operate as an independent Byzantine fault-tolerant system. As each $\mathcal{S}_i$ operates as an independent Byzantine fault-tolerant system,

we require $\mathbf{n}_{\mathcal{S}_i} > 3\mathbf{f}_{\mathcal{S}_i}$, a minimal requirement to enable Byzantine fault-tolerance in an asynchronous environment [20, 21]. We assume that every shard $\mathcal{S} \in \texttt{shards}(\mathfrak{R})$ has a unique identifier $\texttt{id}(\mathcal{S})$.

We assume *asynchronous communication*: messages can get lost, arrive with arbitrary delays, and in arbitrary order. Consequently, it is impossible to distinguish between, on the one hand, a replica that is malicious and does not send out messages, and, on the other hand, a replica that does send out proposals that get lost in the network. As such, CHIMERA can only provide *progress* in periods of *reliable bounded-delay communication* during which all messages sent by good replicas will arrive at their destination within some maximum delay [25, 27]. Further, we assume that communication is *authenticated*: on receipt of a message $m$ from replica $\texttt{R} \in \mathfrak{R}$, one can determine that $\texttt{R}$ did sent $m$ if $\texttt{R} \in \mathcal{G}(\mathfrak{R})$. Hence, faulty replicas are able to impersonate each other, but are not able to impersonate good replicas. To provide authenticated communication under practical assumptions, we can rely on cryptographic primitives such as digital signatures and threshold signatures [40, 52].

**Assumption 2.1.** We assume *coordinating adversaries* that can, at will, choose and control any replica $\texttt{R} \in \mathcal{S}$ in any shard $\mathcal{S} \in \texttt{shards}(\mathfrak{R})$ in the sharded fault-tolerant system as long as, for each shard $\mathcal{S}' \in \texttt{shards}(\mathfrak{R})$, the adversaries only control up to $\mathbf{f}_{\mathcal{S}'}$ replicas in $\mathcal{S}'$.

We use the *object-dataset model* in which data is modeled as a collection of *objects*. Each object $o$ has a unique *identifier* $\texttt{id}(o)$ and a unique *owner* $\texttt{owner}(o)$. In the following, we assume that all owners are *clients* of the system that manages these objects. The only operations that one can perform on an object are *construction* and *destruction*. An object cannot be recreated, as the attempted recreation of an object $o$ will result in a new object $o'$ with a distinct identifier ($\texttt{id}(o) \neq \texttt{id}(o')$).

Changes to object-dataset data are made via transactions requested by clients. We write $\langle\tau\rangle_c$ to denote a transaction $\tau$ requested by a client $c$. We assume that all transactions are *UTXO-like transactions*: a transaction $\tau$ first produces resources by destructing a set of *input objects* and then consumes these resources in the construction of a set of *output objects*. We do not rely on the exact rules regarding the production and consumption of resources, as they are highly application-specific. Given a transaction $\tau$, we write $\texttt{Inputs}(\tau)$ and $\texttt{Outputs}(\tau)$ to denote the input objects and output objects of $\tau$, respectively, and we write $\texttt{Objects}(\tau) = \texttt{Inputs}(\tau) \cup \texttt{Outputs}(\tau)$.

**Assumption 2.2.** Given a transaction $\tau$, we assume that one can determine $\texttt{Inputs}(\tau)$ and $\texttt{Outputs}(\tau)$ a-priori. Furthermore, we assume that every transaction has inputs. Hence, $|\texttt{Inputs}(\tau)| \geq 1$.

Owners of objects $o$ can *express their support* for transactions $\tau$ that have $o$ as their input. To provide this functionality, we can rely on digital signatures [40].

**Assumption 2.3.** If an owner is well-behaved, then an expression of support cannot be forged or provided by any other party. Furthermore, a well-behaved owner of $o$ will only express its support for *a single* transaction $\tau$ with $o \in \texttt{Inputs}(\tau)$, as only one transaction can consume the object $o$, and the owner will only do so after the construction of $o$.

Let $o$ be an object. We assume that there is a well-defined function $\texttt{shard}(o)$ that maps object $o$ to the single shard $\mathcal{S} \in \texttt{shards}(\mathfrak{R})$ that is responsible for maintaining $o$. Given a transaction $\tau$, we write $\texttt{shards}(\tau) = \{\texttt{shard}(o) \mid o \in \texttt{Objects}(\tau)\}$ to denote the shards that are affected by $\tau$. We say that $\tau$ is a *single-shard transaction* if $|\texttt{shards}(\tau)| = 1$ and is a *multi-shard transaction* otherwise.

# 3 Multi-Shard Transaction Processing

Before we introduce CHIMERA, we put forward the correctness requirements we want to maintain in a multi-shard transaction system in which each shard is itself a set of replicas operated as a Byzantine fault-tolerant system. We say that a shard $\mathcal{S}$ performs an action if every good replica in $\mathcal{G}(\mathcal{S})$ performs that action. Hence, any processing decision or execution step performed by $\mathcal{S}$ requires the usage of a *consensus protocol* [14, 16, 30, 42, 43] that coordinates the operations of individual replicas in the system, e.g., a Byzantine fault-tolerant system driven by PBFT [16], POE [32], or HOTSTUFF [57], or a crash fault-tolerant system driven by PAXOS [42]. As these systems are fully-replicated, each replica holds exactly the same data, which is determined by the *sequence of transactions*—the journal—agreed upon via consensus:

**Definition 3.1.** A *consensus protocol* coordinate decision making among the replicas of a resilient cluster $\mathcal{S}$ by providing a reliable ordered replication of *decisions*. To do so, consensus protocols provide the following guarantees:

1. If good replica $\texttt{R} \in \mathcal{S}$ makes a $\rho$-th decision, then all good replicas $\texttt{R}' \in \mathcal{S}$ will make a $\rho$-th decision (whenever communication becomes reliable).

2. If good replicas $\texttt{R}, \texttt{Q} \in \mathcal{S}$ make $\rho$-th decisions, then they make the same decisions.

3. Whenever a good replica learns that a decision $D$ needs to be made, then it can force consensus on $D$.

Let $\tau$ be a transaction processed by a sharded fault-tolerant system. Processing of $\tau$ does not imply execution: the transaction could be invalid (e.g., the owners of affected objects did not express their support) or the transaction could have inputs that no longer exists. We say that the system *commits* to $\tau$ if it decides to apply the modifications prescribed by $\tau$, and we say that the system *aborts* $\tau$ if it decides to not do so. Using this terminology, we put forward the following requirements for any sharded fault-tolerant system:

R1 *Validity*. The system must only processes transaction $\tau$ if, for every input object $o \in \text{Inputs}(\tau)$ with a well-behaved owner $\text{owner}(o)$, the owner $\text{owner}(o)$ supports the transaction.

R2 *Shard-involvement*. The shard $S$ only processes transaction $\tau$ if $S \in \text{shards}(\tau)$.

R3 *Shard-applicability*. Let $D(S)$ be the dataset maintained by shard $S$ at time $t$. The shards $\text{shards}(\tau)$ only commit to execution of transaction $\tau$ at $t$ if $\tau$ consumes only existing objects. Hence, $\text{Inputs}(\tau) \subseteq \bigcup \{D(S) \mid S \in \text{shards}(\tau)\}$.

R4 *Cross-shard-consistency*. If shard $S$ commits (aborts) transaction $\tau$, then all shards $S' \in \text{shards}(\tau)$ eventually commit (abort) $\tau$.

R5 *Service*. If client $c$ is well-behaved and wants to request a valid transaction $\tau$, then the sharded system will eventually *process* $\langle \tau \rangle_c$. If $\tau$ is shard-applicable, then the sharded system will eventually *execute* $\langle \tau \rangle_c$.

R6 *Confirmation*. If the system processes $\langle \tau \rangle_c$ and $c$ is well-behaved, then $c$ will eventually learn whether $\tau$ is committed or aborted.

We notice that shard-involvement is a *local requirement*, as individual shards can determine whether they need to process a given transaction. In the same sense, shard-applicability and cross-shard-consistency are *global* requirements, as assuring these requirements requires coordination between the shards affected by a transaction.

## 4 Core-CHIMERA: Simple Yet Efficient Transaction Processing

The core idea of CHIMERA is to minimize the coordination necessary for multi-shard ordering and execution of transactions. To do so, CHIMERA combines the semantics of transactions in the object-dataset model with the minimal coordination required to assure shard-applicability and cross-shard consistency. This combination results in the following high-level three-step approach towards processing any transaction $\tau$:

1. *Local inputs*. First, every affected shard $S \in \text{shards}(\tau)$ locally determines whether it has all inputs from $S$ that are necessary to process $\tau$.

2. *Cross-shard exchange*. Then, every affected shard $S$ exchanges these inputs to all other shards in $\text{shards}(\tau)$, thereby pledging to use their local inputs in the execution of $\tau$.

3. *Decide outcome*. Finally, every affected shard $S$ decides to commit $\tau$ if all affected shards were able to provide all local inputs necessary for execution of $\tau$.

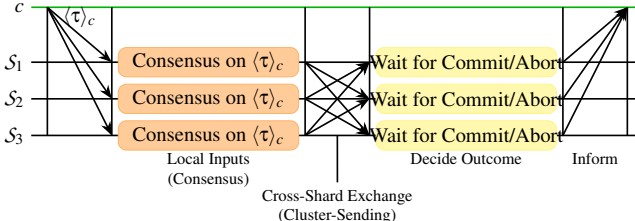

Figure 2: The message flow of CCHIMERA for a 3-shard client request $\langle \tau \rangle_c$ that is committed.

Next, we describe how these three high-level steps are incorporated by CHIMERA into normal consensus steps at each shards. Let shard $S \in \text{shards}(\mathfrak{R})$ receive client request $\langle \tau \rangle_c$. The good replicas in $S$ will first determine whether $\tau$ is valid and applicable. If $\tau$ is not valid or $S \notin \text{shards}(\tau)$, then the good replicas discard $\tau$. Otherwise, if $\tau$ is valid and $S \in \text{shards}(\tau)$, then the good replicas utilize *consensus* to force the primary $\mathcal{P}(S)$ to propose in some consensus round $\rho$ the message $m(S, \tau)_\rho = (\langle \tau \rangle_c, I(S, \tau), D(S, \tau))$, in which $I(S, \tau) = \{o \in \text{Inputs}(\tau) \mid S = \text{shard}(o)\}$ is the set of objects maintained by $S$ that are input to $\tau$ and $D(S, \tau) \subseteq I(S, \tau)$ is the set of currently-available inputs at $S$. Only if $I(S, \tau) = D(S, \tau)$ will $S$ pledge to use the local inputs $I(S, \tau)$ in the execution of $\tau$.

The acceptance of $m(S, \tau)_\rho$ in round $\rho$ by all good replicas completes the *local inputs* step. Next, during execution of $\tau$, the *cross-shard exchange* and *decide outcome* steps are performed. First, the *cross-shard exchange* step. In this step, $S$ broadcasts $m(S, \tau)_\rho$ to all other shards in $\text{shards}(\tau)$. To assure that the broadcast arrives, we rely on a reliable primitive for *cross-shard exchange*, e.g., via an efficient cluster-sending protocol [33]. Then, the replicas in $S$ wait until they receive messages $m(S', \tau)_{\rho'} = (\langle \tau \rangle_c, I(S', \tau), D(S', \tau))$ from all other shards $S' \in \text{shards}(\tau)$.

After cross-shard exchange comes the final *decide outcome* step. After $S$ receives $m(S', \tau)_{\rho'}$ from all shards $S' \in \text{shards}(\tau)$, it decides to *commit* whenever $I(S', \tau) = D(S', \tau)$ for all $S' \in \text{shards}(\tau)$. Otherwise, it decides *abort*. If $S$ decides commit, then all good replicas in $S$ destruct all objects in $D(S, \tau)$ and construct all objects $o \in \text{Outputs}(\tau)$ with $S = \text{shard}(o)$. Finally, each good replica informs $c$ of the outcome of execution. If $c$ receives, from every shard $S'' \in \text{shards}(\tau)$, identical outcomes from $\mathbf{g}_{S''} - \mathbf{f}_{S''}$ distinct replicas in $S''$, then it considers $\tau$ to be successfully executed. In Figure 2, we sketched the working of CCHIMERA.

The *cross-shard exchange* step of CCHIMERA at $S$ involves waiting for other shards $S'$. Hence, there is the danger of deadlocks if the other shards $S'$ never perform their cross-shard exchange steps. To assure that such situations do not lead to a deadlock, we employ two techniques.

1. *Internal propagation*. To deal with situations in which

some shards $S \in$ shards$(\tau)$ did not receive $\langle\tau\rangle_c$ (e.g., due to network failure or due to a faulty client that fails to send $\langle\tau\rangle_c$ to $S$), we allow each shard to learn $\tau$ from any other shard. In specific, $S$ will start consensus on $\langle\tau\rangle_c$ after receiving *cross-shard exchange* related to $\langle\tau\rangle_c$.

2. *Concurrent resolution.* To deal with concurrent transactions that content for the same objects, we allow each shard to accept and execute transactions for different rounds concurrently. To assure that concurrent execution does not lead to inconsistent state updates, each replica implements the following *first-pledge* and *ordered-commit* rules. Let $\tau$ be a transaction with $S \in$ shards$(\tau)$ and $R \in S$. The *first-pledge* rule states that $S$ pledges $o$, constructed in round $\rho$, to transaction $\tau$ only if $\tau$ is the first transaction proposed after round $\rho$ with $o \in$ Inputs$(\tau)$. The *ordered-commit* rule states that $S$ can abort $\tau$ in any order, but will only commit $\tau$ that is accepted in round $\rho$ after previous rounds finished execution.

Abort decisions at shard $S$ on a transaction $\tau$ can often be made without waiting for all shards $S' \in$ shards$(\tau)$: shard $S$ can decide abort after it detects $I(S, \tau) \neq D(S, \tau)$ or after it receives the first message $(\langle\tau\rangle_c, I(S'', \tau), D(S'', \tau))$ with $I(S'', \tau) \neq D(S'', \tau)$, $S'' \in$ shards$(\tau)$. For efficiency, we allow $S$ to abort in these cases.

**Theorem 4.1.** *If, for all shards $S^*$, $\mathbf{g}_{S^*} > 2\mathbf{f}_{S^*}$, and Assumptions 2.1, 2.2, and 2.3 hold, then Core-*CHIMERA *satisfies Requirements R1–R6 with respect to all transactions that are not requested by malicious clients and do not involve objects with malicious owners.*

*Proof.* Let $\tau$ be a transaction. As good replicas in $S$ discard $\tau$ if it is invalid or if $S \notin$ shards$(\tau)$, CCHIMERA provides *validity* and *shard-involvement*. Next, *shard-applicability* follow directly from the decide outcome step.

If a shard $S$ commits or aborts transaction $\tau$, then it must have completed the decide outcome and cross-shard exchange steps. Hence, all shards $S' \in$ shards$(\tau)$ must have exchanged the necessary information to $S$. By relying on cluster-sending for cross-shard exchange, $S'$ requires cooperation of all good replicas in $S'$ to exchange the necessary information to $S$. Hence, we have the guarantee that these good replicas will also perform cross-shard exchange to any other shard $S'' \in$ shards$(\tau)$. As such, every shard $S'' \in$ shards$(\tau)$ will receive the same information as $S$, complete cross-shard exchange, and make the same decision during the decide outcome step, providing *cross-shard consistency*.

Due to internal propagation and concurrent resolution, every valid transaction $\tau$ will be processed by CCHIMERA as soon as it is send to any shard $S \in$ shards$(\tau)$. Hence, every shard in shards$(\tau)$ will perform the necessary steps to eventually inform the client. As all good replicas $R \in S$, $S \in$ shards$(\tau)$, will inform the client of the outcome for $\tau$,

the majority of these inform-messages come from good replicas, enabling the client to reliably derive the true outcome. Hence, CCHIMERA provides *service* and *confirmation*. $\square$

Notice that in the object-dataset model in which we operate, each object can be constructed once and destructed once. Hence, each object $o$ can be part of at-most two committed transactions: the first of which will construct $o$ as an output, and the second of which has $o$ as an input and will consume and destruct $o$. This is independent of any other operations on other objects. As such these two transactions *cannot* happen concurrently. Consequently, we only have concurrent transactions on $o$ if the owner owner$(o)$ expresses support for several transactions that have $o$ as an input. By Assumption 2.3, the owner owner$(o)$ must be malicious in that case. As such, transactions of well-behaved clients and owners will *never abort*.

In the design of CCHIMERA, we take *full* advantage of the above observation: CCHIMERA effectively *eliminates all coordination* when deciding to process a multi-shard transaction due to which all involved shards can process a transaction *independently* with a single consensus step: all communication between shards in CCHIMERA is dedicated to exchange execution state *after* individual shards reach consensus. We can do so as any *aborts*, which could have been prevented with additional coordination, are always due to malicious behavior by clients and owners of objects. Due to this, CCHIMERA will not undo any pledges of objects to the execution of any transactions. This implies that objects that are involved in malicious transactions can get lost for future usage, while not affecting any transactions of well-behaved clients.

## 5 Optimistic-CHIMERA: Robust Transaction Processing

In the previous section, we introduced CCHIMERA, a minimalistic multi-shard transaction processing protocol that relies on properties of UTXO-like transactions to maximize performance. Although the design of CCHIMERA is simple yet effective, we see two shortcomings that limits its use. First, CCHIMERA operates under the assumption that any issues arising from concurrent transactions is due to malicious behavior of clients. As such, CCHIMERA chooses to lock out objects affected by such malicious behavior for any future usage. Second, CCHIMERA requires consecutive consensus and cluster-sending steps, which increases its transaction processing latencies. Next, we investigate how to deal with these weaknesses of CCHIMERA *without giving up* on the minimalistic nature of CCHIMERA.

To do so, we propose Optimistic-CHIMERA (OCHIMERA), which is optimized for the *optimistic* case in which we have no concurrent transactions, while providing a recovery path that can recover from concurrent transactions without locking out objects. At the core of OCHIMERA is assuring that any

issues due to malicious behavior, e.g., concurrent transactions, are *detected* in such a way that individual replicas can recover. At the same time, we want to minimize transaction processing latencies. To bridge between these two objectives, we integrate detection and cross-shard coordination within a single consensus round that runs at each affected shard.

Let $\langle \tau \rangle_c$ be a multi-shard transaction, let $\mathcal{S} \in \text{shards}(\tau)$ be an affected shard with primary $\mathcal{P}(\mathcal{S})$, and let $m(\mathcal{S}, \tau)_{v,\rho} = (\langle \tau \rangle_c, I(\mathcal{S}, \tau), D(\mathcal{S}, \tau))$ be the round-$\rho$ proposal of $\mathcal{P}(\mathcal{S})$ of view $v$ of $\mathcal{S}$. To enable detection of concurrent transactions, OCHIMERA modifies the consensus-steps of the underlying consensus protocol by applying the following high-level idea:

> A replica $R \in \mathcal{S}$, $\mathcal{S} \in \text{shards}(\tau)$, only accepts proposal $m(\mathcal{S}, \tau)_{v,\rho}$ for transaction $\tau$ if it gets confirmation that replicas in each other shard $\mathcal{S}' \in \text{shards}(\tau)$ are also accepting proposals for $\tau$. Otherwise, replica $R$ detects failure.

Next, we illustrate how to integrate the above idea in the three-phase design of PBFT, thereby turning PBFT into a multi-shard aware consensus protocol:

1. *Global preprepare*. Primary $\mathcal{P}(\mathcal{S})$ must send $m(\mathcal{S}, \tau)_{v,\rho}$ to all replicas $R' \in \mathcal{S}'$, $\mathcal{S}' \in \text{shards}(\tau)$. Replica $R \in \mathcal{S}$ only finishes the global preprepare phase after it receives a *global preprepare certificate* consisting of a set $M = \{ m(\mathcal{S}'', \tau)_{v'',\rho''} \mid \mathcal{S}'' \in \text{shards}(\tau) \}$ of preprepare messages from all primaries of shards affected by $\tau$.

2. *Global prepare*. After $R \in \mathcal{S}$, $\mathcal{S} \in \text{shards}(\tau)$, finishes the global preprepare phase, it sends prepare messages for $M$ to all other replicas in $R' \in \mathcal{S}'$, $\mathcal{S}' \in \text{shards}(\tau)$. Replica $R \in \mathcal{S}$ only finishes the global prepare phase for $M$ after, for every shard $\mathcal{S}' \in \text{shards}(\tau)$, it receives a *local prepare certificate* consisting of a set $P(\mathcal{S}')$ of prepare messages for $M$ from $\mathbf{g}_{\mathcal{S}'}$ distinct replicas in $\mathcal{S}'$. We call the set $\{ P(\mathcal{S}'') \mid \mathcal{S}'' \in \text{shards}(\tau) \}$ a *global prepare certificate*.

3. *Global commit*. After replica $R \in \mathcal{S}$, $\mathcal{S} \in \text{shards}(\tau)$, finishes the global prepare phase, it sends commit messages for $M$ to all other replicas in $R' \in \mathcal{S}'$, $\mathcal{S}' \in \text{shards}(\tau)$. Replica $R \in \mathcal{S}$ only finishes the global commit phase for $M$ after, for every shard $\mathcal{S}' \in \text{shards}(\tau)$, it receives a *local commit certificate* consisting of a set $C(\mathcal{S}')$ of commit messages for $M$ from $\mathbf{g}_{\mathcal{S}'}$ distinct replicas in $\mathcal{S}'$. We call the set $\{ P(\mathcal{S}'') \mid \mathcal{S}'' \in \text{shards}(\tau) \}$ a *global commit certificate*.

To minimize inter-shard communication, one can utilize threshold signatures and cluster-sending to carry over local prepare and commit certificates between shards via a few constant-sized messages. The above three-phase *global*-PBFT protocol already takes care of the *local input* and *cross-shard exchange* steps. Indeed, a replica $R \in \mathcal{S}$ that finishes the global

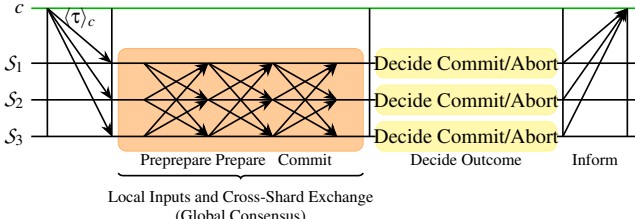

Figure 3: The message flow of OCHIMERA for a 3-shard client request $\langle \tau \rangle_c$ that is committed.

commit phase has accepted global preprepare certificate $M$, which contains all information of other shards to proceed with execution. At the same time, $R$ also has confirmation that $M$ is prepared by a majority of all good replicas in each shard $\mathcal{S}' \in \text{shards}(\tau)$ (which will eventually be followed by execution of $\tau$ within $\mathcal{S}'$). With these ingredients in place, only the *decide outcome* step remains.

The decide outcome step at shard $\mathcal{S}$ is entirely determined by the global preprepare certificate $M$. Shard $\mathcal{S}$ decides to *commit* whenever $I(\mathcal{S}', \tau) = D(\mathcal{S}', \tau)$ for all $(\langle \tau \rangle_c, I(\mathcal{S}', \tau), D(\mathcal{S}', \tau)) \in M$. Otherwise, it decides *abort*. If $\mathcal{S}$ decides commit, then all good replicas in $\mathcal{S}$ destruct all objects in $D(\mathcal{S}, \tau)$ and construct all objects $o \in \text{Outputs}(\tau)$ with $\mathcal{S} = \text{shard}(o)$. Finally, each good replica informs $c$ of the outcome of execution. If $c$ receives, from every shard $\mathcal{S}' \in \text{shards}(\tau)$, identical outcomes from $\mathbf{g}_{\mathcal{S}'} - \mathbf{f}_{\mathcal{S}'}$ distinct replicas in $\mathcal{S}'$, then it considers $\tau$ to be successfully executed. In Figure 3, we sketched the working of OCHIMERA.

We note that OCHIMERA is not the only multi-shard aware consensus protocol recently proposed (e.g., [3, 4]). What sets OCHIMERA apart is how it guarantees correctness *in all environments*, which is determined by how OCHIMERA deals with *non-optimistic cases* in which failure is detected and recovery is necessary. We will detail recovery next. As a first step, we illustrate the ways in which the normal-case of OCHIMERA can fail (e.g., due to malicious behavior of clients, failing replicas, or unreliable communication).

*Example* 5.1. Consider a transaction $\tau$ proposed by client $c$ and affecting shard $\mathcal{S} \in \text{shards}(\tau)$. First, we consider the case in which $\mathcal{P}(\mathcal{S})$ is malicious and tries to set up a coordinated attack. To have maximum control over the steps of OCHIMERA, the primary sends the message $m(\mathcal{S}, \tau)_{v,\rho}$ to only $\mathbf{g}_{\mathcal{S}''} - \mathbf{f}_{\mathcal{S}''}$ good replicas in each shard $\mathcal{S}'' \in \text{shards}(\tau)$. By doing so, $\mathcal{P}(\mathcal{S})$ can coordinate the faulty replicas in each shard to assure failure of any phase at any replica $R' \in \mathcal{S}'$, $\mathcal{S}' \in \tau$:

1. To prevent $R'$ from finishing the global preprepare phase (and start the global prepare phase) for an $M$ with $m(\mathcal{S}', \tau)_{v',\rho'} \in M$, $\mathcal{P}(\mathcal{S})$ simply does not send $m(\mathcal{S}, \tau)_{v,\rho}$ to $R'$.

2. To prevent $R'$ from finishing the global prepare phase (and start the global commit phase) for $M$, $\mathcal{P}(\mathcal{S})$ instructs

the faulty replicas in $\mathcal{F}(\mathcal{S})$ to not send prepare messages for $M$ to $\textsc{r}'$. Hence, $\textsc{r}'$ will receive at-most $\mathbf{g}_{\mathcal{S}} - \mathbf{f}_{\mathcal{S}}$ prepare messages for $M$ from replicas in $\mathcal{S}$, assuring that it will not receive a local prepare certificate $P(\mathcal{S})$ and will not finish the global prepare phase for $M$.

3. Likewise, to prevent $\textsc{r}'$ from finishing the global commit phase (and start execution) for $M$, $\mathcal{P}(\mathcal{S})$ instructs the faulty replicas in $\mathcal{F}(\mathcal{S})$ to not send commit messages to $\textsc{r}'$. Hence, $\textsc{r}'$ will receive at-most $\mathbf{g}_{\mathcal{S}} - \mathbf{f}_{\mathcal{S}}$ commit messages for $M$ from replicas in $\mathcal{S}$, assuring that it will not receive a local commit certificate $C(\mathcal{S})$ and will not finish the global commit phase for $M$.

None of the above attacks can be attributed to faulty behavior of $\mathcal{P}(\mathcal{S})$ as unreliable communication can result in the same outcomes for $\textsc{r}'$. Furthermore, even if communication is reliable and $\mathcal{P}(\mathcal{S})$ is good, replica $\textsc{r}'$ can see the same outcomes due to malicious behavior of the client or of primaries of other shards in $\texttt{shards}(\tau)$:

1. The client $c$ can be malicious and not send $\tau$ to $\mathcal{S}$. At the same time, all other primaries $\mathcal{P}(\mathcal{S}'')$ of shards $\mathcal{S}'' \in \texttt{shards}(\tau)$ can be malicious and not send anything to $\mathcal{S}$ either. In this case, $\mathcal{P}(\mathcal{S})$ will never be able to send any message $m(\mathcal{S}, \tau)_{v,\rho}$ to $\textsc{r}'$, as no replica in $\mathcal{S}$ is aware of $\tau$.

2. If any primary $\mathcal{P}(\mathcal{S}'')$ of $\mathcal{S}'' \in \texttt{shards}(\tau)$ is malicious, then it can prevent some replicas in $\mathcal{S}$ from starting the global prepare phase, thereby preventing these replicas to send prepare messages to $\textsc{r}'$. If $\mathcal{P}(\mathcal{S}'')$ prevents sufficient replicas in $\mathcal{S}$ from starting the global prepare phase, $\textsc{r}'$ will be unable to finish the global prepare phase.

3. Likewise, any malicious primary $\mathcal{P}(\mathcal{S}'')$ of $\mathcal{S}'' \in \texttt{shards}(\tau)$ can prevent replicas in $\mathcal{S}$ from starting the global commit phase, thereby assuring that $\textsc{r}'$ will be unable to finish the global commit phase.

To deal with malicious behavior, OCHIMERA needs a robust recovery mechanism. We cannot simply build that mechanism on top of traditional view-change approaches: these traditional view-change approaches require that one can identify a single source of failure (when communication is reliable), namely the current primary. As Example 5.1 already showed, this property does not hold for OCHIMERA. To remedy this, the recovery mechanisms of OCHIMERA has components that perform *local view-change* and that perform *global state recovery*. The pseudo-code for this recovery protocol can be found in Figure 4. Next, we describe the working of this recovery protocol in detail. Let $\textsc{r} \in \mathcal{S}$ be a replica that determines that it cannot finish a round $\rho$ of view $v$.

First, $\textsc{r}$ determines whether it already has a *guarantee* on which transaction it has to execute in round $\rho$. This is the case when the following conditions are met: $\textsc{r}$ finished

```
1:  event R ∈ S is unable to finish round ρ of view v do
2:      if R finished in round ρ the global prepare phase for M,
            but is unable to finish the global commit phase then
3:          Let P be the global prepare certificate of R for M.
4:          if R has a local commit certificate C(S″) for M then
5:              for S′ ∈ shards(τ) do
6:                  if R did not yet receive a local commit certificate C(S′) then
7:                      Broadcast ⟨VCGlobalSCR : M, P, C(S″)⟩ to all replicas in S′.
8:              else Detect the need for local state recovery of round ρ of view (Figure 5).
9:          else Detect the need for local state recovery of round ρ of view v (Figure 5).
10:     (Eventually repeat this event if R remains unable to finish round ρ.)

11: event R′ ∈ S′ receives message ⟨VCGlobalSCR : M, P, C(S″)⟩ from R ∈ S do
12:     if R′ did not reach the global commit phase for M then
13:         Use M, P, and C(S″) to reach the global commit phase for M.
14:     else Send a commit message for M to R.
```

Figure 4: The view-change *global short-cut recovery path* that determines whether $\textsc{r}$ already has the assurance that the current transaction will be committed. If this is the case, then $\textsc{r}$ requests only the missing information to proceed with execution. Otherwise, $\textsc{r}$ requires at-least local recovery (Figure 5).

the global prepare phase for $M$ with $m(\mathcal{S}, \tau)_{v,\rho} \in M$ and has received a local commit certificate $C(\mathcal{S}'')$ for $M$ from some shard $\mathcal{S}'' \in \texttt{shards}(\tau)$. In this case, $\textsc{r}$ can simply request all missing local commit certificates directly, as $C(\mathcal{S}'')$ can be used to prove to any involved replica $\textsc{r}' \in \mathcal{S}'$, $\mathcal{S}' \in \texttt{shards}(\tau)$, that $\textsc{r}'$ also needs to commit to $M$. To request such missing commit certificates of $\mathcal{S}'$, replica $\textsc{r}$ sends out $\texttt{VCGlobalSCR}$ messages to all replicas in $\mathcal{S}'$ (Line 7 of Figure 4). Any replica $\textsc{r}'$ that receives such a $\texttt{VCGlobalSCR}$ message can use the information in that message to reach the global commit phase for $M$ and, hence, provide $\textsc{r}$ with the requested commit messages (Line 11 of Figure 4).

If $\textsc{r}$ does not have a *guarantee* itself on which transaction it has to execute in round $\rho$, then it needs to determine whether any other replica (either in its own shard or in any other shard) has already received and acted upon such a guarantee. To initiate such local and global state recovery, $\textsc{r}$ simply detects the current view as faulty. To do so, $\textsc{r}$ broadcasts a $\texttt{VCRecoveryRQ}$ message to all other replicas in $\mathcal{S}$ that contains all information $\textsc{r}$ collected on round $\rho$ in view $v$ (Line 4 of Figure 5). Other replicas $\textsc{q} \in \mathcal{S}$ that already have *guarantees* for round $\rho$ can help $\textsc{r}$ by providing all missing information (Line 6 of Figure 5). On receipt of this information, $\textsc{r}$ can proceed with the round (Line 7 of Figure 5). If no replicas can provide the missing information, then eventually all good replicas will detect the need for local recovery, this either by themselves (Line 1 of Figure 5) or after receiving $\texttt{VCRecoveryRQ}$ messages of at-least $\mathbf{f}_{\mathcal{S}} + 1$ distinct replicas in $\mathcal{S}$, of which at-least a single replica must be good (Line 10 of Figure 5).

Finally, if a replica $\textsc{r}$ receives $\mathbf{g}_{\mathcal{S}}$ $\texttt{VCRecoveryRQ}$ messages, then it has the guarantee that at least $\mathbf{g}_{\mathcal{S}} - \mathbf{f}_{\mathcal{S}} \geq \mathbf{f}_{\mathcal{S}} + 1$ of these messages come from good replicas in $\mathcal{S}$. Hence, due to Line 10 of Figure 5, all $\mathbf{g}_{\mathcal{S}}$ good replicas in $\mathcal{S}$ will send

1: **event** R ∈ $\mathcal{S}$ detects the need for local state recovery of round ρ of view $v$ **do**
2:    Let $M$ be any latest global preprepare certificate accepted for round ρ by R.
3:    Let $S$ be $M$ and any prepare and commit certificates for $M$ collected by R.
4:    Broadcast ⟨VCRecoveryRQ : $v$, ρ, $S$⟩.

5: **event** Q ∈ $\mathcal{S}$ receives messages ⟨VCRecoveryRQ : $v$, ρ, $S$⟩ of R ∈ $\mathcal{S}$ and Q has

     1. started the global prepare phase for $M$ with $m(\mathcal{S}, \tau)_{w,\rho} \in M$;
     2. a global prepare certificate for $M$;
     3. a local commit certificate $C(\mathcal{S}'')$ for $M$

  **do**
6:    Send ⟨VCLocalSCR : $M$, $P$, $C(\mathcal{S}'')$⟩ to R ∈ $\mathcal{S}$.

7: **event** R ∈ $\mathcal{S}$ receives message ⟨VCLocalSCR : $M$, $P$, $C(\mathcal{S}'')$⟩ from Q ∈ $\mathcal{S}$ **do**
8:    **if** R did not reach the global commit phase for $M$ **then**
9:      Use $M$, $P$, and $C$ to reach the global commit phase for $M$.

10: **event** R ∈ $\mathcal{S}$ receives messages ⟨VCRecoveryRQ : $v_i$, ρ, $S_i$⟩, $1 \le i \le \mathbf{f}_S + 1$,
     from $\mathbf{f}_S + 1$ distinct replicas in $\mathcal{S}$ **do**
11:    R detects the need for local state recovery of round ρ of view $\min\{v_i \mid 1 \le i \le \mathbf{f}_S + 1\}$.

12: **event** R ∈ $\mathcal{S}$ receives messages ⟨VCRecoveryRQ : $v$, ρ, $S_i$⟩, $1 \le i \le \mathbf{g}_S$,
     from distinct replicas in $\mathcal{S}$ **do**
13:    **if** id(R) $\ne (v+1) \bmod \mathbf{n}_S$ **then**
14:      (R awaits the NewView message of the new primary, Line 14 of Figure 6.)
15:    **else**
16:      Broadcast ⟨NewView : ⟨VCRecoveryRQ : $v$, ρ, $S_i$⟩ $\mid 1 \le i \le \mathbf{g}_S$⟩ to all replicas in $\mathcal{S}$.
17:      **if** there exists a $S_i$ that contains global preprepare certificate $M$,
       but no $S_j$ contains a local commit certificate for $M$ **then**
18:        R initiates global state recovery of round ρ (Line 1 of Figure 6).

Figure 5: The view-change *local short-cut recovery path* that determines whether some Q can provide R with the assurance that the current transaction will be committed. If this is the case, then R only needs this assurance, otherwise $\mathcal{S}$ requires a new view (Figure 6).

1: **event** P ∈ $\mathcal{S}$ initiates global state recovery of round ρ using ⟨NewView : $V$⟩ **do**
2:    Let $T$ be the transactions with global preprepare certificates for round ρ of $\mathcal{S}$ in view $V$.
3:    Let $S$ be the shards affected by transactions in $T$.
4:    Broadcast ⟨VCGlobalStateRQ : $v$, ρ, $V$⟩ to all replicas in $\mathcal{S}' \in S$.
5:    **for** $\mathcal{S}' \in S$ **do**
6:      Wait for VCGlobalStateRQ messages for $V$ from $\mathbf{g}_{\mathcal{S}'}$ distinct replicas in $\mathcal{S}'$.
7:      Let $W(\mathcal{S}')$ be the set of received VCGLOBALSTATERQ messages.
8:    Broadcast ⟨NewViewGlobal : $V$, $\{W(\mathcal{S}') \mid \mathcal{S}' \in S\}$⟩ to all replicas in $\mathcal{S}$.

9: **event** R' ∈ $\mathcal{S}'$ receives message ⟨VCGlobalStateRQ : $v$, ρ, $V$⟩ from P ∈ $\mathcal{S}$ **do**
10:    **if** R' has a global preprepare certificate $M$ with $m(\mathcal{S}, \tau)_{w,\rho} \in M$
     and reached the global commit phase for $M$ **then**
11:      Let $P$ be the global prepare certificate for $M$.
12:      Send ⟨VCGlobalStateR : $v$, ρ, $V$, $M$, $P$⟩ to P.
13:    **else** Send ⟨VCGlobalStateR : $v$, ρ, $V$⟩ to P.

14: **event** R ∈ $\mathcal{S}$ receives valid ⟨NewView : $V$⟩ message from replica P **do**
15:    **if** there exists a ⟨VCRecoveryRQ : $v_i$, ρ, $S_i$⟩ ∈ $V$ that contains
     a global preprepare certificate $M$ with $m(\mathcal{S}, \tau)_{w,\rho} \in M$,
     a global prepare certificate $P$ for $M$, and a local commit certificate $C(\mathcal{S}'')$
     for $M$ **then**
16:      Use $M$, $P$, and $C$ to reach the global commit phase for $M$.
17:    **else if** there exists a ⟨VCRecoveryRQ : $v_i$, ρ, $S_i$⟩ ∈ $V$ that contains
     a global preprepare certificate $M$,
     but no ⟨VCRecoveryRQ : $v_j$, ρ, $S_j$⟩ ∈ $V$ contains a local commit certificate
     for $M$ **then**
18:      R detects the need for global state recovery of round ρ (Line 20 of Figure 6).
19:    **else** (P must propose for round ρ.)

20: **event** R ∈ $\mathcal{S}$ receives valid ⟨NewViewGlobal : $V$, $W$⟩ from P ∈ $\mathcal{S}$ **do**
21:    **if** any message in $W$ is of the form ⟨VCGlobalStateR : $v$, ρ, $V$, $M$, $P$⟩ **then**
22:      Select ⟨VCGlobalStateR : $v$, ρ, $V$, $M$, $P$⟩ ∈ $W$ with latest view $w$,
     $m(\mathcal{S}, \tau)_{w,\rho} \in M$.
23:      Use $M$ and $P$ to reach the global commit phase for $M$.
24:    **else** (P must propose for round ρ.)

Figure 6: The view-change *new-view recovery path* that recovers the state of the previous view based on a NewView proposal of the new primary. As part of the new-view recovery path, the new primary can construct a global new-view that contains the necessary information from other shards to reconstruct the local state.

VCRecoveryRQ, and, when communication is reliable, also receive these messages. Consequently, at this point, R can start the new view by electing a new primary and awaiting the NewView proposal of this new primary (Line 12 of Figure 5). If R is the new primary, then it starts the new view by proposing a NewView. As other shards *could* have already made final decisions depending on local prepare or commit certificates of $\mathcal{S}$ for round ρ, we need to assure that such certificates are not invalidated. To figure out whether such final decisions have been made, the new primary will query other shards $\mathcal{S}'$ for their state whenever the NewView message contains global preprepare certificates for transactions τ, $\mathcal{S}' \in$ shards(τ), but not a local commit certificate to *guarantee* execution of τ (Line 17 of Figure 5).

The new-view process has three stages. First, the new primary P proposes the new-view via a NewView message (Line 12 of Figure 5). If necessary, the new primary P also requests the relevant global state from any relevant shard (Line 1 of Figure 6). The replicas in other shards will respond to this request with their local state (Line 9 of Figure 6). The new primary collects these responses and sends them to all replicas in $\mathcal{S}$ via a NewViewGlobal message. Then, after P sends the NewView message to R ∈ $\mathcal{S}$, R determines whether the NewView message contains sufficient information

to recover round ρ (Line 15 of Figure 6), contains sufficient information to wait for any relevant global state (Line 17 of Figure 6), or to determine that the new primary must propose for round ρ (Line 19 of Figure 6). If R determines it needs to wait for any relevant global state, then R will wait for this state to arrive via a NewViewGlobal message. Based on the received global state, R determines to recover round ρ (Line 21 of Figure 6), or determines that the new primary must propose for round ρ (Line 24 of Figure 6).

Next, we will prove the correctness of the view-change of OCHIMERA. First, using a standard quorum argument, we prove that in a single round of a single view of $\mathcal{S}$, only a single global preprepare message affecting $\mathcal{S}$ can get committed by any other affected shards:

**Lemma 5.1.** *Let* $\tau_1$ *and* $\tau_2$ *be transactions with* $\mathcal{S} \in$ $(shards(\tau_1) \cap shards(\tau_2))$. *If* $\mathbf{g}_S > 2\mathbf{f}_S$ *and there exists shards* $\mathcal{S}_i \in shards(\tau_i)$, $i \in \{1, 2\}$, *such that good replicas* $R_i \in \mathcal{G}(\mathcal{S}_i)$ *reached the global commit phase for global preprepare certificate* $M_i$ *with* $m(\mathcal{S}, \tau_i)_{v,\rho} \in M_i$, *then* $\tau_1 = \tau_2$.

*Proof.* We prove this property using contradiction. We assume $\tau_1 \ne \tau_2$. Let $P_i(\mathcal{S})$ be the local prepare certificate pro-

vided by $\mathcal{S}$ for $M_i$ and used by $\text{R}_i$ to reach the global commit phase, let $S_i \subseteq \mathcal{S}$ be the $\mathbf{g}_\mathcal{S}$ replicas in $\mathcal{S}$ that provided the prepare messages in $P_i(\mathcal{S})$, and let $T_i = S_i \setminus \mathcal{F}(\mathcal{S})$ be the good replicas in $S_i$. By construction, we have $|T_i| \geq \mathbf{g}_\mathcal{S} - \mathbf{f}_\mathcal{S}$. As all replicas in $T_1 \cup T_2$ are good, they will only send out a single prepare message per round $\rho$ of view $v$. Hence, if $\tau_1 \neq \tau_2$, then $T_1 \cap T_2 = \emptyset$, and we must have $2(\mathbf{g}_\mathcal{S} - \mathbf{f}_\mathcal{S}) \leq |T_1 \cup T_2|$. As all replicas in $T_1 \cup T_2$ are good, we also have $|T_1 \cup T_2| \leq \mathbf{g}_\mathcal{S}$. Hence, $2(\mathbf{g}_\mathcal{S} - \mathbf{f}_\mathcal{S}) \leq \mathbf{g}_\mathcal{S}$, which simplifies to $\mathbf{g}_\mathcal{S} \leq 2\mathbf{f}_\mathcal{S}$, a contradiction. Hence, we conclude $\tau_1 = \tau_2$. □

Next, we use Lemma 5.1 to prove that any global prepare certificate that *could* have been accepted by any good affected replica is preserved by OCHIMERA:

**Proposition 5.1.** *Let $\tau$ be a transaction and $m(\mathcal{S},\tau)_{v,\rho}$ be a preprepare message. If, for all shards $\mathcal{S}^*$, $\mathbf{g}_{\mathcal{S}^*} > 2\mathbf{f}_{\mathcal{S}^*}$, and there exists a shard $\mathcal{S}' \in \text{shards}(\tau)$ such that $\mathbf{g}_{\mathcal{S}'} - \mathbf{f}_{\mathcal{S}'}$ good replicas in $\mathcal{S}'$ reached the global commit phase for $M$ with $m(\mathcal{S},\tau)_{v,\rho} \in M$, then every successful future view of $\mathcal{S}$ will recover $M$ and assure that the good replicas in $\mathcal{S}$ reach the commit phase for $M$.*

*Proof.* Let $v^* \leq v$ be the first view in which a global prepare certificate $M^*$ with $m(\mathcal{S},\tau^*)_{v^*,\rho} \in M^*$ satisfied the premise of this proposition. Using induction on the number of views after the first view $v^*$, we will prove the following two properties on $M^*$:

1. every good replica that participates in view $w$, $v^* < w$, will recover $M^*$ upon entering view $w$ and reach the commit phase for $M^*$; and

2. no replica will be able to construct a local prepare certificate of $\mathcal{S}$ for any global preprepare certificate $M^\dagger \neq M^*$ with $m(\mathcal{S},\tau^\dagger)_{w,\rho} \in M^\dagger$, $v^* < w$.

The base case is view $v^* + 1$. Let $S' \subseteq \mathcal{G}(\mathcal{S}')$ be the set of $\mathbf{g}_{\mathcal{S}'} - \mathbf{f}_{\mathcal{S}'}$ good replicas in $\mathcal{S}'$ that reached the global commit phase for $M^*$. Each replica $\text{R}' \in S'$ has a local prepare certificate $P(\mathcal{S})$ consisting of $\mathbf{g}_\mathcal{S}$ prepare messages for $M^*$ provided by replicas in $\mathcal{S}$. We write $S(\text{R}') \subseteq \mathcal{G}(\mathcal{S})$ to denote the at-least $\mathbf{g}_\mathcal{S} - \mathbf{f}_\mathcal{S}$ good replicas in $\mathcal{S}$ that provided such a prepare message to $\text{R}'$.

Consider any valid new-view proposal $\langle \text{NewView} : V \rangle$ for view $v^* + 1$. If the conditions of Line 15 of Figure 6 hold for global preprepare certificate $M^\dagger$ with $m(\mathcal{S},\tau^\ddagger)_{w,\rho} \in M^\ddagger$, then we recover $M^\ddagger$. As there is a local commit certificate for $M^\ddagger$ in this case, the premise of this proposition holds on $M^\ddagger$. As $v^*$ is the first view in which the premise of this proposition hold, we can use Lemma 5.1 to conclude that $w = v^*$, $M^\ddagger = M^*$, and, hence, that the base case holds if the conditions of Line 15 of Figure 6 hold. Next, we assume that the conditions of Line 15 of Figure 6 do not hold, in which case $M^*$ can only be recovered via global state recovery. As the first step in global state recovery is proving that the condition of Line 17

of Figure 6 holds. Let $T \subseteq \mathcal{G}(\mathcal{S})$ be the set of at-least $\mathbf{g}_\mathcal{S} - \mathbf{f}_\mathcal{S}$ good replicas in $\mathcal{S}$ whose VCRecoveryRQ message is in $V$ and let $\text{R}' \in S'$. We have $|S(\text{R}')| \geq \mathbf{g}_\mathcal{S} - \mathbf{f}_\mathcal{S}$ and $|T| \geq \mathbf{g}_\mathcal{S} - \mathbf{f}_\mathcal{S}$. Hence, by a standard quorum argument, we conclude $S(\text{R}') \cap T \neq \emptyset$. Let $\text{Q} \in (S(\text{R}') \cap T)$. As $\text{Q}$ is good and send prepare messages for $M^*$, it must have reached the global prepare phase for $M^*$. Consequently, the condition of Line 17 of Figure 6 holds and to complete the proof, we only need to prove that any well-formed NewViewGlobal message will recover $M^*$.

Let $\langle \text{NewViewGlobal} : V, W \rangle$ be any valid global new-view proposal for view $v^* + 1$. As $\text{Q}$ reached the global prepare phase for $M^*$, any valid global new-view proposal must include messages from $\mathcal{S}' \in \text{shards}(\tau)$. Let $U' \subseteq \mathcal{S}'$ be the replicas in $\mathcal{S}'$ of whom messages VCGlobalStateR are included in $W$. Let $V' = U' \setminus \mathcal{F}(\mathcal{S}')$. We have $|S'| \geq \mathbf{g}_{\mathcal{S}'} - \mathbf{f}_{\mathcal{S}'}$ and $|V'| \geq \mathbf{g}_{\mathcal{S}'} - \mathbf{f}_{\mathcal{S}'}$. Hence, by a standard quorum argument, we conclude $S' \cap V' \neq \emptyset'$. Let $\text{Q}' \in (S' \cap V')$. As $\text{Q}'$ reached the global commit phase for $M^*$, it will meet the conditions of Line 23 of Figure 6 and provide both $M^*$ and a global prepare certificate for $M^*$. Let $M^\ddagger$ be any other global preprepare certificate in $W$ accompanied by a global prepare certificate. Due to Line 22 of Figure 6, the global preprepare certificate for the newest view of $\mathcal{S}$ will be recovered. As $v^*$ is the newest view of $\mathcal{S}$, $M^\ddagger$ will only prevent recovery of $M^*$ if it is also a global preprepare certificate for view $v^*$ of $\mathcal{S}$. In this case, Lemma 5.1 guarantees that $M^\ddagger = M^*$. Hence, any replica $\text{R}$ will recover $M^*$ upon receiving $\langle \text{NewViewGlobal} : V, W \rangle$.

Now assume that the induction hypothesis holds for all views $j$, $v^* < j \leq i$. We will prove that the induction hypothesis holds for view $i + 1$. Consider any valid new-view proposal $\langle \text{NewView} : V \rangle$ for view $i + 1$ and let $M^\ddagger$ with $m(\mathcal{S},\tau^\ddagger)_{w,\rho} \in M^\ddagger$ be any global preprepare certificate that is recovered due to the new-view proposal $\langle \text{NewView} : V \rangle$. Hence, $M^\ddagger$ is recovered via either Line 16 of Figure 6 or Line 23 of Figure 6. In both cases, there must exist a global prepare certificate $P$ for $M^\ddagger$. As $\langle \text{NewView} : V \rangle$ is valid, we must have $w \leq i$. Hence, we can apply the second property of the induction hypothesis to conclude that $w \leq v^*$. If $w = v^*$, then we can use Lemma 5.1 to conclude that $M^\ddagger = M^*$. Hence, to complete the proof, we must show that $w = v^*$. First, the case in which $M^\ddagger$ is recovered via Line 16 of Figure 6. Due to the existence of a global commit certificate $C$ for $M^\ddagger$, $M^\ddagger$ satisfies the premise of this proposition. By assumption, $v^*$ is the first view for which the premise of this proposition holds. Hence, $w \geq v^*$, in which case we conclude $M^\ddagger = M^*$. Last, the case in which $M^\ddagger$ is recovered via Line 23 of Figure 6. In this case, $M^\ddagger$ is recovered via some message $\langle \text{NewViewGlobal} : V, W \rangle$. Analogous to the proof for the base case, $V$ will contain a message VCRecoveryRQ from some replica $\text{Q} \in S(\text{R}')$. Due to Line 2 of Figure 5, $\text{Q}$ will provide information on $M^*$. Consequently, a prepare certificate for $M^*$ will be obtained via global state recovery, and we also conclude $M^\ddagger = M^*$. □

Lemma 5.1 and Proposition 5.1 assure that no transaction that could-be-committed by any replica will ever get lost by the system. Next, we bootstrap these technical properties to prove that all good replicas can always recover such could-be-committed transactions.

**Proposition 5.2.** *Let $\tau$ be a transaction and $m(\mathcal{S}, \tau)_{v,\rho}$ be a prepreparemessage. If, for all shards $\mathcal{S}^*$, $\mathbf{g}_{\mathcal{S}^*} > 2\mathbf{f}_{\mathcal{S}^*}$, and there exists a shard $\mathcal{S}' \in \text{shards}(\tau)$ such that $\mathbf{g}_{\mathcal{S}'} - \mathbf{f}_{\mathcal{S}'}$ good replicas in $\mathcal{S}'$ reached the global commit phase for $M$ with $m(\mathcal{S}, \tau)_{v,\rho} \in M$, then every good replica in $\mathcal{S}$ will accept $M$ whenever communication becomes reliable.*

*Proof.* Let $R \in \mathcal{S}$ be a good replica that is unable to accept $M$. At some point, communication becomes reliable, after which $R$ will eventually trigger Line 1 of Figure 4. We have the following cases:

1. If $R$ meets the conditions of Line 4 of Figure 4, then $R$ has a local commit certificate $C(\mathcal{S}'')$, $\mathcal{S}'' \in \text{shards}(\tau)$. This local commit certificate certifies that at least $\mathbf{g}_{\mathcal{S}''} - \mathbf{f}_{\mathcal{S}''}$ good replicas in $\mathcal{S}''$ finished the global prepare phase for $M$. Hence, the conditions for Proposition 5.1 are met for $M$ and, hence, any shard in $\text{shards}(\tau)$ will maintain or recover $M$. Replica $R$ can use $C(\mathcal{S}'')$ to prove this situation to other replicas, forcing them to commit to $M$, and provide any commit messages $R$ is missing (Line 11 of Figure 4).

2. If $R$ does not meet the conditions of Line 4 of Figure 4, but some other good replica $Q \in \mathcal{S}$ does, then $Q$ can provide all missing information to $R$ (Line 6 of Figure 5). Next, $R$ uses this information (Line 7 of Figure 5), after which it meets the conditions of Line 4 of Figure 4.

3. Otherwise, if the above two cases do not hold, then all $\mathbf{g}_{\mathcal{S}}$ good replicas in $\mathcal{S}$ are unable to finish the commit phase. Hence, they perform a view-change. Due to Proposition 5.1, this view-change will succeed and put every replica in $\mathcal{S}$ into the commit phase for $M$. As all good replicas in $\mathcal{S}$ are in the commit phase, each good replica in $\mathcal{S}$ will be able to make a local commit certificate $C(\mathcal{S})$ for $M$, after which they meet the conditions of Line 4 of Figure 4. □

Finally, we use Proposition 5.2 to prove *cross-shard-consistency*.

**Theorem 5.2.** *Optimistic-*CHIMERA *maintains cross-shard consistency.*

*Proof.* Assume a single good replica $R \in \mathcal{S}$ commits or aborts a transaction $\tau$. Hence, it accepted some global prepreparecertificate $M$ with $m(\mathcal{S}, \tau)_{v,\rho} \in M$. Consequently, $R$ has local commit certificates $C(\mathcal{S}')$ for $M$ of every $\mathcal{S}' \in \text{shards}(\tau)$. Hence, at least $\mathbf{g}_{\mathcal{S}'} - \mathbf{f}_{\mathcal{S}'}$ good replicas in $\mathcal{S}'$ reached the global commit phase for $M$, and we can apply Proposition 5.2 to

conclude that any good replica $R'' \in \mathcal{S}''$, $\mathcal{S}'' \in \text{shards}(\tau)$ will accept $M$. As $R''$ bases its commit or abort decision for $\tau$ on the same global prepare certificate $M$ as $R$, they will both make the same decision, completing the proof. □

Due to the similarity between OCHIMERA and CCHIMERA, one can use the details of Theorem 4.1 to prove that OCHIMERA provides *validity*, *shard-involvement*, and *shard-applicability*. Via Theorem 5.2, we proved *cross-shard-consistency*. We cannot prove *service* and *confirmation*, however. The reason for this is simple: even though OCHIMERA can detect and recover from accidental faulty behavior and accidental concurrent transactions, OCHIMERA is not designed to gracefully handle targeted attacks: OCHIMERA is optimistic in the sense that it is optimized for the situation in which faulty behavior (including concurrent transactions that content for the same objects) is rare. Still, in all cases, OCHIMERA maintains cross-shard consistency, however. Moreover, in the optimistic case in which shards have good primaries and no concurrent transactions exist, progress is guaranteed whenever communication is reliable:

**Proposition 5.3.** *If, for all shards $\mathcal{S}^*$, $\mathbf{g}_{\mathcal{S}^*} > 2\mathbf{f}_{\mathcal{S}^*}$, and Assumptions 2.1, 2.2, and 2.3 hold, then Optimistic-*CHIMERA *satisfies Requirements R1–R6 in the optimistic case.*

OCHIMERA cannot defend against denial-of-service attacks targeted at blocking individual replicas and shards from participating. Unfortunately, no existing consensus protocol is able to deal with such attacks. Furthermore, as is the case for other multi-shard consensus protocols, coordinated attempts can prevent OCHIMERA from making progress in periods when the optimistic assumption does not hold. At the core of such attacks is the ability for malicious clients and malicious primaries to corrupt the operations of shards coordinated by good primaries, as already shown in Example 5.1. Due to Theorem 5.2, such attacks will *never* affect consistency in OCHIMERA, however.

To further reduce the impact of targeted attacks, one can make primary election non-deterministic, e.g., by using shard-specific distributed coins to elect new primaries in individual shards [11, 13]. Finally, we remark that we have presented OCHIMERA with a per-round checkpoint and recovery method. In this simplified design, the recovery path only has to recover at-most a single round. Our approach can easily be generalized to a more typical multi-round checkpoint and recovery method, however. Furthermore, we believe that the way in which OCHIMERA extends PBFT can easily be generalized to other consensus protocols, e.g., POE [32] and HOTSTUFF [57].

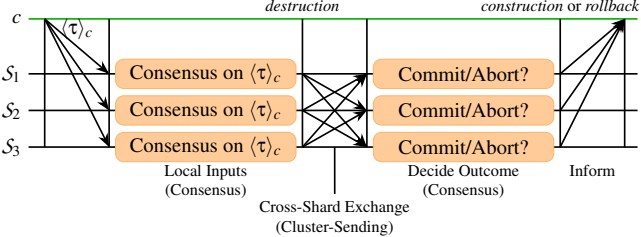

Figure 7: The message flow of RCHIMERA for a 3-shard client request $\langle\tau\rangle_c$ that is committed.

# 6 Resilient-CHIMERA: Transaction Processing Under Attack

In the previous section, we introduced OCHIMERA, a general-purpose minimalistic and efficient multi-shard transaction processing protocol. OCHIMERA is designed with the assumption that malicious behavior is rare, due to which it can minimize coordination in the normal-case while requiring intricate coordination when recovering from attacks. As an alternative to the optimistic approach of OCHIMERA, we can apply a *pessimistic* approach to CCHIMERA to gracefully recover from concurrent transactions that is geared towards minimizing the influence of malicious behavior altogether. Next, we explore such a pessimistic design via *resilient*-CHIMERA (RCHIMERA).

The design of RCHIMERA builds upon the design of CCHIMERA by adding additional coordination to the cross-shard exchange and decide outcome steps. As in CCHIMERA, the acceptance of $m(\mathcal{S},\tau)_\rho$ in round $\rho$ by all good replicas completes the *local inputs* step. Before cross-shard exchange, the replicas in $\mathcal{S}$ destruct the objects in $D(\mathcal{S},\tau)$, thereby fully pledging these objects to $\tau$ until the commit or abort decision. Then, $\mathcal{S}$ performs cross-shard exchange by broadcasting $m(\mathcal{S},\tau)_\rho$ to all other shards in $\text{shards}(\tau)$, while the replicas in $\mathcal{S}$ wait until they receive messages $m(\mathcal{S}',\tau)_{\rho'}$ from all other shards $\mathcal{S}' \in \text{shards}(\tau)$.

After cross-shard exchange comes the final *decide outcome* step. After $\mathcal{S}$ receives $m(\mathcal{S}',\tau)_{\rho'}$ from all shards $\mathcal{S}' \in \text{shards}(\tau)$, the replicas force a *second consensus step* that determines the round $\rho^*$ at which $\mathcal{S}$ decides *commit* (whenever $I(\mathcal{S}',\tau) = D(\mathcal{S}',\tau)$ for all $\mathcal{S}' \in \text{shards}(\tau)$) or *abort*. If $\mathcal{S}$ decides commit, then, in round $\rho^*$, all good replicas in $\mathcal{S}$ construct all objects $o \in \text{Outputs}(\tau)$ with $\mathcal{S} = \text{shard}(o)$. If $\mathcal{S}$ decides abort, then, in round $\rho^*$, all good replicas in $\mathcal{S}$ reconstruct all objects in $D(\mathcal{S},\tau)$ (rollback). Finally, each good replica informs $c$ of the outcome of execution. If $c$ receives, from every shard $\mathcal{S}' \in \text{shards}(\tau)$, identical outcomes from $\mathbf{g}_{\mathcal{S}'} - \mathbf{f}_{\mathcal{S}'}$ distinct replicas in $\mathcal{S}'$, then it considers $\tau$ to be successfully executed. In Figure 7, we sketched the working of RCHIMERA.

We notice that processing a multi-shard transaction via

RCHIMERA requires *two* consensus steps per shard. In some cases, we can eliminate the second step, however. First, if $\tau$ is a multi-shard transaction with $\mathcal{S} \in \text{shards}(\tau)$ and the replicas in $\mathcal{S}$ accept $(\langle\tau\rangle_c, I(\mathcal{S},\tau), D(\mathcal{S},\tau))$ with $I(\mathcal{S},\tau) \neq D(\mathcal{S},\tau)$, then the replicas can immediately abort whenever they accept $(\langle\tau\rangle_c, I(\mathcal{S},\tau), D(\mathcal{S},\tau))$. Second, if $\tau$ is a single-shard transaction with $\text{shards}(\tau) = \{\mathcal{S}\}$, then the replicas in $\mathcal{S}$ can immediately decide commit or abort whenever they accept $(\langle\tau\rangle_c, I(\mathcal{S},\tau), D(\mathcal{S},\tau))$. Hence, in both cases, processing of $\tau$ at $\mathcal{S}$ only requires a single consensus step at $\mathcal{S}$. Next, we prove the correctness of RCHIMERA:

**Theorem 6.1.** *If, for all shards $\mathcal{S}^*$, $\mathbf{g}_{\mathcal{S}^*} > 2\mathbf{f}_{\mathcal{S}^*}$, and Assumptions 2.1, 2.2, and 2.3 hold, then Resilient-CHIMERA satisfies Requirements R1–R6.*

*Proof.* Let $\tau$ be a transaction. As good replicas in $\mathcal{S}$ discard $\tau$ if it is invalid or if $\mathcal{S} \notin \text{shards}(\tau)$, RCHIMERA provides *validity* and *shard-involvement*. Next, *shard-applicability* follow directly from the decide outcome step.

If a shard $\mathcal{S}$ commits or aborts transaction $\tau$, then it must have completed the decide outcome and cross-shard exchange steps. Hence, all shards $\mathcal{S}' \in \text{shards}(\tau)$ must have exchanged the necessary information to $\mathcal{S}$. By relying on cluster-sending for cross-shard exchange, $\mathcal{S}'$ requires cooperation of all good replicas in $\mathcal{S}'$ to exchange the necessary information to $\mathcal{S}$. Hence, we have the guarantee that these good replicas will also perform cross-shard exchange to any other shard $\mathcal{S}'' \in \text{shards}(\tau)$. Consequently, every shard $\mathcal{S}'' \in \text{shards}(\tau)$ will receive the same information as $\mathcal{S}$, complete cross-shard exchange, and make the same decision during the decide outcome step, providing *cross-shard consistency*.

A client can force service on a transaction $\tau$ by choosing a shard $\mathcal{S} \in \text{shards}(\tau)$ and sending $\tau$ to all good replicas in $\mathcal{G}(\mathcal{S})$. By doing so, the normal mechanisms of consensus can be used by the good replicas in $\mathcal{G}(\mathcal{S})$ to force acceptance on $\tau$ in $\mathcal{S}$ and, hence, bootstrapping acceptance on $\tau$ in all shards $\mathcal{S}' \in \text{shards}(\tau)$. Due to cross-shard consistency, every shard in $\text{shards}(\tau)$ will perform the necessary steps to eventually inform the client. As all good replicas $R \in \mathcal{S}$, $\mathcal{S} \in \text{shards}(\tau)$, will inform the client of the outcome for $\tau$, the majority of these inform-messages come from good replicas, enabling the client to reliably derive the true outcome. Hence, RCHIMERA provides *service* and *confirmation*. $\square$

# 7 The Ordering of Transactions in CHIMERA

Having introduced the three variants of CHIMERA in Sections 4, 5, and 6, we will now analyze the ordering guarantees provided by CHIMERA. We further refer to Section 8 for a detailed comparison of the three variants of CHIMERA. Here, we will show that CHIMERA provides serializable execution [6, 9].

The data model utilized by CCHIMERA, OCHIMERA, and RCHIMERA guarantees that any object $o$ can only be involved

in at-most *two* committed transactions: one that *constructs* $o$ and another one that *destructs* $o$. Assume the existence of such transactions $\tau_1$ and $\tau_2$ with $o \in \text{Outputs}(\tau_1)$ and $o \in \text{Inputs}(\tau_2)$. Due to *cross-shard-consistency* (Requirement R4), the shard shard($o$) will have to execute both $\tau_1$ and $\tau_2$. From these observations, we can derive a serializable order on all committed transactions:

**Theorem 7.1.** *A sharded fault-tolerant system that uses the object-dataset data model, processes UTXO-like transactions, and satisfies Requirements R1-R5 commits transactions in a serializable order.*

*Proof.* Assume the existence of transactions $\tau_1$ and $\tau_2$ with $o \in \text{Outputs}(\tau_1)$ and $o \in \text{Inputs}(\tau_2)$. Due to *shard-applicability* (Requirement R3), shard shard($o$) will execute $\tau_1$ strictly before $\tau_2$. Now consider the relation

$$\prec := \{(\tau, \tau') \mid (\text{the system committed to } \tau \text{ and } \tau') \wedge$$
$$(\text{Outputs}(\tau) \cap \text{Inputs}(\tau') \neq \emptyset)\}.$$

Obviously, we have $\prec(\tau_1, \tau_2)$. To prove that all committed transactions are executed in a *serializable* ordering, we first prove the following:

> If we interpret transactions as nodes and $\prec$ as an edge relation, then the resulting graph is *acyclic*.

The proof is by contradiction. Let $G$ be the graph-interpretation of $\prec$. We assume that graph $G$ is cyclic. Hence, there exists transactions $\tau_0, \ldots, \tau_{m-1}$ such that $\prec(\tau_i, \tau_{i+1})$, $0 \leq i < m - 1$, and $\prec(\tau_{m-1}, \tau_0)$. By the definition of $\prec$, we can choose objects $o_i$, $0 \leq i < m$, with $o_i \in (\text{Outputs}(\tau_i) \cap \text{Inputs}(\tau_{(i+1) \bmod m}))$. Due to *cross-shard-consistency* (Requirement R4), the shard shard($o_i$), $0 \leq i < m$, executed transactions $\tau_i$ and $\tau_{(i+1) \bmod m}$. Consider $o_i$, $0 \leq i < m$, and let $t_i$ be the time at which shard shard($o_i$) executed $\tau_i$ and constructed $o_i$. Due to *shard-applicability* (Requirement R3), we know that shard shard($o_i$) executed $\tau_{(i+1) \bmod m}$ strictly after $t_i$. Moreover, also shard shard($o_{(i+1) \bmod m}$) must have executed $\tau_{(i+1) \bmod m}$ strictly after $t_i$ and we derive $t_i < t_{(i+1) \bmod m}$. Hence, we must have $t_0 < t_1 < \cdots < t_{m-1} < t_0$, a contradiction. Consequently, $G$ must be acyclic.

To derive a serializable execution order for all committed transactions, we simply construct a directed acyclic graph in which transactions are nodes and $\prec$ is the edge relation. Next, we *topologically sort* the graph to derive the searched-for ordering. ∎

We notice that CHIMERA only provides serializability for *committed* transactions: concurrent transactions that content for the same objects will always be aborted and, hence, will not be executed and will not affect the serializable order of execution of transactions. It is this flexibility in dealing with aborted transactions that allows all variants of CHIMERA to operate with minimal and fully-decentralized coordination

between shards; while still providing strong isolation for all committed transactions.

# 8 Analysis of the Three CHIMERA Variants

In the previous sections, we proposed three variants of CHIMERA and showed their correctness. Next, we analyze the benefits and costs of the three CHIMERA multi-shard transaction processing protocols and compare them with state-of-the-art multi-shard transaction processing protocols. A summary of this analysis can be found in Figure 8.

## 8.1 A Comparison of CHIMERA Variants

First, Figure 8 provides a high-level comparison of the costs of each of the three CHIMERA protocols to process a single transaction $\tau$ that affects $s = |\text{shards}(\tau)|$ distinct shards. For the normal-case behavior, we compare the complexity in the number of *sequential communication phases* (which, in the idle case, are the main determinant for client latencies), the number of *consensus steps* per shard and *cross-shard exchange* steps between shards (which together determine the bandwidth costs and put an upper bound on throughput). As one can see, all three protocols have a low number of *phases*, due to which all three can provide low latencies toward clients. In environments in which cross-shard communication has low latency, OCHIMERA will be able to provide lower latencies than both CCHIMERA and RCHIMERA, as its optimistic design eliminates one phase of communication (at the cost of requiring cross-shard communication in every phase).

Next, we compare how the three protocols deal with malicious behavior by clients and by replicas. If no clients behave malicious, then all transactions will *commit*. In all three protocols, malicious behavior by clients can lead to the existence of concurrent transactions that affect the same object. Upon detection of such concurrent transactions, all three protocols will *abort*. The consequences of such an abort are different in the three protocols.

In CCHIMERA, objects affected by aborted transactions remain pledged and cannot be reused. In practice, this loss of objects can provide an incentive for clients to not behave malicious, but does limit the usability of CCHIMERA in non-incentivized environments. Both OCHIMERA and RCHIMERA deal with concurrent transactions by aborting them via the normal-case of the protocol. The three CHIMERA protocols are resilient against malicious replicas: only malicious primaries can affect the normal-case operations of these protocols. If malicious primaries behave sufficiently malicious to affect the normal-case operations, their behavior is detected, and the primary is replaced. In both CCHIMERA and RCHIMERA, dealing with a malicious primary in a shard can be done completely in isolation of all other shards. In OCHIMERA, which is optimized with the assumption that failures are rare, the failure of a primary while processing a

| Protocol | Principle Technique | Phases[a] (Cross-Shard) | Consensus Steps Total | Consensus Steps Sequential | Cross-Shard Communication[b] | Transaction Abort Causes | Transaction Concurrency and Ordering | Failure Recovery (method and when) |
|---|---|---|---|---|---|---|---|---|
| CCHIMERA | Independent Consensus UTXO Data Model | 4 (1) | $s$ | 1 | 1 (CS, A2A) | Faulty Only | Data Model Pledges (Incentive) | Local Recovery Local Primary Failure |
| OCHIMERA | Multi-Shard Consensus UTXO Data Model | 3 (3) | $s$ | 1 | 3 (GC, A2A) | Faulty Only | Data Model Aborts | Local and Global Recovery Any Primary Failure |
| RCHIMERA | Distributed Commit UTXO Data Model | 7 (1) | $2s$ | 2 | 1 (CS, A2A) | Faulty Only | Data Model Aborts | Local Recovery Local Primary Failure |
| AHL [17] | Reference Committee Non-Blocking Locks | 19 (4) | $2s + 2$ | 5 | 4 (CS, O2A) | Failed Locks | Reference Committee Locks & Aborts | Local Recovery Local Primary Failure |
| CHAINSPACE [1] | Distributed Commit locking Locks | 11 (2) | $2s$ | 3 | 2 (CS, A2A) | Failed Locks | Distributed Commit Locks & Aborts | Local Recovery Local Primary Failure |
| RINGBFT [51] | Linear Commit Blocking Locks | $8s - 5$ $(2s - 2)$ | $2s - 1$ | $2s - 1$ | $2s - 2$ (CS, O2O) | Invalid Only | Linear Commit Blocking Locks | Local Recovery Local Primary Failure |
| SHARPER [4] | Multi-Shard Consensus Shard-Wide Blocking Locks | 3 (3) | $s$ | 1 | 3 (GC, A2A) | Failed Locks (Shard-Wide) | Multi-Shard Consensus Shard-Wide Locks & Aborts | Global Recovery Any Primary Failure, Concurrency |

[a]Total number of consecutive communication phases. For protocols that use a local consensus protocol, we count three consecutive phases per consensus step (e.g., using PBFT), and we count a single phase per cluster-sending step.
[b]We write *CS* to indicate *cluster-sending* and *MS* to indicate *multi-shard consensus*; and we write *A2A* to denote all-to-all communication, O2A to denote one-to-all or all-to-one communication, and O2O to denote one-to-one communication between involved shards.

Figure 8: Comparison of the three CHIMERA protocols for processing a transaction that affects *s* shards. We compare the normal-case complexity. the mechanism used to deal with concurrent transactions (due to malicious clients), and the mechanisms used to provide failure recovery.

transaction τ can lead to view-changes in all shards affected by τ.

In conclusion, we see that the three CHIMERA variants each make their own tradeoff between *normal-case costs* and ability to deal with faulty behavior (by both clients and other replicas), with RCHIMERA being robust against any attack at the cost of 2 consensus decisions per transaction per involved shard.

## 8.2   Comparison With the State-of-the-Art

Several recent papers have proposed specialized systems that combine sharding with consensus-based resilient systems. Examples include systems such as AHL [17], BYSHARD [34], CAPER [3], CHAINSPACE [1], RING-BFT [51], and SHARPER [4], which all use sharding for data management and transaction processing. Next, we compare the design of CHIMERA in detail with AHL [17], CHAINSPACE [1], RINGBFT [51], and SHARPER [4], and briefly look at BYSHARD [34] and CAPER [3].

**AHL [17].** AHL uses a *centralized* commit protocol to order all multi-shard transactions. In specific, AHL [17] uses a reference committee that leads a *centralized two-phase commit protocol* (Centralized 2PC) [29, 49] that is implemented via consensus steps and cluster-sending. Furthermore, AHL uses non-blocking locks to provide transaction isolation due to which valid transactions can be aborted, whereas in CHIMERA only faulty transactions (e.g., by malicious clients) are aborted. By using Centralized 2PC, AHL eliminates any all-to-all communication between shards affected by a transaction in favor of one-to-all communication between

the reference committee and the affected shards. Due to this, AHL takes five consecutive consensus rounds, more than twice the number of rounds required by the costliest CHIMERA variants. As reported in the original evaluation of AHL [17, Section 7.3], the reference committee will become a bottleneck for performance when processing workloads heavy in multi-shard transactions (even if none of these transactions are concurrent), while AHL shows excellent performance when processing single-shard transactions [34].

**CHAINSPACE [1].** CHAINSPACE uses a *distributed two-phase commit protocol* (Distributed 2PC) [29, 49], that is implemented via consensus steps and cluster-sending, to order all multi-shard transactions. Furthermore, similar to AHL, CHAINSPACE uses non-blocking locks to provide transaction isolation due to which valid transactions can be aborted. The operations of this commit protocol are similar to the design of RCHIMERA, except that CHAINSPACE does not take advantage of any specific properties of the data model (e.g., to provide isolation). A further minor difference between CHAINSPACE and RCHIMERA is that CHAINSPACE distinguishes between shards that are used as inputs and shards that are used as outputs and only informs output shards after the input shards finish processing a transaction, due to which transaction processing in CHAINSPACE takes one more round as in RCHIMERA.

**RINGBFT [51].** RINGBFT uses a *linear two-phase commit protocol* (Linear 2PC) [29, 49], that is implemented via consensus steps and cluster-sending, to order all multi-shard transactions. Due to the usage of Linear 2PC, RINGBFT is able to utilize blocking locks in a deadlock-free manner to

provide transaction isolation. Due to this usage of locks, RINGBFT is the only protocol besides CHIMERA that is able to always process valid transactions without spurious aborts. Furthermore, the usage of Linear 2PC minimizes cross-shard communication costs, as all communication is between pairs-of-affected-shards (no all-to-all, one-to-all, or all-to-one communication). The benefits of RINGBFT come at a cost, however, as the linear design imposes a *linear* amount of consecutive consensus and cross-shard communication steps in terms of the shards affected by the transaction, whereas all other proposals require a constant number of consecutive steps.

**SHARPER [4].** SHARPER uses a *multi-shard consensus protocol* to order all multi-shard transactions. The operations of this multi-shard consensus protocol are conceptually similar to the design of OCHIMERA, except that SHARPER does not take advantage of any specific properties of the data model (e.g., to provide isolation or to simplify recovery). Furthermore, SHARPER requires that affected shards process their multi-shard transactions in a common processing order, due to which SHARPER can only processing a single multi-shard transaction at a time. In effect, this imposes a per-shard lock on multi-shard transaction processing, limiting concurrent execution even in the absence of transactions that content for the same data objects. Finally, the philosophy of SHARPER is to serve as a single unified protocol that can support both PAXOS-style crash fault-tolerance and malicious behavior, and it remains an important research question as to whether SHARPER can be extended to the general-purpose unreliable communication and attack models supported by OCHIMERA. In specific, we believe OCHIMERA improves on the resilience of SHARPER by providing a *robust* local and global view-change mechanism that can deal with per-shard replica failures, per-shard primary failures, and coordinated attacks by replicas and clients to disrupt global consensus steps.

**BYSHARD [34] and CAPER [3].** BYSHARD [34] proposes a framework in which one can evaluate many distinct protocols based on the application of two-phase commit and two-phase locking in a consensus-based environment. Specific instances of BYSHARD correspond with the approaches taken by CHAINSPACE and RINGBFT, while AHL can be seen as a restricted case of the BYSHARD protocols that utilize distributed orchestration. The differences between, on the one hand, CHIMERA and, on the other hand, AHL, CHAINSPACE, and RINGBFT, extend to the BYSHARD framework. The design of CAPER [3] shares similarities with the design of SHARPER.

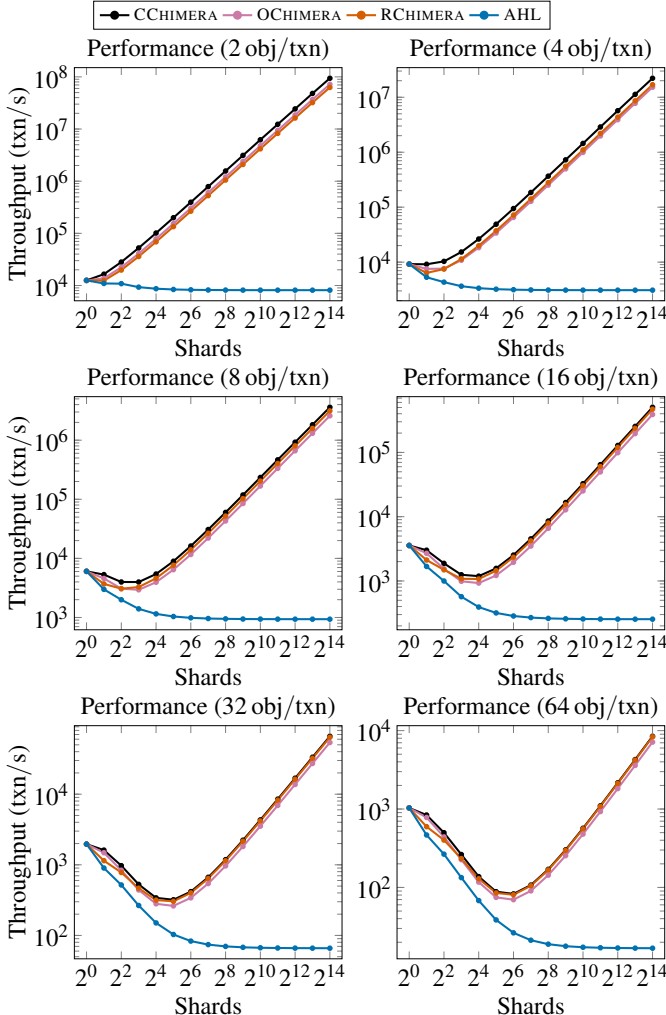

Figure 9: Throughput of the three CHIMERA protocols as a function of the number of shards.

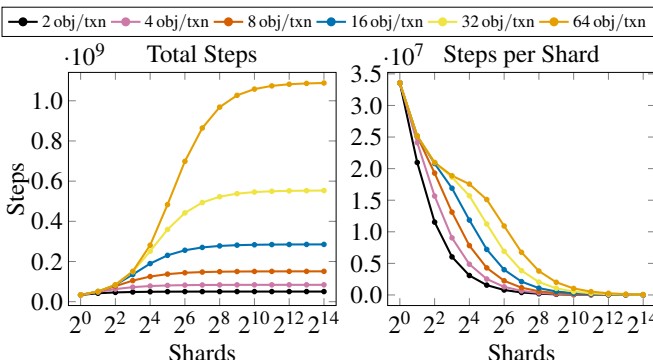

Figure 10: Number of consensus steps (amount of work) in terms of the number of transactions that affect a shard.

## 8.3 The Performance Potential of CHIMERA

Finally, we modelled the performance benefits of CHIMERA. To do so, we have modeled the maximum throughput of each of these protocols in an environment where each shard has seven replicas (of which two can be faulty) and each replica has a bandwidth of $1\,\mathrm{Gbit\,s}^{-1}$. We have chosen to optimize CCHIMERA, OCHIMERA, and RCHIMERA to minimize *processing latencies* over minimizing bandwidth usage, as reducing processing latencies is the goal of the design of CHIMERA. In specific, we do *not* use request batching, we use a one-phase broadcast-based cross-shard exchange steps, and we do not use *threshold signatures*. In cases when one does not want to optimize for processing latencies and individual replicas have spare computational power, then one can utilize threshold signatures to further boost throughput by a constant factor (at the cost of the per-transaction processing latency).

As a baseline for comparison, we have also included AHL [17]. For AHL, we used an additional shard as a reference committee (hence, if we use *n* shards in the experiment, then AHL can use $n+1$).

In Figure 9, we have visualized the maximum attainable throughput for each of the protocols as a function of the number of shards and as a function of the number of objects affected by each transaction when processing a workload with 50% multi-shard transactions. In Figure 10, we have visualized the number of per-shard steps performed by the system (for CCHIMERA and OCHIMERA, this is equivalent to the number of per-shard consensus steps, for RCHIMERA this is half the number of per-shard consensus steps).

As one can see from these results, all three CHIMERA protocols have excellent scalability: increasing the number of shards will increase the overall throughput of the system. Sharding does come with clear overheads, however, increasing the number of shards also increases the number of shards affected by each transaction, thereby increasing the overall number of consensus steps. This is especially true for very large transactions that affect many objects (that can affect many distinct shards). Hence, as one can see from the results, the benefits of sharding only truly add up for large multi-shard transactions when scaling beyond the size of individual transactions.

In comparison with AHL, we see a large improvement in performance. Unfortunately, due to the high ratio of multi-shard transactions, the performance of AHL is hindered by the throughput of the reference committee used by AHL. These findings are in line with the original evaluation of AHL [17, Section 7.3]. A closer look at the data does reveal *excellent* scalability of AHL with regards to single-shard transactions: the load of all shards *except* the reference committee drops drastically when increasing the number of shards.

## 9 Related Work

Distributed systems are typically employed to either increase reliability (e.g., via consensus-based fault-tolerance) or to increase performance (e.g., via sharding). Consequently, there is abundant literature on such distributed systems, distributed databases, and sharding (e.g., [49, 53, 54]) and on consensus-based fault-tolerant systems (e.g., [10, 14, 19, 30, 53]). Furthermore, in Section 8.2, we reviewed related work on multi-shard permissioned consensus-based systems. Next, we focus on other works that deal with sharding in fault-tolerant systems.

A few fully-replicated consensus-based systems utilize sharding at the level of consensus decision making, this to improve consensus throughput *without* adopting a multi-shard design [2, 22, 26, 31]. In these systems, only a small subset of all replicas, those in a single shard, participate in the consensus on any given transaction, thereby reducing the costs to replicate this transaction without improving storage and processing scalability.

Recently, there has also been promising work on sharding and techniques supporting sharding for permissionless blockchains. Examples include techniques to enable sidechains, blockchain relays, and atomic swaps [23, 24, 35, 37, 41, 56, 58], which each enable various forms of cooperation between blockchains (including simple cross-chain communication and cross-chain transaction coordination). Unfortunately, these permissionless techniques are several orders of magnitudes slower than comparable techniques for traditional fault-tolerant systems, making them incomparable with the design of CHIMERA discussed in this work.

## 10 Conclusion

In this paper, we took a new look at the problem of multi-shard transaction processing in consensus-based systems. In specific, we proposed the study of *sharded consensus-based systems* that use restrictions on the workloads supported to improve performance over general-purpose methods.

To initiate this study, we introduced Core-CHIMERA, Optimistic-CHIMERA, and Resilient-CHIMERA, three fully distributed approaches towards multi-shard fault-tolerant transaction processing. The design of these approaches is geared towards processing UTXO-like transactions in sharded distributed ledger networks. Due to the usage of UTXO-like transactions, the three CHIMERA variants can minimize cost to an absolute minimum, while maximizing performance, thereby showing the potential of *restricting the types of supported workloads*. This potential is further underlined by our comparison with the state-of-the-art protocols, in which we see that the three CHIMERA variants both have lower costs and complexity.

Although the workloads supported by CHIMERA are minimalistic, we believe that our results can be generalized to more-general settings. In specific, we believe that the combi-

nation of sharding and *Conflict-free Replicated Data Types* (CRDTs) [44] has great potential to provide high performance in a consensus-based environment.

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
