# OpenReview forum: "Problem: Sharded Resilient Transaction Processing with Minimal Costs"
_JSYS/2022/Oct_Papers — Revise_

### Official Review · Reviewer_X8kA · 2022-11-01
**I think this paper can be made appropriate for JSys with a little reframing.**

**Decision:**

Weak accept: good paper with flaws that can be fixed in three months

**Review:**

Summary
This paper presents a problem new to the JSys community, specifically:
    "Can one reduce the cost of coordination in the design of sharded consensus-based systems by limiting the types of workloads supported?"
The authors define a "sharded consensus-based system," and demonstrate that coordination costs can be reduced in practice using their own "CChimera" family of protocols, with comparison to previous protocols for unlimited workloads.

The paper argues that this problem is important, as it impacts the design of performance-sensitive sharded blockchains, which may implement sharded consensus-based systems. Such blockchains may want to limit their workloads in order to improve performance using protocols like CChimera.


Strengths
- problem is relevant to the community
- problem seems relatively important
- formalization of "sharded consensus-based system" seems useful, at least in the "UTXO-like" restriction.
- CChimera is indeed "optimal" under their definitions

Weaknesses
- The paper spends a lot of time on its proposed solution (CChimera), beyond motivating the problem itself or explaining why their formalization is the right one.
- It is not clear to me that the "sharded consensus-based system" formalization accurately reflects the designs used in sharded blockchains. For example, Polkadot and Eth2 both use a central "beacon" chain, which actually executes consensus, to checkpoint all the shards (which need no consensus of their own).
- The limitations of CChimera seems so weak that they allow a considerably weaker system than "sharded consensus" to provide very similar guarantees, with less latency. See detailed comments.
- There are no proofs that the improvements over AHL represented by OChimera or RChimera are inherent to "limiting the types of workload supported." Indeed, it seems possible to implement an extremely general state machine on top of either, and achieve low cost of coordination. This calls the seemingly positive answer to the central question into doubt.

Detailed Comments
Reviewer 3UEy and I both noticed that, for CChimera's limited types of workloads, it seems that a sharded consensus-based system may be unnecessary in the first place, when something simpler will suffice.  I will try to explain what I mean. Consider a simpler (non-consensus) based system, which would provide liveness for "honest" owners:
- A node recognizes that a UTXO in OUTPUTS(T) has been created when it receives a message ACK(T) from every member of a quorum from each of SHARDS(T).
- A node N on shard S broadcasts ACK(T) when it receives a transaction T, and N recognizes that all resources in INPUTS(T) on shard S exist and have not yet been consumed, and T is valid. It then marks all resources in INPUTS(T) on shard S as consumed (in N's local storage).
- Safety:
  - No 2 TXs can consume the same resource (it'd be marked as consumed by a quorum intersection in that resource's shard).
  - No resource in OUTPUTS(T) can be recognized as created until all its inputs are consumed.
- Liveness:
  - If only one TX is authorized to consume a given resource, then everything terminates in 1 broadcast.
  - A TX which consumes as input a contested resource may stall indefinitely.
  - A TX which consumes as input a not-yet-created resource may stall indefinitely.
- I note that transitive dependence on contested resources could result in stalling, which might be a weakness against CChimera. I think this violates R6.
- HOWEVER: you could require that clients themselves pass along a "proof this UTXO was created" to anyone else wanting to use the UTXO as input, and then you would get R6. Such a proof could consist of aggregated ACK messages from the transaction that created the UTXO. In fact, if you do this, nodes only have to store "consumed" sets. They don't even have to store "known to exist" sets.
This simpler setup calls into question the usefulness of CChimera as a proof that limiting workload types can  decrease cost of coordination in a sharded consensus-based system. It's not surprising that we can reduce coordination cost when we didn't really need a sharded consensus-based system in the first place.

In their response to previous reviewers, the authors promised to clarify "how to generalize OChimera and RChimera to general-purpose workloads." I'm not sure this is clear in the current paper. Moreover, I think it undercuts their central thesis. If it is possible " to generalize OChimera and RChimera to general-purpose workloads," then, given that OChimera and RChimera perform well, it is not clear whether one can "reduce the cost of coordination in the design of sharded consensus-based systems by limiting the types of workloads supported?" I thought the authors' intent was to "give a preliminary positive answer."

Some of the terminology in section 3 is not uniform. For example, you declare that "Processing of T does not imply execution: the transaction could be invalid...," but then in the "high-level three-step approach toward processing..." From section 4, you "commit T if all affected shards...", which sounds like "execution" (by your earlier definition), not processing. Indeed, processing and execution are often conflated.

I am not clear on the purpose of the "Internal propagation" and "concurrent resolution" techniques after Definition 4.1. Notably, they are not used in the proof of Theorem 4.3, so they seem unnecessary.

I am not clear on the justifications for the difference between OChimera's consensus and PBFT. Why not choose a single PBFT leader from any leader of any shard, and then use standard PBFT? Since OChimera consensus fails if any leader chosen is corrupt, picking a single leader would be strictly more likely to succeed.

It seems to me that RChimera represents a 2-Phase-Commit protocol, featuring 1 consensus phase in which all resources are locked, and then one in which the locks are released with either a commit or abort. Unless I am wrong, phrasing it this way (explicitly using 2-Phase-Commit terminology) would likely make your description shorter and easier for distributed systems researchers to quickly understand.

The paper should probably note that replicas should be able to locally determine whether a transaction is "valid" or "well-formed" or obeys system invariants (e.g. no double-spends), and only then is it "applicable." Right now, it just says  "The validity of transactions is a local requirement: whether a transaction τ is valid can be determined by checking whether all owners of inputs of τ support that transaction."

I think this paper can be made appropriate for JSys with a little reframing.
1. This is a problem paper, so focus on the problem: what is are the "unlimited" types of workloads, and for what applications do you think "limited types of workloads" will be acceptable?
2. Spend more time explaining how your formalization of sharded consensus-based systems is the right one. What existing or proposed systems does it accurately describe?
3. Given that Reviewer 3UEy and I both immediately tried to think of a way to meet CChimera's use case without consensus, directly address either why we are mistaken (for which a more formal proof may be required than was in the rebuttal), or argue that your answer is only "preliminary," and CChimera proves that limited workloads can improve performance, but the search for usefully limited workloads that still require consensus-based sharded systems is ongoing.
4. What precisely is the purpose of OChimera and RChimera in presenting this problem?
  - If OChimera and RChimera can only address limited workloads (but not as limited as CChimera), point that out, as they form additional "positive" answers to your question.
  - If OChimera and RChimera can address unlimited workloads, you need to explain exactly how to convert, say, AHL style transactions to RChimera transactions. Moreover, if you still want to argue that one can "reduce the cost of coordination in the design of sharded consensus-based systems by limiting the types of workloads supported," you have to emphasize performance improvements (either theoretical, practical, or both) of CChimera over RChimera or OChimera, and preferably show that these are fundamental, rather than stemming from design choices made. Comparing OChimera and RChimera to AHL etc. then serves mainly to prove that you're comparing to "better than state of the art," rather than a straw-man.




Minor Remarks

Introduction, paragraph 3: I'm not sure it's technically true that permissions blockchains can have higher throughput or that they must be fully-replicated. Permissioned chains can use sharded storage, and non-permissioned proof-of-stake chains can use the same consensus mechanisms (e.g. PBFT) as permissions ones.

 "CChimera will operate perfectly" : you have to define "perfectly." In particular, you'll have to argue that one round of cross-shard broadcast is indeed "perfect."

Definition 3.1: define "decisions."

Definition 3.1 part 3: what does it mean to "force consensus." This is clearly getting at some kind of censorship resistance, but it's not spelled out, and it's not clear that known consensus protocols have this.

R4: Makes it sound like a system can commit a transaction without having processed it. This doesn't sound right.

R5: "Shard-applicable" is defined only with reference to a time. Here, do you mean eventually shard-applicable? Eventually always shard-applicable?

Section 4, "The good replicas in S will first determine whether T is valid and applicable": applicable at what time? We haven't processed it yet.

Section 4: "The primary P(S)" unstated assumption: this is the first time we've assumed we have a something called a primary.

Section 4: "Hence, there is the danger of deadlocks if the other shards S ′ never perform their crossshard exchange steps." You never explain how deadlock would be avoided if the other shards S' never perform their cross shard exchange steps. In fact, I'm pretty sure it would deadlock if  the other shards S' never perform their cross shard exchange steps. Internal propagation and Concurrent resolution seem unrelated to this point.

Section 4: ": CChimera effectively eliminates all coordination when deciding to process a multi-shard transaction" No it doesn't. It requires a phase of cross-shard broadcast, before which a transaction isn't really "committed."

Section 5: "View v of S" is used before it is defined (Is it defined?)

Section 5: The use of the term "Commit" within the processing (not execution) portion of the protocol is confusing. It is different from the commit in the execution part of the protocol.

Figure 5, "event R ∈ S receives messages ⟨ VCRecoveryRQ : v i,ρ,S i⟩ , 1 ≤ i ≤ f S +1, from f S +1 distinct replicas in S do" is confusing. Do the subscripts really need to be 1..f_S+1? Why not just use _ for v_i and S_i, since we don't use those indices?

Figure 6, line 15: is the comma meant to denote "AND"? How do I read this?

Proposition 5.3: "optimistic case" is not well defined.

Section 5: "other multi-shard consensus protocols" such as? What exactly is a multi-shard consensus protocols? Citations?

Section 6: suddenly it is possible to re-construct objects in D(S,T)? We've never reconstructed objects before. Couldn't we just mark them as "in use" or something?

Section 7: "It is this flexibility in dealing with aborted transactions that allows all variants of C HIMERA to operate with minimal and fully-decentralized coordination between shards" It is? It's not clear why this flexibility is necessary or sufficient.

Figure 8: the sequential column and the cross-shard communication column are a bit confusing. CChimera requires 1 consensus step + one cross-shard communication step, while OChimera's 1 consensus step includes the 3 cross-shard communication steps.

Figure 8: OChimera ends up looking very similar to SharPer. Did you need to introduce OChimera, or could you just have used SharPer?

Figure 8: What does the word "Incentive" mean here?

Figure 9: With large numbers of objects, shouldn't most multi-shard transactions use all shards when there are only a few shards? Why does Chimera do any better than AHL?

Figure 9: Are you picking objects uniformly at random? It's not clear how many shards each transaction touches.

Figure 10: Why is AHL not in this figure? Also, how does OChimera ever beat CChimera? It the underlying consensus even slower than OChimera's PBFT variant?

Figure 11: "Step" here is not defined until the bottom of the page, and even then it's defined implicitly in the parenthetical "(for CC HIMERA and OC HIMERA, this is equivalent to the number of per-shard consensus steps, for RC HIMERA this is half the number of per-shard consensus steps)."

Section 8: "In environments in which inter-shard and intra-shard communication have similar (high) message delays, OC HIMERA will typically have lower latencies than CC HIMERA due to the large impact message delays have on the latency of consensus and cluster-sending steps, this even when CC HIMERA has higher throughput than OC HIMERA." I don't follow. How many message delays does each have? Are they different?

Figure 12: "Performance" did you mean "throughput"?

Figure 12: "malicious shards" did you mean "shards with a malicious participant"?





**Expertise:**

Actively publishing in this area

**Useful:**

yes

---

### Official Review · Reviewer_yNuG · 2022-11-14
**PROBLEM: SHARDED RESILIENT TRANSACTION PROCESSING WITH MINIMAL COSTS**

**Decision:**

Weak accept: good paper with flaws that can be fixed in three months

**Review:**

**Summary of the paper**
The paper looks into the problem of minimizing required coordination to process transactions in a sharded consensus-based system. Specifically, the paper investigates whether limiting the types of supported workload can significantly reduce the required coordination among shards. The paper proposes the CHIMERA-family protocols (i.e., CCHIMERA, OCHIMERA, and RCHIMERA), which are used to process UTXO-transactions under different assumptions about the environment. Modeling results show that CHIMERA-family protocols reduce the coordination among shards compared to alternative solutions and achieve better scalability with number of shards.

**Comments for authors**
* The idea of building a protocol to address a specific kind of workload is novel and interesting. Below are my comments that could help improve the paper.
* The introduction motivates the use of distributed ledger technology (DLT), but it does not motivate the idea of limiting the workload to UTXO-like transactions. However, I think the paper should justify the choice of UTXO-like transactions, discuss some examples of where they are being used, and whether they are prevalent or not.
* Can CHIMERA protocols be extended to support other types of transactions? If so, the paper would benefit from discussing that.
* I felt that that the discussion of RCHIMERA is shallow. For instance, why do replicas have to force a second step of consensus?
* The paper mentions at different places that, for several steps, either all-to-all or an all-to-one-to-all communication can be used, however, it does not state any conditions on the “one” replica in the all-to-one-to-all communication. Does this replica must be good? What if it is malicious?
* Page 6: “CCHIMERA effectively eliminates all coordination when deciding to process a multi-shard..”. This statement seems inaccurate as “Cross-shard exchange” step has to be performed before reaching a decision about a transaction.
* One main concern I have is that the evaluation results are based on modeling and not based on real implementation of the proposed protocols. So, I am not sure to what level the results can be trusted as modeling might omit significant details. Nonetheless, below are my comments about presented results.
* Figure 8 compares only the number of consensus steps needed for each protocol during the normal case. I guess it would be more insightful if it includes the complexity of each step (maybe as number of messages) as well as the number of steps (phases) needed for recovery mechanism.
* I felt the selected systems for comparison is a bit biased.
	- page 17: “We excluded CHAINSPACE, as it has a design very similar to RCHIMERA (but slightly more costly due to some overheads RCHIMERA can avoid).” I guess you should include CHAINSPACE in order to evaluate how much you gain by avoiding these overheads.
	-  The throughput experiment (Figure 9) compared against AHL while the latency experiment (Figure 10) compared against RINGBFT. The paper states: “RINGBFT is optimized to minimize communication costs and maximize throughput (by inducing very high latencies for multi-shard transactions)”. So, I guess for fair comparison, both Figure 9 and Figure 10 must include AHL and RINGBFT.
	- Why does Figure 12 not include other systems?

* Nits
	- Page 5, 11, 13, 15: “that content” -> that contend
	- page 6: “in a asynchronous” -> an asynchronous
	- Figure 8 has “GC” symbol but caption does not state what does it mean
	- Figure 10 misses the X-axis title
	- Figure 12 misses the Y-axis title


**Expertise:**

Follow the literature closely, last published 5+ years ago

**Useful:**

yes

---

### Official Review · Reviewer_MJ9g · 2022-11-24
**Paper Review**

**Decision:**

Weak reject: interesting papers with flaws, not sure if they can be fixed in three months

**Review:**

This paper aims to improve the performance of blockchain systems by allowing the processing of multi-shard transactions. The authors propose a family of multi-shard transaction processing protocols, namely CHIMERA. I do think the problem is important, and the proposed solutions are reasonable. However, I have major concerns at the problem formulation and evaluation. In short, I couldn’t follow them – the problem is not clearly defined, nor well justified. The evaluation is based on some sort of simulation or modeling, but I don’t think it meets the bar of a good system journal. In general, I feel that the model and the specification are mixed up, which makes it difficult to judge what exactly the system is trying to do. More specifically,

- Assumption 2.2 starts with how the transaction should look like -- which I personally think should be a specification, rather than an assumption. It's also non-typical that each transaction needs to have an input. Doesn't that rule out PoW? Or can the input field be a dummy one or a null? This needs to be explained so that readers have a clear picture.
- Owner of an object is not defined.
- It's not clear what it means by "shards that are affected by a transaction." Is an input object affected? If yes, why?
- It's also strange that the model assumes that "a set of replicas operated as a Byzantine fault-tolerant (consensus) system." What if a shard adopts a weaker abstraction? E.g., a Byzantine atomic register? Or a DAG-based system? They don't exactly use consensus. If the model/solution only works for consensus-based systems, then this should be clearly specified early in the introduction.
- Even if a shard is operating on a consensus-based system, the following assumption is not precise: "As these systems are fully-replicated, each replica holds exactly the same data." There could be delay, so it's possible that a replica is straggling and doesn't have the up-to-date data.
- It's not clear what "reliable ordered replication" means. This doesn't seem to be a standard term. Please define it clearly.
- Definition 3.1 is quite strong. It rules out most randomized consensus systems, e.g., PoW or PoS which may switch on different chains. Therefore, I'm not sure how practical the proposed solution is.
- The correctness condition requires each input object owner to approve the transaction, which seems too expensive or even impossible in practical systems. What if the owner is faulty or slow? In real workload, it's possible that an object is a hotkey, which will penalize the owner, because it needs to approve tons of transactions. I'm not sure this is a realistic or favorable assumption to adopt.
- It's not clear to me what the difference is between commit and execute. Please define.
- R2 is not very clear. Why can't the system off-load processing to other shards or systems?
- In R3, \tau "consumes only existing objects." This requirement is not clear to me at all. What is time t? What if some object does not exist at time t, but appears at time t+1. Should the transaction be committed? What if the object exists at time t, but then later got removed at time t+1? Should the transaction be committed?
- I don't understand why validity is a local property. What if the owners in shard S all approved the transaction, but the owners in another shard S' disapproved the transaction. I thought this transaction should be viewed as invalid, but then each shard alone cannot check for the validity condition.
- I personally couldn’t interpret the evaluation section at all. I understand that it’s some sort of simulation based on some performance modeling, primarily on communication patterns. However, I don’t see many details for me to understand the meaning of the figures. It’s stated that the network is 1 Gbit/s, and there is some latency (message delay = 15ms in Figure 10). However, more information is needed. Is this a typical number? Are we looking at only a cluster setup? Why? Wouldn’t the wide-area network setting be more reasonable for the targeted case? If the number is an average, why? What is causing fluctuations of the performance? (It's “performance modeling” anyway. Is the delay varying?) What is the size of the transaction? Why 7 and 61 replicas per shard? Are they typical? Why only testing 50% of multi-shard transactions? Does the percentage matter? Why isn’t AHL tested in Section 8.4? Why only a particular Byzantine strategy is tested? In Figure9, why is it a v shape for certain configurations?

In short, there are many missing details that prevent me from fully appreciating this submission, nor understanding the performance benefit.

**Expertise:**

Actively publishing in this area

**Useful:**

yes

---

### Meta-Review · Area_Chair_wpC2 · 2022-11-24

**Recommendation:** Revise
**Confidence:** 5

**Metareview:**

Dear Authors, thank you for submitting to JSys. After reading the reviews, we recommend a REVISE decision. We believe the concerns of reviewers can be addressed in the next three months (potentially sooner, but we wanted to give you enough time to handle the comments). After the submission of your revised version, reviewers will evaluate your modifications and a final decision will be made.

Two reviewers appreciate the contributions and are mostly happy with the revision from the previous submission. The other reviewer pointed out major concerns in the problem formulation and evaluation. In particular, unclear definitions and evaluation made it difficult for the reviewer to evaluate the paper with confidence.  In general, most reviewers are optimistic for a future acceptance of the paper; hence, we believe all concerns can be addressed during the paper revision. Reviewers have included suggestions for improving the paper, please consider them.

Isolating two major concerns:
- Improve problem formulation and motivation. Consider addressing the comments in your revision.
- Improve the evaluation, particularly better explanation of the results and setup.

We look forward to receiving your revised version. Thank you one more time for submitting,

Lewis Tseng and Roberto Palmieri Area chairs of JSys

---

### Decision · Program_Chairs · 2022-11-24

Revise